# The 2025 Foundation Model Transparency Index

**Alexander Wan**[†]
*UC Berkeley*

**Kevin Klyman**[†*]
*Stanford University*

**Sayash Kapoor**[†]
*Princeton University*

**Nestor Maslej**
*Stanford University*

**Shayne Longpre**
*Massachusetts Institute of Technology*

**Betty Xiong**
*Stanford University*

**Percy Liang**
*Stanford University*

**Rishi Bommasani**[†]
*Stanford University*

**Reviewed on OpenReview:** *https://openreview.net/forum?id=1jT253Xtyf*

## Abstract

Foundation model developers are among the world's most important companies. As these companies become increasingly consequential, how do their transparency practices evolve? The 2025 Foundation Model Transparency Index is the third edition of an annual effort to characterize and quantify the transparency of foundation model developers. The 2025 FMTI introduces new indicators related to data acquisition, usage data, and monitoring and evaluates companies like Alibaba, DeepSeek, and xAI for the first time. The 2024 FMTI reported that transparency was improving, but the 2025 FMTI finds this progress has deteriorated: the average score out of 100 fell from 58 in 2024 to 40 in 2025. Companies are most opaque about their training data and training compute as well as the post-deployment usage and impact of their flagship models. While companies tend to disclose evaluations of model capabilities and risks, limited methodological transparency, third-party involvement, reproducibility, and reporting of train-test overlap pose challenges. In spite of this general trend, IBM stands out as a positive outlier, scoring 95, in contrast to the lowest scorers, xAI and Midjourney, at just 14. Several groups of companies score higher than the mean: open model developers, enterprise-focused B2B companies, companies that prepare their own transparency reports, and signatories to the EU AI Act General Purpose-AI Code of Practice. The five members of the Frontier Model Forum we score end up in the middle of the Index: we posit that major companies aim to avoid particularly low rankings but also lack incentives to be highly transparent. As policymakers around the world increasingly mandate certain types of transparency, this work reveals the current state of transparency

---

[*]In October 2025, Kevin Klyman began a role at Google. All FMTI 2025 scores were finalized before this date and he was not involved in the project after this date. His contributions were independently reviewed by Rishi Bommasani.
[†] indicates equal contribution. Direct correspondence to nlprishi@stanford.edu.

for foundation model developers, how it may change given newly enacted policy, and where more aggressive policy interventions are necessary to address critical information deficits.

# 1 Introduction

AI companies are vital to the global economy. While the technology they build, namely foundation models, garners significant attention, the companies are themselves distinctive. For example, OpenAI is the most valuable private company in the world (Hammond & Kinder, 2025), and Anthropic is one of the fastest-growing technology companies in history (Hammond & Criddle, 2025). And the technologies they build have wide-ranging societal impacts. AI is the fast-adopted technology in history accumulating 1.2 billion consumer users in 3 years (Microsoft, 2025) while enterprises are rapidly integrating AI into core business function (Appel et al., 2025). Given the key role these companies play in shaping the future, transparency regarding their practices is an essential public good. Transparency about how AI companies operate is necessary to ensure corporate governance, mitigate societal harms from AI, and promote competition.

However, transparency is a nebulous concept—and imprecision about what it means and how it should be operationalized for AI introduces confusion and inhibits progress. On the other hand, measurement quantifies the status quo and orients progress. Measurement efforts in AI concentrate on benchmarking the capabilities and risks of AI as a technology. While critical, more emphasis is needed on the measurement of AI companies. These companies make the core decisions that shape the technology and it is their incentives that mediate the trajectory of AI development. In particular, the measurement of the transparency of AI companies is vital both for tracking the state of the current AI ecosystem and for encouraging developers to adopt more transparent practices.

The Foundation Model Transparency Index (FMTI) is a measurement instrument specifically designed to measure the transparency of AI companies. This paper describes the 2025 Foundation Model Transparency Index, which is the third edition of annual index that began in 2023. The general method of the Index can be decomposed as follows: (i) designing indicators that serve as the scoring criteria for transparency, (ii) selecting major foundation model companies to assess, (iii) gathering information on companies' practices, (iv) scoring companies on indicators based on gathered information, and (v) engaging companies to cooperatively clarify their practices and incrementally improve their disclosures. Compared to the previous edition, the 2025 FMTI adjusts the methodology for the first three steps. We update the indicators to reflect the current AI ecosystem and expand the set of companies engaged by contacting 23 and scoring 13 companies (AI21 Labs, Alibaba, Amazon, Anthropic, DeepSeek, Google, IBM, Midjourney, Mistral, Meta, OpenAI, Writer, xAI), which includes Chinese companies for the first time. To gather information, we ask companies to submit transparency reports that disclose their practices as we did in 2024. While the majority of companies did so, some key companies did not. Therefore, we manually gather information about 6 companies (Alibaba, Anthropic, DeepSeek, Midjourney, Mistral, xAI) as the basis for our scoring. In this process, we also explored the information retrieval capabilities of AI agents, finding that agents can meaningfully improve information discovery on company practices. However, AI agents still fall short of completely replacing this discovery process: the FMTI team still manually reviewed each piece of information retrieved by the agent. Overall, the 2025 FMTI took our team a year to execute given its complexity and extensive engagement with companies.

We find that the overall level of transparency in 2025 is low: companies score 40.69 out of 100 on average. Companies can be divided into three groups: the top scorers (IBM, Writer, AI21 Labs; average = 78), the middle (Anthropic, Google, Amazon, OpenAI, DeepSeek, Meta, Alibaba; average = 36), and the bottom scorers (Mistral, Midjourney, xAI; average = 15). In particular, IBM scores the highest in FMTI history at 95/100, including by disclosing 6 indicators that no other company discloses. Overall, this heterogeneity reveals that current transparency practices most directly reflect the priority placed on transparency, instead of systemic pressures that incentivize or discentivize transparency.

Breaking down the overall scores by topic, companies are most opaque but also the most varied in disclosing information about the upstream resources involved in building their flagship models. The individual topics that are the most opaque are the two critical inputs for building models, namely training data and training compute, and post-deployment outputs of the models, namely usage data and the resulting impact of models on the economy. Flagship models tend to be subject to an extensive range of evaluations, but the value of these evaluations for external actors is limited: companies often provided limited insight into evaluation methods, insufficient detail to enable external reproduction, and only some enable third-party evaluators to assess their models pre-deployment. No company adequately discloses the extent to which their models are

trained on data that overlaps with the evaluations they are tested on. For many of these specific topics, we do not currently expect transparency to improve based on current market forces, which warrants consideration of whether policy intervention is desirable to address information deficits.

Because we score a variety of companies, we can stratify results by different company-level axes of variation. Developers of open flagship models tend to be more transparent than their closed counterparts, but open developers fall into two groups: two are quite transparent (IBM, AI21 Labs; average = 81), while three of the most influential open developers in the ecosystem are relatively opaque (DeepSeek, Meta, Alibaba; average = 30). Enterprise-focused companies (IBM, AI21 Labs, Writer, Amazon) are consistently and considerably more transparent than consumer-focused companies or those that pursue hybrid business strategies: the top 3 companies on the 2025 FMTI are enterprise-focused. Five of the most important companies in the ecosystem belong to the Frontier Model Forum (Amazon, Anthropic, Google, Meta, OpenAI): these five companies all score in the exact middle of the 2025 FMTI, suggesting they share a common incentive to not score low on the index but also lack the incentive to significantly differentiate themselves based on strong transparency performance. Two pairs of these companies show high correlation in their practices ((Amazon, Google), (Anthropic, OpenAI)): Anthropic essentially dominates OpenAI by disclosing sufficient information on almost a strict superset of the indicator.[1] Signatories of the European Union's AI Act Code of Practice tend to score slightly higher than non-signatories, and US companies tend to score slightly higher than non-US companies. In both cases, the difference is mostly attributable to greater transparency on the downstream domain (e.g. more information about usage policies). Companies who prepare transparency reports themselves and engage more extensively with the FMTI on updating their disclosures score considerably higher, reflecting that transparency scores on the FMTI are considerably mediated by the effort companies put in.

The FMTI is one of the only metrics designed specifically for AI companies that has been tracked over time. Longitudinally, scores in 2025 (average = 40.69) have declined from their 2024 levels (average = 58), reversing the progress observed in 2024 back to the 2023 levels when the Index first launched (average = 37). Most companies scored in the past two years have decreased their score in 2025 with Meta cutting its score in half and Mistral by more than two-thirds.[2] Of the 6 companies assessed in all three years (AI21 Labs, Amazon, Anthropic, Google, Meta, OpenAI), much has changed: Meta and OpenAI were first and second of this group in 2023, but now are last and second-to-last, respectively. On the other hand, AI21 Labs has dramatically improved its score from 25 and in 2023 to 66 in 2025. In some cases, companies have directly regressed: they disclosed information on an indicator in the past but no longer do.[3]

The 2025 Foundation Model Transparency Index reveals the current state of public information in relation to major AI companies. By quantifying and comparing the practices of these companies, we expect it will contribute towards advancing transparency, both through the direct engagement with these companies and the indirect support of other agents of change (e.g. policymakers, journalists, clients, consumers, investors). For example, defining indicators and collating transparency reports can buttress initiatives to build industry standards and norms, including via mandated disclosure requirements. In parallel, our effort makes progress towards understanding underlying scientific questions: when is transparency genuinely at odds with other values, what are the costs of transparency?

## 2 Background on the Foundation Model Transparency Index

The inaugural edition of the Foundation Model Transparency Index was released in October 2023 (Bommasani et al., 2023a). The process was as follows: the Index team defined the original 100 FMTI indicators, compiled public information on 10 companies, scored companies based on this information against the indicators, sent these scores to the companies to rebut, and published the final results. The overall results demonstrated low levels of transparency (the average score was 37/100 and the top score was 54/100), but significant heterogeneity (82 of the 100 indicators were scored by at least one company). Developers clustered into

---

[1]The two exceptions, where OpenAI discloses sufficient information but Anthropic does not, are the AI bug bounty indicator and the data retention and deletion policy indicator.

[2]Scores went down from 2024 to 2025 as follows: Amazon (-2), Anthropic (-5), Google (-6), Meta (-29), Mistral (-37), OpenAI (-14).

[3]This is true even for top scorers in certain cases: in 2024, AI21 Labs disclosed training compute, energy usage, and carbon emissions ($6.00 \times 10^{23}$ FLOPs, $570,000 - 760,000$ kWh, $2 - 300$ tCO2eq) but in 2025 it does not.

three groups: four well-above the mean (Meta, Hugging Face, OpenAI, Stability AI), three around the mean (Google, Anthropic, Cohere) and three well-below the mean (AI21 Labs, Inflection, Amazon). Developers scored for their flagship open models (i.e. Meta, Hugging Face, Stability AI) generally scored higher often due to increased transparency about upstream resources (e.g. data, labor, compute) used to build the model. All of the companies were opaque on certain key issues, namely training data, data labor, computational costs, risks and mitigations, feedback mechanisms, and downstream impact. The 2023 Foundation Model Transparency Index received media coverage (Roose, 2023; Hao, 2023) and was incorporated into major AI policy efforts such as the EU AI Act's transparency requirements for general-purpose AI models and the Foundation Model Transparency Act introduced in Congress.

The second edition of the Foundation Model Transparency Index was released in May 2024, shortly after the first edition, to build on the initial findings and clarify the immediate response to the first edition (Bommasani et al., 2024b). The 2024 FMTI retained most aspects of the 2023 FMTI process, including the 100 indicators, but required companies to proactively submit reports (Bommasani et al., 2024c) rather than relying only on information that companies had previously made public. In spite of the short turnaround from the first edition, 14 companies submitted transparency reports as requested. In response, the FMTI team more extensively engaged employees at many of these companies over the course of several months to understand their practices and how companies made decisions about disclosures. The results demonstrated a clear improvement in transparency: the average score rose from 37 in 2023 to 58 in 2024, and the top score rose from 54 in 2023 to 85 in 2024. Due to the change in methodology, companies were able to publish information in their 2024 FMTI transparency reports that was previously not public. This was responsible for much of the improvement in transparency: companies made new information public in relation to 16.6 indicators on average with every company assessed in both years publishing new information. For three developers (AI21 Labs, Aleph Alpha, Writer), new information they disclosed constituted roughly half the points awarded. New information was concentrated in three areas: (i) the use of human labor in the creation of training data, where multiple companies clarified they do not use human labor, (ii) the use of compute to train models, where multiple companies for the first time revealed information about the number of FLOPs and hardware used to train their flagship models, and (iii) the usage policies governing user interactions with the model, where multiple companies clarified how their policies operate and are enforced. While the 2023 Index led to significant external stakeholder engagement, the 2024 Index more directly impacted company processes (companies prepared standardized reports and developed internal transparency processes for FMTI) and transparency outcomes (companies disclosed new information in their transparency reports).

The value proposition of an index depends on conducting multiple editions to clarify longitudinal trends. In particular, many aspects of the AI ecosystem have changed over the course of past year. New entrants have built high-profile foundation models (e.g. Alibaba, DeepSeek), while others have dramatically changed their business models (e.g. Aleph Alpha), organizational structure (e.g. Meta), and corporate status (e.g. OpenAI). The data ecosystem has become more complex as new data generation methods like reinforcement learning shift focus away from internet-centric pretraining as ongoing copyright litigation advances.[4] The compute ecosystem has also evolved as the computational demands of foundation model training and inference have prompted unprecedented investments into energy and data center infrastructure in the United States and around the world (Metz, 2025). The core technological paradigm has evolved with a greater emphasis on test-timing scaling and agents. This had led to increased capabilities (e.g. models achieved gold medal performance on the International Mathematics Olympiad) and risks (e.g. multiple companies indicated mitigations were necessary to bring biorisks down to acceptable levels to permit release). The growth of this ecosystem has amounted to more extensive adoption across the global economy with early empirical work clarifying how AI contributes to productivity and labor market disruption (Brynjolfsson et al., 2025). Societally, multiple jurisdictions have enacted regulation on foundation models (e.g. the European Union's EU AI Act, California's Transparency in Frontier Artificial Intelligence Act) as governments more extensively adopt AI, including for military purposes (Capoot, 2025).

---

[4]For example, Anthropic settled in Bartz vs. Anthropic for $1.5 billion (Brittain, 2025).

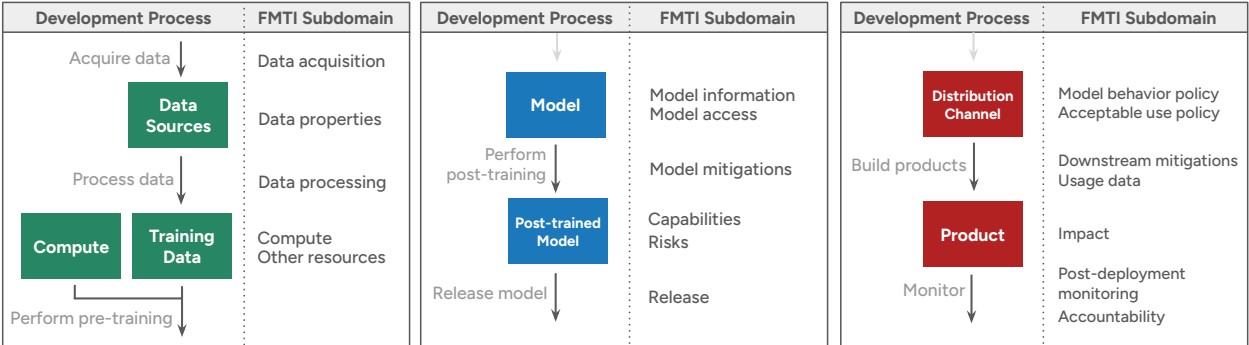

Figure 1: **FMTI 2025 Subdomains.** The 18 subdomains in the newest set of indicators compared to the supply-chain of foundation models. Our indicators address the resources used to develop models (*left*); properties of the model and release process (*middle*); and the downstream impacts of model usage (*right*).

## 3 Indicators

A transparency indicator is defined by its canonical name, a brief definition, a more detailed set of notes that articulate what practices would satisfy the indicator, and an example of a satisfactory disclosure. Indicators are organized hierarchically into subdomains and domains to facilitate multi-level analysis. To concretize transparency, Bommasani et al. (2023a) defined the original 100 FMTI indicators based on the literature on AI transparency. These indicators were used in both the 2023 and 2024 edition of the Index. However, for the 2025 edition, we make significant changes. Below we describe the old 2023 indicators, the new 2025 indicators, and the rationale for the changes.

### 3.1 2023-2024 Indicators

The 100 indicators are organized hierarchically into 3 domains (upstream, model, downstream) and 23 subdomains therein.

32 upstream indicators address the resources involved in foundation model development. Primarily, this relates to transparency around training data, labor practices, computational costs, code, and technical decisions. For example, the compute subdomain covers indicators like the amount of hardware used to train a model, the owner of that hardware, the associated computational cost and training duration, and resulting energy and environmental impacts of using that compute to build the model.

33 model indicators address the foundation model itself. Primarily, this relates to transparency around basic model properties, model access, capabilities, risks, and mitigations. For example, the risk subdomain covers whether risks are enumerated and are legible to laypersons, whether risks are evaluated pre-deployment for both unintentional (e.g. bias) and malicious (e.g. disinformation) types of risks, and whether external parties evaluate risk.

35 downstream indicators address the distribution and usage of models. Primarily, this relates to distribution practices, policies on usage, model behavior, and data protection, documentation to enable use, and downstream impact in society and the economy. For example, the distribution subdomain covers how release decisions are made by the developer, what channels are used to distribute the foundation model, whether it is integrated into other products and services by the developer, and the terms under which it is distributed.

Overall, these indicators are the byproduct of a wealth of research in these different subdomains spanning labor (Gray & Suri, 2019; Crawford, 2021; Hao & Seetharaman, 2023), data (Bender & Friedman, 2018; Gebru et al., 2018; Longpre et al., 2023b;a), compute (Lacoste et al., 2019; Schwartz et al., 2020; Patterson et al., 2021; Luccioni & Hernández-García, 2023), evaluation (Liang et al., 2023), safety (Cammarota et al., 2020; Longpre et al., 2024), privacy (EU, 2016; Brown et al., 2022; Vipra & Myers West, 2023; Winograd,

2023), policies (Kumar et al., 2022; Weidinger et al., 2021; Brundage et al., 2020), and impact (Tabassi, 2023; Weidinger et al., 2023).

## 3.2  2025 Indicators

To design the 2025 FMTI indicators, we reviewed recent literature, developments across the AI ecosystem, and our learnings for the original 2023 indicators. The indicators, which are enumerated in Figure 2, were subject to external review by the FMTI advisory board and AI researchers. The 2025 FMTI indicators continue to focus on coverage of the AI supply chain and, accordingly, are organized into subdomains that span the supply chain as shown in Figure 1. Since the high-level abstraction of the supply chain involving upstream resources, models, and downstream uses remains unchanged, we retain the same three domains.

**Upstream (Figure 3).**  The 34 upstream indicators address the resources involved in foundation model development and are organized into 6 subdomains:

- **Data Acquisition (12 indicators).** Assesses transparency regarding how and why data was acquired to build the model, such as the sources of public data, usage data, licensed data, novel human-generated data, and synthetic data. For each data acquisition method, one indicator asks the company to disclose the top-5 data sources for that method and 1-2 additional indicators address deeper information about each method (e.g. compensation for licensed data, instructions given to data laborers). *A new subdomain restructuring indicators from the previous Data and Data Labor subdomain and covering a broader range of data acquisition methods.*

- **Data Processing (3 indicators).** Assesses transparency regarding how companies transform the data they acquire into what they use to train their foundation model, including the methods, purpose, and techniques for data processing. *A new subdomain containing merged indicators from the previous Data and Data Mitigations subdomain and an indicator on the implementation of data processing methods.*

- **Data Properties (5 indicators).** Assesses transparency regarding the properties of the data used to build the foundation model, including data size, language composition, domain composition, external access, and replicability. *A new subdomain containing indicators from the previous Data and Data Access subdomain. Splits previous indicator on data sources into two more specific indicators on language & domain composition.*

- **Compute (9 indicators).** Assesses transparency regarding the hardware and computation used to build the model, as well as the resulting energy use and environmental impacts and how compute is allocated. *Largely the same as the previous editions, but more explicitly delineates between compute used for development and compute used for the final training run.*

- **Methods (3 indicators).** Assesses basic technical specifications for the model's training stages and objectives, as well as access to the code used to train the model. *Largely the same as the previous editions.*

- **Other Resources (2 indicators).** Assesses transparency regarding the cost of training the model and the structure of the organization doing so. *A completely new subdomain.*

Data acquisition and provenance remains poorly understood (Longpre et al., 2023a) even though models are trained on immense and increasing amounts of data. In particular, almost all models are trained on publicly available data either via existing datasets or through web crawling (Solove & Hartzog, 2025), which is approaching saturation point of such public data. Beyond public data, developers also employed user data (Rogers, 2025), introducing questions of adequate notice to users (King et al., 2025) and exacerbating legal and privacy risks (Tramèr et al., 2024). To acquire more data, developers also have licensed data from third parties such as news publishers and online platforms (Sweeting, 2024), though little is known about these contracts, including the total cost and remuneration for individual content creators (Nam, 2024; Tseng

**2025 Foundation Model Transparency Index Indicators**

| Upstream | Model | Downstream |
|---|---|---|
| Data acquisition methods | Basic model properties | Distribution channels with usage data |
| Public datasets | Deeper model properties | Amount of usage |
| Crawling | Model dependencies | Classification of usage data |
| Usage data used in training | Benchmarked inference | Data retention and deletion policy |
| Notice of usage data used in training | Researcher credits | Geographic statistics |
| Licensed data sources | Specialized access | Internal products and services |
| Licensed data compensation | Open weights | External products and services |
| New human-generated data sources | Agent protocols | Users of internal products and services |
| Instructions for data generation | Capabilities taxonomy | Consumer/enterprise usage |
| Data laborer practices | Capabilities evaluation | Enterprise users |
| Synthetic data sources | External reproducibility of capabilities evaluation | Government use |
| Synthetic data purpose | Train-test overlap | Benefits assessment |
| Data processing methods | Risks taxonomy | AI bug bounty |
| Data processing purpose | Risks evaluation | Responsible disclosure policy |
| Data processing techniques | External reproducibility of risks evaluation | Safe harbor |
| Data size | Pre-deployment risk evaluation | Security incident reporting protocol |
| Data language composition | External risk evaluation | Misuse incident reporting protocol |
| Data domain composition | Mitigations taxonomy | Post-deployment coordination with government |
| External data access | Mitigations taxonomy mapped to risk taxonomy | Feedback mechanisms |
| Data replicability | Mitigations efficacy | Permitted, restricted, and prohibited model behaviors |
| Compute usage for final training run | External reproducibility of mitigations evaluation | Model response characteristics |
| Compute usage including R&D | Model theft prevention measures | System prompt |
| Development duration for final training run | Release stages | Intermediate tokens |
| Compute hardware for final training run | Risk thresholds | Internal product and service mitigations |
| Compute provider | Versioning protocol | External developer mitigations |
| Energy usage for final training run | Change log | Enterprise mitigations |
| Carbon emissions for final training run | Foundation model roadmap | Detection of machine-generated content |
| Water usage for final training run | Top distribution channels | Documentation for responsible use |
| Internal compute allocation | Quantization | Permitted and prohibited users |
| Model stages | Terms of use | Permitted, restricted, and prohibited uses |
| Model objectives | | AUP enforcement process |
| Code access | | AUP enforcement frequency |
| Organizational chart | | Regional policy variations |
| Model cost | | Oversight mechanism |
| | | Whistleblower protection |
| | | Government commitments |

Figure 2: **2025 Foundation Model Transparency Index Indicators.** The new 100 indicators defined in 2025, organized into upstream, model, and downstream domains.

.

et al., 2025). In addition to existing data, developers have generated new data via both human labor and synthetic data generation. Human data labor is an established area of advocacy on human rights, especially given human-produced training data may specifically be used to address model behaviors related to risky tendencies (Al Hammada, 2024), with companies like Turing and Mercor entering this space alongside a major acquisition of Scale AI by Meta in the past year. Synthetic data has become more promising as model capabilities improve and more sophisticated pipelines involving reinforcement learning have been developed (Kapania et al., 2025). The resulting data that developers acquire is then extensively processed before being used to train foundation models, with a broad literature addressing data processing and core data properties like size, coverage, and access (Muennighoff et al., 2025; Radford et al., 2022; Üstün et al., 2024; Longpre et al., 2025b; Soldaini et al., 2024). Alongside data, compute is another critical resource for developing foundation models. The scale of compute expenditure, the resultant demand for increased compute and energy infrastructure, and the geopolitics surrounding GPUs have all increased the salience and complexity of understanding compute allocation in foundation model development (Pilz et al., 2025). For this reason, the environmental impact of AI and its accurate measurement has become an increasingly divisive topic, especially as the computational costs of AI may substantively influence national-level resource allocation of energy, water, and electricity (Luccioni & da Costa, 2025; International Energy Agency, 2025). Beyond the extensive focus on data and compute, the upstream indicators also cover code (Initiative, 2024) and organizational structure as other determining factors in shaping the development of foundation models, as well as the cumulative cost of model development (Maslej et al., 2025; Casper et al., 2025).

**Model (Figure 4).** The 30 model indicators address properties, functions, and release of the foundation model and are organized into 6 subdomains:

- **Model Information (4 indicators).** Assesses transparency on properties that depends largely on the model itself, including basic model properties, inference time/compute, detailed model architecture, and model dependencies (e.g. teacher model used for distillation). *A new subdomain that covers a wider range of model properties in fewer indicators. It combines the previous Model Basics and Inference subdomains and adds two new indicators.*

- **Model Access (4 indicators).** Assesses transparency on how and to whom the developer provides model access (e.g. whether the developer provides open-weights access, whether the developer discloses the supported agent protocols). *Still focuses on Model Access, but refactors the previous edition to target more specific information on specialized model access and adds an indicator on agent protocols.*

- **Capabilities (4 indicators).** Assesses transparency regarding the capabilities that the developer specifically optimizes for during post-training and the evaluation of these capabilities. *Still focuses on Capabilities, but replaces two indicators from the previous edition that ask the developer to define/describe multiple model capabilities to a more specific indicator that asks the model to taxonomize the capabilities that were optimized for during post-training. Adds a new indicator on train-test overlap.*

- **Risks (5 indicators).** Assesses transparency regarding the risks the developer considers when developing the model and the evaluation of these risks. Also assesses transparency on external pre-deployment/risk evaluations. *Merges the previous Risks (including both intentional and unintentional harms), Limitations, and Trustworthiness subdomains into a single Risks subdomain with a single set of evaluations ("risks" in 2025 refers to "risks", "limitations", and "trustworthiness" in previous editions). Replaces two indicators asking the developer to define/describe multiple risks to a more specific indicator that asks the model to taxonomize the risks that were considered when developing the model. Adds two indicators on external risk evaluation.*

- **Model Mitigations (5 indicators).** Assesses transparency regarding the post-training mitigations implemented and the evaluation of these mitigations. *Replaces indicators from the previous edition on the description/demonstration of mitigations implemented into an indicator asking the developer to disclose a taxonomy of the post-training mitigations implemented and an indicator asking the*

**Upstream Indicators for the 2025 Foundation Model Transparency Index**

| Indicator | Definition |
|---|---|
| Data acquisition methods | What methods does the developer use to acquire data used to build the model? |
| Public datasets | What are the top-5 sources (by volume) of publicly available datasets acquired for building the model? |
| Crawling | If data collection involves web-crawling, what is the crawler name and opt-out protocol? |
| Usage data used in training | What are the top-5 sources (by volume) of usage data from the developer's products and services that are used for building the model? |
| Notice of usage data used in training | For the top-5 sources of usage data, how are users of these products and services made aware that this data is used for building the model? |
| Licensed data sources | What are the top-5 sources (by volume) of licensed data acquired for building the model? |
| Licensed data compensation | For each of the top-5 sources of licensed data, are details related to compensation disclosed? |
| New human-generated data sources | What are the top-5 sources (by volume) of new human-generated data for building the model? |
| Instructions for data generation | For each of the top-5 sources of human-generated data, what instructions does the developer provide for data generation? |
| Data laborer practices | For the top-5 sources of human-generated data, how are laborers compensated, where are they located, and what labor protections are in place? |
| Synthetic data sources | What are the top-5 sources (by volume) of synthetic data acquired for building the model? |
| Synthetic data purpose | For the top-5 sources of synthetically generated data, what is the primary purpose for data generation? |
| Data processing methods | What are the methods the developer uses to process acquired data to determine the data directly used in building the model? |
| Data processing purpose | For each data processing method, what is its primary purpose? |
| Data processing techniques | For each data processing method, how does the developer implement the method? |
| Data size | Is the size of the data used in building the model disclosed? |
| Data language composition | For all text data used in building the model, what is the composition of languages? |
| Data domain composition | For all the data used in building the model, what is the composition of domains covered in the data? |
| External data access | Does a third-party have direct access to the data used to build the model? |
| Data replicability | Is the data used to build the model described in enough detail to be externally replicable? |
| Compute usage for final training run | Is the amount of compute used in the model's final training run disclosed? |
| Compute usage including R&D | Is the amount of compute used to build the model, including experiments, disclosed? |
| Development duration for final training run | Is the amount of time required to build the model disclosed? |
| Compute hardware for final training run | For the primary hardware used to build the model, is the amount and type of hardware disclosed? |
| Compute provider | Is the compute provider disclosed? |
| Energy usage for final training run | Is the amount of energy expended in building the model disclosed? |
| Carbon emissions for final training run | Is the amount of carbon emitted in building the model disclosed? |
| Water usage for final training run | Is the amount of clean water used in building the model disclosed? |
| Internal compute allocation | How is compute allocated across the teams building and working to release the model? |
| Model stages | Are all stages in the model development process disclosed? |
| Model objectives | For all stages that are described, is there a clear description of the associated learning objectives or a clear characterization of the nature of this update to the model? |
| Code access | Does the developer release code that allows third-parties to train and run the model? |
| Organization chart | How are employees developing and deploying the model organized internally? |
| Model cost | What is the cost of building the model? |

Figure 3: **2025 Upstream Indicators.** The 34 upstream indicators in the 2025 FMTI. The full specification of every indicator can be found at `https://www.github.com/stanford-crfm/fmti`.

## Model Indicators for the 2025 Foundation Model Transparency Index

| Indicator | Definition |
|---|---|
| Basic model properties | Are all basic model properties disclosed? |
| Deeper model properties | Is a detailed description of the model architecture disclosed? |
| Model dependencies | Is the model(s) the model is derived from disclosed? |
| Benchmarked inference | Is the compute and time required for model inference disclosed for a clearly-specified task on clearly-specified hardware? |
| Researcher credits | Is a protocol for granting external entities API credits for the model disclosed? |
| Specialized access | Does the developer disclose if it provides specialized access to the model? |
| Open weights | Are the model's weights openly released? |
| Agent Protocols | Are the agent protocols supported for the model disclosed? |
| Capabilities taxonomy | Are the specific capabilities or tasks that were optimized for during post-training disclosed? |
| Capabilities evaluation | Does the developer evaluate the model's capabilities prior to its release and disclose them concurrent with release? |
| External reproducibility of capabilities evaluation | Are code and prompts that allow for an external reproduction of the evaluation of model capabilities disclosed? |
| Train-test overlap | Does the developer measure and disclose the overlap between the training set and the dataset used to evaluate model capabilities? |
| Risks taxonomy | Are the risks considered when developing the model disclosed? |
| Risks evaluation | Does the developer evaluate the model's risks prior to its release and disclose them concurrent with release? |
| External reproducibility of risks evaluation | Are code and prompts to allow for an external reproduction of the evaluation of model risks disclosed? |
| Pre-deployment risk evaluation | Are the external entities have evaluated the model pre-deployment disclosed? |
| External risk evaluation | Are the parties contracted to evaluated model risks disclosed? |
| Mitigations taxonomy | Are the post-training mitigations implemented when developing the model disclosed? |
| Mitigations taxonomy mapped to risk taxonomy | Does the developer disclose how the post-training mitigations map onto the taxonomy of risks? |
| Mitigations efficacy | Does the developer evaluate and disclose the impact of post-training mitigations? |
| External reproducibility of mitigations evaluation | Are code and prompts to allow for an external reproduction of the evaluation of post-training mitigations disclosed? |
| Model theft prevention measures | Does the developer disclose the security measures used to prevent unauthorized copying ("theft") or unauthorized public release of the model weights? |
| Release stages | Are the stages of the model's release disclosed? |
| Risk thresholds | Are risk thresholds disclosed? |
| Versioning protocol | Is there a disclosed protocol for versioning and deprecation of the model? |
| Change log | Is there a disclosed change log for the model? |
| Foundation model roadmap | Is a forward-looking roadmap for upcoming models, features, or products disclosed? |
| Top distribution channels | Are the top-5 distribution channels for the model disclosed? |
| Quantization | Is the quantization of the model served to customers in the top-5 distribution channels disclosed? |
| Terms of use | Are the terms of use of the model disclosed? |

Figure 4: **2025 Model Indicators.** The 30 model indicators in the 2025 FMTI. The full specification of every indicator can be found at `https://www.github.com/stanford-crfm/fmti`.

## Downstream Indicators for the 2025 Foundation Model Transparency Index

| Indicator | Definition |
|---|---|
| Distribution channels with usage data | What are the top-5 distribution channels for which the developer has usage data? |
| Amount of usage | For each of the top-5 distribution channels, how much usage is there? |
| Classification of usage data | Is a representative, anonymized dataset classifying queries into usage categories disclosed? |
| Data retention and deletion policy | Is a policy for data retention and deletion disclosed? |
| Geographic statistics | Across all forms of downstream use, are statistics of model usage across geographies disclosed? |
| Internal products and services | What are the top-5 internal products or services using the model? |
| External products and services | What are the top-5 external products or services using the model? |
| Users of internal products and services | How many monthly active users are there for each of the top-5 internal products or services using the model? |
| Consumer/enterprise usage | Across all distribution channels for which the developer has usage data, what portion of usage is consumer versus enterprise? |
| Enterprise users | Across all distribution channels for which the developer has usage data, what are the top-5 enterprises that use the model? |
| Government use | What are the 5 largest government contracts for use of the model? |
| Benefits Assessment | Is an assessment of the benefits of deploying the model disclosed? |
| AI bug bounty | Does the developer operate a public bug bounty or vulnerability reward program under which the model is in scope? |
| Responsible disclosure policy | Does the developer clearly define a process by which external parties can disclose model vulnerabilities or flaws? |
| Safe harbor | Does the developer disclose its policy for legal action against external evaluators conducting good-faith research? |
| Security incident reporting protocol | Are major security incidents involving the model disclosed? |
| Misuse incident reporting protocol | Are misuse incidents involving the model disclosed? |
| Post-deployment coordination with government | Does the developer coordinate evaluation with government bodies? |
| Feedback mechanisms | Does the developer disclose a way to submit user feedback? If so, is a summary of major categories of feedback disclosed? |
| Permitted, restricted, and prohibited model behaviors | Are model behaviors that are permitted, restricted, and prohibited disclosed? |
| Model response characteristics | Are desired model response characteristics disclosed? |
| System prompt | Is the default system prompt for at least one distribution channel disclosed? |
| Intermediate tokens | Are intermediate tokens used to generate model outputs available to end users? |
| Internal product and service mitigations | For internal products or services using the model, are downstream mitigations against adversarial attacks disclosed? |
| External developer mitigations | Does the developer provide built-in or recommended mitigations against adversarial attacks for downstream developers? |
| Enterprise mitigations | Does the developer disclose additional or specialized mitigations for enterprise users? |
| Detection of machine-generated content | Are mechanisms that are used for detecting content generated by this model disclosed? |
| Documentation for responsible use | Does the developer provide documentation for responsible use by downstream developers? |
| Permitted and prohibited users | Is a description of who can and cannot use the model on the top-5 distribution channels disclosed? |
| Permitted, restricted, and prohibited uses | Which uses are explicitly allowed, conditionally permitted, or strictly disallowed under the acceptable use policy for the top-5 distribution channels? |
| AUP enforcement process | What are the methods used by the developer to enforce the acceptable use policy? |
| AUP enforcement frequency | Are statistics on the developer's AUP enforcement disclosed? |
| Regional policy variations | Are differences in the developer's acceptable use or model behavior policy across geographic regions disclosed? |
| Oversight mechanism | Does the developer have an internal or external body that reviews core issues regarding the model prior to deployment? |
| Whistleblower protection | Does the developer disclose a whistleblower protection policy? |
| Government commitments | What commitments has the developer made to government bodies? |

Figure 5: **2025 Downstream Indicators.** The 36 downstream indicators in the 2025 FMTI. The full specification of every indicator can be found at `https://www.github.com/stanford-crfm/fmti`.

*developer to disclose how the taxonomy of mitigations maps onto the taxonomy of risks. Also adds an indicator on mitigations for model-theft.*

- **Release (8 indicators).** Assesses transparency on the model release process, including release decision-making (release stages, risk thresholds), documentation for model updates (versioning protocol, change-log), future model/product releases, and how the model is distributed (top distribution channels, quantization, terms-of-service). *A new subdomain that expands the Model domain to cover the model release process: combines the previous (Downstream) Model Updates subdomain, three indicators from the previous (Downstream) Distribution subdomain, and four new indicators.*

Like previous editions, this domain includes indicators that assess transparency on information about the model itself (Model Information): e.g. basic information expected by model documentation standards (Mitchell et al., 2019; Crisan et al., 2022; Bommasani et al., 2023b) like model size or architecture. However, foundation model architectures have also, over time, deviated from vanilla transformers/diffusion-models (Groeneveld et al., 2024), creating a need for transparency into properties of models beyond high-level descriptions of architecture or components. Next, Model Access addresses the level of access given by model developers across the spectrum of model release (Solaiman et al., 2019; Sastry, 2021; Shevlane, 2022; Liang, 2022; Solaiman, 2023). However, beyond the amount of access afforded, the *nature* of the access provided is also important: for example, subsidized access enables third-party research into model risks (Longpre et al., 2024) and agent protocols enable interoperability across agents (Ehtesham et al., 2025; Rao Surapaneni et al., 2025).

The Capabilities, Risks, and Model Mitigations subdomains, like previous editions, are based on how these factors influence the societal impact of foundation models (Tabassi, 2023; Weidinger et al., 2023). However, we've found that the way capabilities, risks, and mitigations are characterized often differs from developer to developer. As such, the newest edition includes indicators asking the developer to *taxonomize* the capabilities optimized for during post-training, the risks considered while developing the model, and the mitigations implemented while developing the model. Indicators on the actual evaluation of models based on these taxonomies build upon existing best-practices of rigorous and reproducible benchmarking (McCaslin et al., 2025; Gao et al., 2021; Lipton & Steinhardt, 2019; Kapoor et al., 2023; Uuk et al., 2024). In particular, indicators on evaluation reproducibility are motivated by the OSI definition of open-source AI that highlight the importance of publicly available artifacts like code (Initiative, 2024) and empirical investigations that point to the outsized impact that "minor" implementation details can have on results (Biderman et al., 2024). Researchers have also highlighted the need for and lack of public information like train-test overlap necessary for the actual *interpretation* of evaluation results (Zhang et al., 2025). Developers frequently use contracted expert evaluators to uncover model risks (Longpre et al., 2025a) but there remains many uncertainties like the lack of standardization (Ruth E. Appel, 2024) and the amount of independence (Santeri Koivula & Alejandro Tlaie, 2025). Finally, the indicator on model theft prevention measures is motivated by relevant cybersecurity guidance (NIST, 2024; Nevo et al., 2024).

The Release subdomain targets transparency into how developers release and distribute models. Model update indicators are motivated by work on the version control and updating of AI systems (Sathyavageesran et al., 2022; Hashesh, 2023; Chen et al., 2023). Future model/product release indicators are motivated by the broader market impacts of model or product releases (Vipra & Korinek, 2023; Cobbe et al., 2023). Indicators on distribution are motivated by work on AI supply chains (Bommasani et al., 2023b; Vipra & Korinek, 2023; Cen et al., 2023; Cobbe et al., 2023; Widder & Wong, 2023; Brown, 2023; Cen et al., 2025).

**Downstream (Figure 5).** The 36 downstream indicators in the 2025 edition targets model usage and how the developer addresses the impact of that usage and are organized into 7 subdomains.

- **Usage Data (5 indicators).** Assesses transparency on the amount of usage across distribution channels, usage categories, data retention policy, and geographic statistics. *A new subdomain containing 4 new indicators and a previous indicator on geographic statistics.*

- **Impact (7 indicators).** Assesses transparency regarding how the model is used, who uses the model. It covers the products and services that use the model, the amount of users across those

products and services, the nature of the model's users (consumer, enterprise, or government). It also asks the developer to disclose benefits assessments of the model's deployment. *Although the focus is the same at a high-level, the previous editions asks for more direct disclosures of impact (e.g. asking the developer to disclose the affected market sectors). The newest edition instead assesses impact by asking for disclosures that describe the composition of the applications and users. Only a single indicator (internal products and services) is retained from the previous editions.*

- **Post-deployment Monitoring (7 indicators).** Assesses transparency on how the developer monitors and mitigates risks after deployment. It assesses developers on policies that enable the external evaluation of models (AI bug bounty, responsible disclosure policy, safe harbor), reporting protocols for security/misuse incidents, coordination with governments, and feedback mechanisms. *A completely new subdomain.*

- **Model Behavior Policy (4 indicators).** Assesses transparency on acceptable/unacceptable model behavior. It covers restricted behaviors, response characteristics, the default system prompt, and intermediate (e.g. chain-of-thought) tokens. *Although the high-level focus is the similar to previous editions, the focus of the newest edition targets more indirect information about the MBP like the model response characteristics and system prompt. Only a single indicator is retained from the previous editions (permitted, restricted, and prohibited model behaviors).*

- **Downstream Mitigations (5 indicators).** Assesses transparency on risk mitigations to be implemented downstream: covers built-in/recommended mitigations, mechanisms for detecting machine generation, specialized mitigations for enterprises, and the documentation of responsible use for developers. *A new subdomain with three new indicators on mitigations implemented downstream. This new subdomain also includes previous indicators on Documentation for Deployers and the detection of machine-generated content.*

- **Acceptable Use Policy (5 indicators).** Assesses transparency on the AUP and its enforcement: including prohibited uses across distribution channels, the process and statistics describing enforcement, and finally variations across geographies. *Has the same focus as the previous Usage Policy subdomain, but replaces indicators on justification/appeals to enforcement frequency and policy variations (both AUP and MBP).*

- **Accountability (3 indicators).** Assesses transparency on organizational accountability mechanisms, including oversight mechanisms to review issues prior to model deployment, whistleblower protection policies, commitments made to government bodies. *A completely new subdomain.*

Usage data indicators assess transparency into the amount and nature of model usage. These indicators were motivated by literature on AI supply chains (Bommasani et al., 2023b; Vipra & Korinek, 2023; Cen et al., 2023; Cobbe et al., 2023; Widder & Wong, 2023; Brown, 2023), existing regulation on user data retention more broadly (EU, 2016), and existing methods employed by developers to disclose anonymized categories of model usage (Tamkin et al., 2024; Appel et al., 2025). The Impact subdomain assessing transparency into model usage based on dependent products/services and consumer/enterprise/government users, motivated by risks arising from AI usage in high-risk domains (Solaiman et al., 2019; Vipra & Korinek, 2023; Cobbe et al., 2023; Brown, 2023; Shevlane, 2022). Indicators on post-deployment monitoring were motivated by recommendations on safe-harbors for external evaluation (Longpre et al., 2024; 2025a), existing reporting protocols for cybersecurity incidents[5], and the growing role of governmental organizations in conducting pre-deployment evaluations of models (NIST). In addition, a growing body of work explores how adverse event reporting and incident reporting can be operationalized specifically for AI (Gailmard et al., 2025; Wei & Heim, 2025; Bommasani et al., 2025). Indicators on Model Behavior Policy were motivated by past work on AI behavior and risk mitigations (Reuter & Schulze, 2023; Qi et al., 2023). We update this subdomain to reflect technical developments that present new transparency considerations e.g. with reasoning models & intermediate tokens (Chen et al., 2025). Acceptable use policy indicators were motivated by transparency reporting requirements and disclosures for policy enforcement by social media platforms (Commission,

---

[5]e.g. https://aws.amazon.com/security/security-bulletins

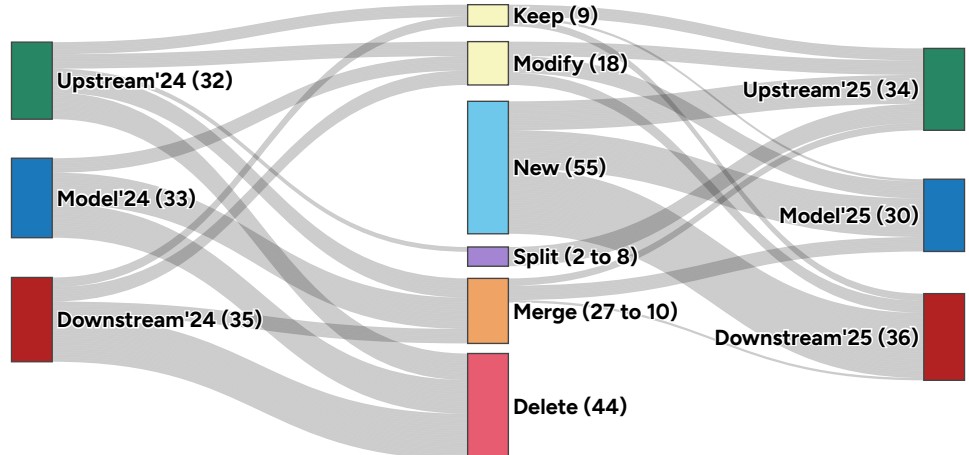

Figure 6: **Changes in Indicators Across Domain.** We make significant changes across all three domains: 55 indicators in the 2025 Index are completely new and only 9 indicators are kept the same across the two editions (*Kept*). The remaining 36 indicators were derived from indicators in 2024 Index either by making significant modifications to the definition/notes (*Modify*), splitting an indicator (*Split*), or merging multiple indicators into one (*Merge*).

2022).[6] Indicators on Accountability were motivated by the myriad voluntary commitments made by model developers (Wang et al., 2025a), whistleblower protections present in regulations (Act, 2025; Bommasani et al., 2025), and recommendations of disclosures into developers' internal governance mechanisms (Kolt et al., 2024).

## 3.3   Summary of Changes

The past two editions of the FMTI provided several forms of feedback: we better understand what companies currently disclose, how their employees conceptualize transparency including potential tradeoffs with other organizational priorities, and why different stakeholders benefit from increased transparency. In parallel, much has also changed about foundation models and the AI ecosystem (e.g. the financial costs involved in model training, the technical paradigm with test-time scaling). We summarize the changes we made to the indicators based on a few high-level design choices with the mapping between old and new indicators shown in Figure 6.

**Raising the bar**   For several indicators, past editions involved awarding points to companies for disclosures that were not useful. For example, we do not believe it is useful to describe a language model's capabilities so broadly as to say the model is capable of "text generation". Therefore, we made some indicators stricter by more clearly defining the specific piece of information that should be disclosed. For the aforementioned example, the old indicator awarded a point for "any clear, but potentially incomplete, description of multiple capabilities" whereas the new indicator awards a point for "a list of capabilities that were *specifically optimized for during post-training*".

**Codifying reproducibility**   For several indicators, past editions had an unclear standard for what sufficed as external reproducibility. Therefore, we made the decision to codify our standard for reproducibility by requiring open-sourced code and prompts to claim that a particular evaluation is externally reproducible. We also add new indicators that ask the developer to release code that allows third-parties to reproduce model development (e.g. processing training data).

---

[6]See https://transparency.meta.com/reports/community-standards-enforcement/ for an example.

**Focusing on the head of the distribution**   For several topics as a whole, past editions reveal that companies disclose nothing on these topics. Based in part on dialogue with companies, we focused our attention to the head of the distribution. For example, instead of seeking information on every data source, we instead restrict attention to the top-5 data sources. While this introduces new complexity (i.e. how to rank to determine the top 5), we were lenient as our focus is the most important contributions rather than the exact methodology for identifying them.

**Identifying developer-level information deficits**   For several indicators, past editions sought information that would entail the developer acquiring this information from another party. For example, if a developer trains their model using an external cloud service, then they would need information from the cloud service to report energy or environmental costs. Based in part on dialogue with companies, we now award points if the developer discloses they do not have this information and names the party that they depend upon to provide this information. In particular, if a contractual agreement prohibits the disclosure of this information, then we generally award the indicator subject to clarity about the contractual obligation.

**Modernizing the indicators**   For several topics, the 2023 indicators simply do not cover new developments that are critical to our understanding of foundation models in 2025. Across all domains, we introduced new indicators to cover these topics of increased salience. For example, in Upstream, we ask about synthetic data generation; in Model, we ask about supported agent-protocols; and in Downstream, we ask about government commitments made by model developers.

**Targeting organization practices**   In the AI ecosystem, the Foundation Model Transparency Index plays a distinctive role by focusing on AI companies as important organizations alongside the important technologies they build. To build on this strength, we introduced new indicators to increase our focus on the companies themselves. In the upstream domain, we add an indicator about the developer's organization structure, namely the internal organization chart of employees involved in foundation model development. In the model domain, we add an indicator about the organization's future roadmap as it plans for upcoming models. In the downstream domain, we add an indicator about the organization's oversight, including through review provided by external overseers.

## 4   Methods

In this section we describe how we selected developers, gathered information about their practices, and scored this information against the indicators to determine each company's score on the 2025 FMTI.

### 4.1   Developer selection

For this edition of the Index, we engaged more companies by contacting 23 foundation model developers, compared to 19 in 2024. Consistent with the principles established in the 2024 edition, we maintained our focus on companies developing prominent foundation models, considering diversity in company type, model modality, and geographic representation. Consequently, the 2025 FMTI includes Chinese AI companies for the first time. These companies have developed much more capable models in the past year, thereby playing a greater role in the AI ecosystem. By diversifying the companies, the 2025 FMTI provides a more comprehensive view of global transparency practices.

Of the 23 companies contacted, 7 agreed to submit transparency reports—a decrease from the 14 participants in FMTI 2024. All 7 companies that submitted reports were returning participants from the previous year: AI21 Labs, Amazon, Google, IBM, Meta, OpenAI, and Writer. Each company selected its flagship foundation model as of May 2025 in consultation with our team, following the same guidance as in 2024: selection based on a combination of resource expenditure, model capabilities, and societal impact. As before, we exclude developers that are not companies (e.g. non-profit or academic developers) as their practices naturally differ from profit-seeking companies. We also exclude developers that have not released a foundation model in 2025 in order to focus on the practices of active developers.

To maintain comprehensive coverage despite reduced voluntary participation, we evaluated 6 companies that did not submit reports. These companies were selected based on their significance to the ecosystem: xAI (Grok 3) and Anthropic (Claude 4) for their position at the technical frontier; DeepSeek (DeepSeek-1) and Alibaba (Qwen 3) as leading Chinese developers; Midjourney (Midjourney V7) for image model representation; and Mistral (Medium 3) for European representation.

This left 10 developers that we contacted that did not participate: Zhipu AI and Stability AI did not respond to our outreach, while 01.AI, Adobe, Apple, Baidu, Cohere, Microsoft, and Nvidia all declined to participate. Companies offered a variety of justifications for declining to participate, including one company stating it does not view itself as an AI developer, another stating that this type of transparency is not appropriate for enterprise-focused companies, and another stating that their team was too busy to prepare a report.

In total, we evaluated 13 foundation model developers for the 2025 Foundation Model Transparency Index. Table 1 presents all developers, their flagship models, and some of their key characteristics. Throughout our analysis, we treat all 13 companies uniformly while acknowledging the methodological differences in data collection where relevant.

| Name | Flagship Model | Release | Input | Output | HQ |
|---|---|---|---|---|---|
| AI21 Labs[†] | Jamba-1.6 | Open weights | T | T | Israel |
| Alibaba | Qwen 3 | Open weights | T | T | China |
| Amazon[†] | Nova Premier | API | T, I, V | T | USA |
| Anthropic | Claude 4 | API | T, I | T | USA |
| DeepSeek | DeepSeek-R1 | Open weights | T | T | China |
| Google[†] | Gemini 2.5 | API | T, I, A, V | T | USA |
| IBM[†] | Granite 3.3 | Open weights | T | T | USA |
| Meta[†] | Llama 4 | Open weights | T, I | T | USA |
| Midjourney | Midjourney V7 | API | T, I | I | USA |
| Mistral | Mistral Medium 3 | API | T, I | T | France |
| OpenAI[†] | o3 | API | T, I | T | USA |
| Writer[†] | Palmyra X5 | API | T, I | T | USA |
| xAI | Grok 3 | API | T | T | USA |

Table 1: **Selected Foundation Model Developers.** Information on the 13 selected foundation model developers: the developer name, its flagship model, the release strategy for the model, the model's input and output modalities, and the developer's headquarters. T, I, A, and V abbreviate text, image, audio, and video modalities, respectively. † indicates the developer submitted a transparency report for FMTI 2025.

## 4.2 Information gathering

We employed two distinct information gathering approaches for FMTI 2025, depending on company participation.

**Developer-submitted reports.** For the 7 companies that agreed to participate in the Foundation Model Transparency Index, we followed the same process established as in the 2024 edition. Companies received detailed instructions to prepare transparency reports that directly addressed each of the 100 indicators. Developers were given 4 weeks to compile their reports, with extensions of up to 2 weeks granted upon request. This timeline allowed companies to gather information across different internal teams and to vet release of specific information. The report format remained unchanged from 2024, though the specific indicators changed (as discussed in the previous section).

**Hybrid information gathering approach.** We implemented a hybrid approach combining human search with automated information gathering for the 6 companies that did not submit reports (Alibaba, Anthropic, DeepSeek, Midjourney, Mistral, and xAI).[7] We prepared transparency reports following the methodology

---

[7]As some reports are prepared by the FMTI-team but others are prepared by the developers themselves, there is a bias towards developer-prepared reports since model developers have access to information that's not public or difficult to find. This

established in FMTI 2023, systematically searching for publicly available information published by the companies. Sources included company websites, GitHub repositories, model cards, arXiv papers, and reports submitted to regulatory bodies. Each indicator for each company was scored by two members of the FMTI team. While some documentation for DeepSeek and Alibaba was available in English, this was not always the case. For non-English materials, a Chinese-speaking author of this paper annotated the relevant indicators.

In parallel, we deployed an automated evaluation agent (Appendix A) to independently assess these same 6 companies using identical source constraints. This served two purposes: first, to benchmark how well AI agents can perform complex workflows in gathering information and assessing it against a transparency report prepared by the FMTI team; and second, to improve our information gathering by identifying relevant content that the FMTI team may have overlooked. The agent was built around Anthropic's Claude 4 API and uses Anthropic's native tool-calling capabilities.

After both information gathering steps were completed, we compared findings to identify cases where the agent discovered information missed by the team, where the team found information the agent missed, or where both identified equivalent information. Since our goal was comprehensive information gathering rather than scoring at this stage, we incorporated all relevant findings from both sources into the final reports. The agent evaluation occurred concurrently with the companies' report preparation. We provide more details on the agent scaffold we developed in Appendix A.

### 4.3 Agent performance compared to FMTI team

We compared the performance of the agent with the human expert in the FMTI team for uncovering relevant information for transparency reports. In particular, annotators from the FMTI team looked at the information found by the agent, and marked it as finding the same/similar information as found by the human team, or rated either the human team/the agent as having found more relevant information for the indicator.

We found that the agent demonstrated both complementary strengths and limitations compared to human evaluators across the 100 transparency indicators. The agent missed information that human evaluators found for 4-16 indicators per company (mean = 8), while simultaneously identifying additional information overlooked by humans for 8-17 indicators per company (mean = 13). We report how many additional indicators the FMTI team found compared to the agent for each company in Appendix A.

Strikingly, the agent discovered more relevant information that the human team missed compared to the human team finding relevant information that the agent missed for five of the six companies evaluated, with only Alibaba showing equal performance (16 indicators in both directions). This suggests that automated agents can meaningfully aid systematic search tasks. This pattern varied notably across the three transparency domains. For upstream indicators, the agent primarily missed information that the human team found (0-8 indicators) while finding minimal additional content compared to the human team (0-1 indicators). We think this is because finding transparency information for upstream indicators typically requires reading and understand a few documents in depth (such as model cards) rather than finding shallow information scattered across documents. Conversely, the agent excelled at discovering downstream information, finding additional content for 3-14 indicators that humans missed, particularly for Alibaba (14) and Mistral (12). Model-level indicators showed more balanced performance, with the agent both missing and finding comparable amounts of information across companies. These results show the potential for automated agents to dramatically reduce evaluation costs in domains where manual information discovery is prohibitively expensive; in this case, the human expert team took weeks of person effort with automated evaluation while achieving comparable information coverage.

Finally, note that expert effort was still required to discern which of the agent's outputs were correct. While the FMTI team's transparency reports primarily suffered false negatives (failing to find relevant information), the agent exhibited both false positives (retrieving irrelevant information that superficially appeared relevant) and false negatives. In many cases, determining whether the agent's findings were actually relevant to the specific indicator required deep domain expertise to properly assess. This pattern suggests that automated

---

bias is intentional: from the 2024 FMTI, we know that engaging with companies directly allows for the public disclosure of more information (i.e., increased transparency). We choose to prioritize this over consistent scoring across companies. We explore this further in §5.2

agents are better suited for augmenting human transparency evaluation teams as human judgment remains essential for verifying the relevance and accuracy of discovered information.

### 4.4 Scoring

Following the information gathering phase, we scored each company on all 100 indicators and engaged companies to finalize the scores and reports.

**Scoring process.** For each of the 1,300 (indicator, developer) pairs, two FMTI team members assigned scores independently given the available information. The agreement rate was 89.4% (138 disagreements), which is the highest agreement rate across any edition of the Index (85.3% in 2024; 85.2% in 2023). In cases of disagreement between scorers, the researchers discussed the discrepancy and reached consensus based on the established scoring criteria to determine the score along with its justification. Consequently, for each company we prepared an initial scored report that consolidates the disclosure, our initial score, and our initial justification for all 100 indicators.

**Company response.** Once we prepared the initial reports for all 13 companies, we sent each company this initial report. In particular, even if a company did not provide a report of their disclosures, we still provided them the initial scored report. Companies were given one week to initially respond to scores, leading to many companies engaging in extensive email exchanges and scheduling virtual meetings to discuss the scores. On average, companies clarified, corrected, or otherwise updated disclosures for 18.9 indicators (median = 21). Following the finalization of the 2025 FMTI scores, this yielded an average increase of 9.71 (median = 9) with AI21 Labs seeing the largest score change during the response phase with an increase of 19 points.[8]

Following this period, we finalized the 2025 FMTI scores and reports. The published transparency reports were sent to each company prior to release for final validation and the public release materials were sent to companies a few days prior to release as a professional courtesy. Overall, similar to the 2024 FMTI, the process of repeated engagement fostered extended dialogue and increased trust.

### 4.5 Timeline

1. **Indicator design (January – March 2025).** The FMTI team designed the 2025 FMTI indicators.

2. **Indicator review (March – April 2025).** External reviewers and the FMTI advisory board reviewed the 2025 FMTI indicators.

3. **Company solicitation (late April 2025).** The FMTI team contacted 23 companies to understand if they would participate in the 2025 FMTI by submitting reports.

4. **Report preparation (May – June 2025).** 7 companies prepared transparency reports and the FMTI team prepared 6 additional reports.

5. **Initial scoring (July – August 2025).** The FMTI team scored the 13 reports.

6. **Company response (August – September 2025).** The FMTI team sent the companies their scores and engaged companies to understand their responses.

7. **Finalized scoring (September – December 2025).** The FMTI team finalized the 2025 FMTI scores, sent the companies the finalized reports for validation, wrote the paper, and released the 2025 FMTI.

## 5 Results

**2025 FMTI high-level trends.** The average score for the 2025 FMTI is a 40.69, which reflects the generally poor state of transparency across the AI ecosystem. Figure 7 depicts the overall scores for each

---

[8]While the vast majority of the score changes during the response phase were directly responsive to company engagement, a small number of changes were brought about by standardizing the scoring criterion across companies to ensure consistency.

**Foundation Model Transparency Index Scores by Domain, 2025**
Source: 2025 Foundation Model Transparency Index

Figure 7: **Scores by Domain.** The 2025 FMTI scores for each company disaggregated by domain.

company along with how those scores are computed as the sum of the scores for each of the three domains. IBM is the clear standout with the highest score in the history of the Index at 95 out of 100. Beyond IBM, Writer and AI21 Labs also achieve high scores that place them more than one standard deviation above the mean. Behind these companies, the majority of companies have scores of 26–46, which are within a standard deviation of the mean. The lowest-scoring companies are Mistral, Midjourney, and xAI, with Midjourney and xAI's score of 14 being the second-lowest in FMTI history.[9] The wide range of 81 reflects significant variation in current transparency practices.

| Domain | Min | Mean | Median | Max | $s$ |
|---|---|---|---|---|---|
| Upstream | 0 | 9.2 | 6 | 34 | 9.9 |
| Model | 5 | 13.8 | 13 | 28 | 6.2 |
| Downstream | 7 | 17.8 | 18 | 33 | 8.8 |

Table 2: **Domain-level Statistics.** Aggregate statistics on the domain-level 2025 FMTI scores.

The domain-level scores (Table 2) provide insight into sources of variation in company practices that explain the measured difference in the overall scores on the 2025 Index. In general, companies score lowest on the upstream domain and highest on the downstream domain. However, both of these domains demonstrate significant variation. Stratifying the domain-level scores across the three clusters of high-scoring, average-scoring, and low-scoring companies (see Figure 7) reveals that (i) low-scoring companies earn many of their points from the downstream domain and (ii) average-scoring companies are score similarly on the model domain while quite varied for the other two domains. Below, we further analyze the trends within each of the three domains.

The upstream domain, which covers topics like training data and training compute, is the least transparent yet most heterogeneous. Two companies (Midjourney, Mistral) do not score any indicators in the entire domain and three others (OpenAI, xAI, Anthropic) score at most 3 points out of the available 34 in this domain. Only one indicator in the entire upstream domain is satisfied by a significant majority of companies:

---

[9]In 2023, Amazon received a score of 12 on the inaugural FMTI, compared to their 2025 score of 39 that puts them in the top half of all companies assessed this year.

9 of the 13 companies disclose their compute provider, which is especially easy for most companies to disclose. The limited transparency of many companies on this domain prompts consideration of a general explanation. While multiple societal objectives motivate upstream transparency, current societal conditions create systemic incentives for opacity. For example, ongoing copyright litigation against several foundation model developers disincentivizes transparency about training data. Large-scale investment in compute and infrastructure may lead companies to see this as a core competitive differentiator, leading to opacity as a deliberate strategic decision.

While systemic incentives can explain low upstream transparency, they fail to explain significant variation. IBM receives points for every indicator in this domain, while Midjourney and Mistral receive none of them. Even excluding these three companies, Figure 8 shows that for every large subdomain of the upstream domain, at least one company scores 0% while another scores 90%. This range suggests that organizational choice is a core factor in upstream transparency. Current conditions in the ecosystem neither compel companies to demonstrate non-zero transparency nor prevent them from achieving very high levels of transparency.

The model domain, which covers topics like model capabilities and risks, is the smallest and least heterogeneous of the three domains. In particular, every company receives the two indicators for publishing a change log and terms of use. And no company discloses train-test overlap (see Zhang et al., 2025) nor do they enable external reproducibility of model-level mitigations. Yet the domain still contains significant differences across companies. This variation is seen at both company and the indicator level. Three companies (Midjourney, Mistral, xAI) score less than 10 of the indicators while two companies score more than 20 of the indicators (AI21 Labs, IBM) in the 30-indicator model domain. 8 of the indicators are satisfied by at most 3 companies, 7 of the indicators are satisfied by at least 9 companies, and the remaining half of the indicators are satisfied by between 4 to 8 of the 13 companies.

The downstream domain, which covers topics like post-deployment monitoring and acceptable usage policies, is the largest and most transparent of the three domains. Of the 36 indicators in the downstream domain, 11 are satisfied by at least 9 companies. Every company sufficiently discloses both the permitted and prohibited uses, as well as permitted, restricted, and prohibited uses, of their model in their acceptable usage policy. While transparency is higher on average in this domain than the others, 6 companies (Alibaba, DeepSeek, Meta, Midjourney, Mistral, xAI) scores at most a 12 (i.e. at most a third of the upstream indicators). Certain subdomains are opaque on average (see Figure 8), namely impact (29%) and usage data (25%) with two companies scoring well (IBM at 80% and 100%; Writer at 86% and 100%). The results indicate that transparency is high when disclosures cover basic user-facing and legally-obligated subjects, but low when tied to downstream usage and the resulting post-deployment consequences.

As with upstream indicators, systemic incentives explain some, but not most, downstream transparency. High disclosure rates cluster around release-stage obligations (such as acceptable use policies, system-behavior guidelines, and responsible-use documentation) which align with regulatory expectations and standard product-deployment practices. Outside of the acceptable use policy subdomain, scores range from 0% to 100% across nearly all subdomains. Notably, two companies (Alibaba and xAI) score 0% on three out of six subdomains; and DeepSeek scores 0% on all but two subdomains (model behavior policy and acceptable use policy). This variation indicates that organizational choice, rather than structural constraints, drives downstream transparency: firms face only minimal pressure to disclose beyond baseline release norms, yet some (IBM, Writer, AI21 Labs) opt for comprehensive transparency while others (DeepSeek, Alibaba, xAI) provide almost none.

## 5.1 Subdomain-level results

**Developers are opaque on training data.** Data properties is the lowest scoring subdomain, with companies scoring just 15% on average. Eight companies score none of the 5 indicators in this subdomain, failing to disclose the data size, language composition, or domain composition, or to provide external access to the data or instructions for replicating the data. Of these indicators, data size is where companies are the most transparent, with four open-weight model developers (Alibaba, DeepSeek, IBM, Meta) and Writer disclosing the size of their data used to build the model.

**Foundation Model Transparency Index Scores by Major Dimensions of Transparency, 2025**
Source: 2025 Foundation Model Transparency Index

| Major Dimensions of Transparency | Jamba 1.6 | Qwen 3 | Nova Premier | Claude 4 | DeepSeek-R1 | Gemini 2.5 | Granite 3.3 | Llama 4 | Midjourney V7 | Medium 3 | o3 | Palmyra X5 | Grok 3 | Average |
|---|---|---|---|---|---|---|---|---|---|---|---|---|---|---|
| Data Acquisition | 92% | 17% | 17% | 25% | 17% | 33% | 100% | 33% | 0% | 0% | 8% | 58% | 0% | 31% |
| Data Properties | 0% | 20% | 0% | 0% | 20% | 0% | 100% | 20% | 0% | 0% | 0% | 40% | 0% | 15% |
| Compute | 22% | 11% | 11% | 0% | 44% | 11% | 100% | 22% | 0% | 0% | 0% | 100% | 11% | 26% |
| Model Information | 75% | 75% | 0% | 25% | 75% | 0% | 100% | 75% | 0% | 0% | 0% | 75% | 0% | 38% |
| Model Access | 50% | 50% | 50% | 50% | 50% | 50% | 100% | 50% | 0% | 25% | 0% | 50% | 0% | 40% |
| Capabilities | 75% | 50% | 50% | 25% | 50% | 25% | 75% | 50% | 0% | 25% | 25% | 50% | 25% | 40% |
| Risks | 60% | 0% | 40% | 60% | 20% | 20% | 100% | 20% | 0% | 0% | 60% | 40% | 0% | 32% |
| Model Mitigations | 60% | 0% | 60% | 80% | 20% | 40% | 80% | 0% | 0% | 20% | 80% | 40% | 0% | 37% |
| Release | 88% | 63% | 75% | 75% | 63% | 88% | 100% | 50% | 63% | 38% | 63% | 88% | 50% | 69% |
| Usage Data | 20% | 0% | 20% | 60% | 0% | 0% | 80% | 0% | 20% | 0% | 20% | 100% | 0% | 25% |
| Impact | 71% | 0% | 0% | 29% | 0% | 29% | 86% | 14% | 29% | 14% | 14% | 86% | 0% | 29% |
| Post-deployment Monitoring | 71% | 0% | 57% | 57% | 0% | 43% | 100% | 29% | 0% | 43% | 71% | 86% | 0% | 43% |
| Model Behavior Policy | 100% | 50% | 75% | 100% | 75% | 75% | 100% | 75% | 25% | 0% | 75% | 50% | 75% | 67% |
| Acceptable Use Policy | 80% | 60% | 80% | 100% | 60% | 80% | 80% | 40% | 60% | 60% | 60% | 80% | 60% | 69% |
| Downstream Mitigations | 100% | 40% | 100% | 100% | 0% | 100% | 100% | 80% | 40% | 80% | 100% | 100% | 40% | 75% |
| **Average** | **64%** | **29%** | **42%** | **52%** | **33%** | **40%** | **93%** | **37%** | **16%** | **20%** | **38%** | **69%** | **17%** | |

Figure 8: **Scores by Major Dimension of Transparency.** The average score for each company disaggregated by major dimension of transparency. Major dimensions refer to selected large subdomains within the 2025 FMTI.

Few companies transparently describe how they acquire data used to build their system: the average score for the subdomain is 31% with only AI21 Labs, IBM, and Writer scoring above 50% and four companies (Midjourney, Mistral, OpenAI, xAI) scoring 10% or below. For example, only IBM discloses the licensed data sources it uses to build its foundation model despite widespread reporting on licensing agreements to incorporate such data into pretraining corpora (Miller & Bass, 2024; Wiggers, 2023; Tong et al., 2024).

As in the 2023 and 2024 iterations of the Index, the extreme opacity on data remains the area where transparency is most lacking. Access to data and information about data is essential for enabling reproducible research, promoting downstream innovation, and accurately contextualizing model evaluations (Longpre et al., 2023a).

**Training compute continues to lack transparency, especially for the most compute-intensive models.** Compute is another upstream subdomain where companies disclose especially little information. IBM and Writer, which both score 100%, are the only two companies to score above 50% in the domain; DeepSeek places third with 44% as it discloses the development duration for the final training run, the compute hardware for the final training run, the compute provider, and the internal compute allocation. 9 of the 14 companies disclosed their compute provider, the sole compute indicator that more than half of companies scored. IBM and Writer are the only companies that disclose compute usage for the final training run, compute usage including R&D, and the energy and water usage for the final training run. Critically, the models that are conjectured to consume the most compute based on estimates from Epoch[10] are the same models where developers are most opaque.

**There is little to no information on the environmental impact of AI.** Companies are highly opaque about the environmental impact of building foundation models. 10 companies disclose none of the key information related to environmental impact: AI21 Labs, Alibaba, Amazon, Anthropic, DeepSeek, Google, Midjourney, Mistral, OpenAI, and xAI. Of the Big Tech companies based in the United States, which are among the largest companies by market capitalization and have billions of users, only Meta discloses any environmental-impact related information, stating in its model card that "Estimated total location-based greenhouse gas emissions were 1,999 tons CO2eq for training. Since 2020, Meta has maintained net zero greenhouse gas emissions in its global operations and matched 100% of its electricity use with clean

---

[10]https://epoch.ai/data/ai-models

and renewable energy; therefore, the total market-based greenhouse gas emissions for training were 0 tons CO2eq."[11]

The environmental impact of AI systems has become a major issue as datacenter buildouts continue at an unprecedented rate, which has contributed to energy price hikes (Saul et al., 2025). In response, some companies disclose the environmental impact of an average input-output interaction (Luccioni et al., 2024). However, this is incomplete: it does not provide information on the impact of model training and is insufficient for understanding the *cumulative costs* of model use, since most companies do not release information about the total amount of usage.

**Disclosures of model stages and objectives have declined**   The remaining two upstream subdomains, methods and other resources, show mixed results. Roughly half of the companies disclose the model stages and objectives, a decrease from previous iterations of the index. This includes both open and closed model developers such as Anthropic, Meta, and OpenAI. IBM is the only company to release code that allows third parties to train and run the model, demonstrating the limits—even among open-weight developers—in operationalizing the benefits from fully open-source software (Initiative, 2024). Finally, only AI21 Labs, IBM, and Writer disclose their organizational chart for teams involved with their flagship model's development and deployment.

**Companies rarely disclose the cost of building their models.**   The cumulative resource expenditure across the upstream domain can be distilled to a single number: how much money does a company spend to build its flagship model? IBM and Writer are the only companies to disclose this amount, which is essential for understanding whether the costs of foundation model development favorably amortize through repeated use along the AI supply chain, and in turn computing the return on investment for high-stakes model development. IBM discloses that "We estimate our total Granite 3 8B model cost to be \$10M, where \$4M was spent on data processing, \$2M is spent on hyperparameter searches, and \$2M on the final pre-training run, and \$2M on post-training and post-training experiments." Writer discloses that the their models costs "Around 7 – 8million with 6M on compute and around 1.5M around R&D". These costs reveal very different distributions: IBM's reporting suggests a 40-40-20 split across data, R&D compute and final training run compute while Writer indicates a 0-20-80 split over the same three categories. We emphasize the foundational importance of this number on the market's ability to reason about current AI costs and their trajectory, as well as that estimating this quantity externally is very difficult, meaning foundation model developers are largely unique in their ability to advance community-wide understanding of this topic.

**Many companies do not disclose basic information about the model itself.**   In the model domain, we first consider essential and basic information about the model itself. Amazon, Google, Midjourney, Mistral, OpenAI and xAI do not score any indicators in the model information subdomain, such as the basic model information indicator (which includes input modality, output modality, model size, model components, and model architecture). 5 companies disclose information related to deeper model properties (i.e. a detailed description of the model architecture), 7 disclose model dependencies (the models the foundation model depends on and how it is derived from those models), and just 2 (IBM and Writer) disclose benchmarked inference statistics.

This opacity demonstrates the inadequacy of common AI documentation artifacts. While many of these companies publish a model card or technical report, these documents generally do not contain essential information about the foundation model itself. Developers' documentation for deployers may include the input and output modalities, but with the rest of the model as a blackbox.

**Information about access to foundation models is limited.**   Foundation models are widely used, but companies do not disclose only limited information about how they provide access to external entities. 5 companies disclose that they provide API credits to external researchers (Anthropic, Google, IBM, Meta, and Writer). Companies regularly provide access to their systems to customers and trusted third parties, but only Amazon and IBM disclose if they provides specialized access and statistics on the number of users granted

---

[11]1,999 tons of CO2eq is equates to the annual electricity use of 268 homes in the United States (US EPA, 2024).

access across academia, industry, non-profits, and governments, to one significant figure. 5 companies openly release the weights of their flagship foundation models, providing deeper access to the public as a whole.

9 companies disclose the agent protocols they support, while Meta, Midjourney xAI, and OpenAI do not. For instance, Alibaba's GitHub repository for Qwen Agent—which is openly licensed under Apache 2.0—includes MCP Cookbooks. This is a much greater degree of transparency than some other companies, who simply state the protocols (e.g. MCP, A2A) they support.

**Companies disclose capabilities evaluations, but the evaluations are not reproducible or transparent.** Capabilities is one of the highest scoring subdomains, with companies scoring 40% on average. However, performance is uneven, with 12 of 13 companies disclosing capabilities evaluations of their models (all but Midjourney), 2 disclosing code and prompts that allow for external parties to reproduce capabilities evaluations (AI21 Labs and IBM), and none disclosing the overlap between the training set and the test set.

In this edition of the Index, we prioritize information regarding capabilities that were optimized for during post-training. 7 companies disclose a taxonomy of these capabilities, while others tend to list out areas where their foundation models are capable but do not clarify whether these capabilities were intentionally built-in by the developer.

Companies have an incentive to disclose evaluation results as it helps market their AI products as more capable than their competitors'. Investors, media, and the public rely on public evaluation results to make decisions about which foundation models are most useful, but evaluation leaderboards are often flawed (Singh et al., 2025) and the results on these evaluations lack transparency. Without additional information, it is unclear how much trust we should put in them (Zhang et al., 2025).

**Companies disclose less information about risks than capabilities.** As in the previous editions of the Index, companies share less information about the risks of their foundation models than their capabilities, scoring 32% on the subdomain. Just 4 of 13 companies evaluate risks prior to release and report results upon release (AI21 Labs, Anthropic, IBM, and OpenAI) and only IBM releases an externally reproducible risk evaluation. Whereas companies often publish capabilities evaluation results to improve the commercial success of their models, reporting risks could increase liability and the likelihood of lawsuits from consumers and enterprises harmed by those risks.

Companies also often do not disclose who they collaborate with to evaluate risk. 5 companies (AI21 Labs, Amazon, Anthropic, IBM, and OpenAI) disclose the external parties they contract with to evaluate risk, but major companies like Google, Meta, xAI, and Alibaba do not. On the other hand, 9 companies do disclose their risk taxonomies (i.e. the risks they considered when developing the model). As a result, external parties can conduct assessments of risks aligned with those taxonomies, though these risk taxonomies may be narrower than those considered by civil society, researchers, or policymakers.

**Companies often disclose the mitigations they use, but not how effective they are.** As with capabilities and risks, companies disclose high level information about mitigations, such as taxonomies, but do not share more granular information, such as evaluations of efficacy or externally reproducible evaluations. Just 3 companies, Anthropic, IBM, and OpenAI, disclose evaluations of the efficacy of the mitigations that they use. No companies disclose reproducible mitigations.

8 companies (AI21 Labs, Amazon, Anthropic, DeepSeek, Google, IBM, OpenAI, and Writer) disclose a taxonomy of the post-training mitigations that are implemented when developing the model, though only 5 of them (AI21 Labs, Amazon, Anthropic, IBM, and OpenAI) map these mitigations to specific risks. Without information about how post-training mitigations map onto the risks that companies consider when developing their foundation models, it is difficult to determine why a mitigation was implemented or how to conduct a third-party evaluation of whether or not it actually addresses a relevant risk.

In recent years, researchers and policymakers have highlighted the importance of model-weight security (Nevo et al., 2024; NIST, 2024). In particular, most of the US companies committed to the Biden administration to implement certain security practices. Wang et al. (2025a) find that this commitment is the one where there is the least evidence that companies are making good on their commitment: of the 16 companies that signed on

to these commitments, 11 made no information public as of December 31, 2024. The 2025 FMTI, in contrast to the earlier work of Wang et al. (2025a), finds more positive evidence because it led to some companies making new disclosures: now, 8 of the 13 companies do disclose the security measures they implement to prevent unauthorized copying or unauthorized public release of their flagship model's weights. A common concern is that transparency into (cyber)security practices will undermine the very efficacy of those security practices. While there are complex tradeoffs in this space, we believe transparency at the level required by the relevant FMTI 2025 indicator can be achieved with minimal, if any, loss to security. Even companies who justify opacity on other indicators on the grounds of security, namely Anthropic and Google, appear to agree as both companies sufficiently disclose information for this indicator to receive a point.

**Companies share substantial information about the release of their foundation models.** Release is tied for the second-highest scoring subdomain, with an average score of 69%. For example, every company discloses its terms of use and a change log, 2 of 4 indicators scored by every company. Terms of use contain significant information about the developer organization and how liability flows between developers, deployers, and users, providing advantages for developers who disclose such information when mounting a defense in court. Change logs help explain changes to downstream developers, making foundation models easier to use given the rapid pace of continuous updates and deployments. For companies that do not disclose a versioning protocol, however, including Amazon, DeepSeek, Meta, Midjourney, Mistral, and xAI, it may be difficult for downstream developers to understand changes in new model launches or ensure they use the correct iteration of the model.

7 developers disclose the stages of the model's release. Release stages have become increasingly important as developers have invested billions into developing productionized versions of foundation models, meaning that many go through lengthy release processes including internal deployment, working with trusted third party testers, A/B testing, and availability in certain product surfaces and jurisdictions, and general availability. For instance, Google discloses "As appropriate, we use a multi-layered approach to model deployment that may start with testing internally, then releasing to trusted testers externally, then opening up to a small portion of our user base (for example, Gemini Ultra users first). We may also phase our country and language releases, constantly testing to ensure mitigations are working as intended before we expand. And finally, we have careful protocols and additional testing and mitigations required before a product is released to under 18s."

7 developers disclose risk thresholds, which we define as thresholds that determine when a risk level is unacceptably high to a developer (e.g. leading to the decision to not release a model), moderately high (e.g. triggering additional safety screening), or low enough to permit normal usage. Risk thresholds have become more salient as policymakers in the US, EU, and other jurisdictions have mandated that companies draft and disclose such thresholds (METR, 2025). Companies that do not disclose risk thresholds are often non-U.S. firms (Alibaba, DeepSeek, and Mistral), though the lowest scoring companies (Midjourney and xAI) also do not disclose risk thresholds. 9 developers disclosed a foundation model roadmap, a forward-looking roadmap for upcoming models, features, or products, and the top 5 distribution channels for the model. Foundation model roadmaps have become increasingly popular as some companies chart a course towards their goal of AGI while others court investment for larger training runs in the year ahead. Finally, distribution channels are essential for understanding how a model is launched and deployed, which also plays a key role in downstream transparency.

**As with data used to build the model, companies do not share significant data about usage.** Usage data is the lowest-scoring subdomain outside of the upstream domain, with companies scoring just 25% on average. For example, no company provides detailed statistics about the amount of usage of their flagship foundation model. Anthropic has become a leader in transparency for usage data with the release of insights from one million consumer conversations as well as analysis of the economic tasks carried out using Claude (Tamkin et al., 2024; Appel et al., 2025). B2B companies, such as IBM and Writer, disclose that they key lack information from deployers about the intensity and types of usage of the foundation models they develop. And while companies publish privacy policies, we find that privacy policies from Alibaba, Amazon, Google, Meta, xAI and others lack a key component: they do not disclose if and how a user's data will be removed from the corpus of training data if a user requests their data be deleted.

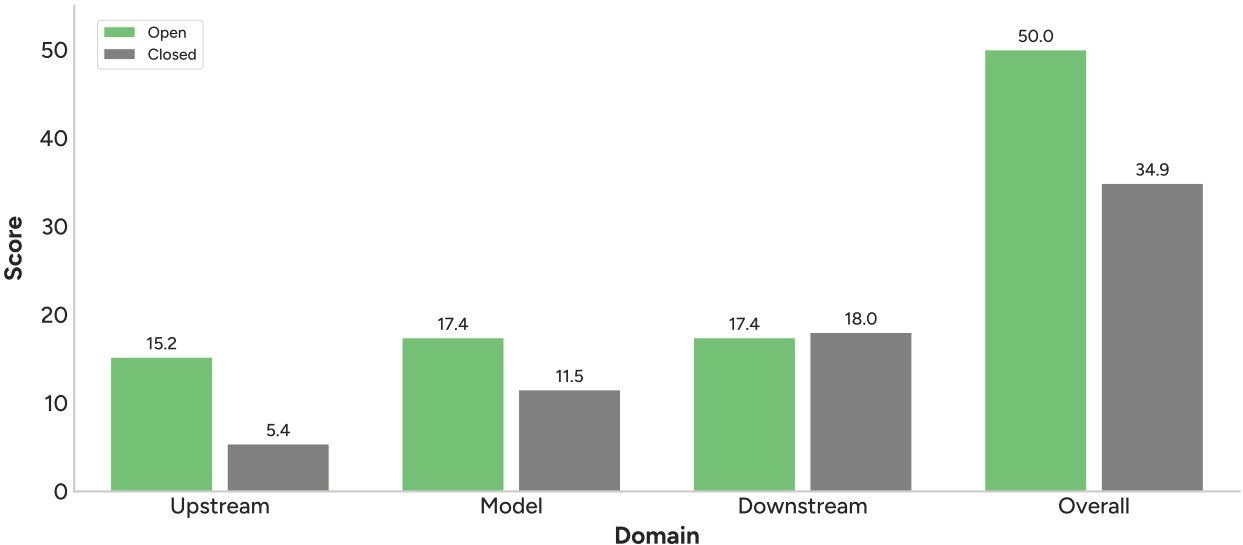

Figure 9: **Scores by Release Strategy.** The average 2025 FMTI score for developers of open-weight vs. closed-weight flagship foundation models. The 5 open developers are AI21 Labs, Alibaba, DeepSeek, IBM, and Meta. The 8 closed developers are Amazon, Anthropic, Google, Midjourney, Mistral, OpenAI, Writer, and xAI. Several companies release other foundation models with different release strategies than the strategy they employ for their designated flagship model for the 2025 FMTI.

## 5.2 Trends across groups of companies

**Openness correlates with—but isn't sufficient for—transparency.** Transparency and openness are two related concepts that are sometimes used interchangeably. Here, we differentiate the two: transparency refers to public access to information whereas openness refers to public access to (at least) model weights (Solaiman, 2023; Kapoor et al., 2024). The 2023 FMTI found that developers with open-weight flagship models generally received higher scores, while the 2024 FMTI found the same trend directionally but with less separation between developers of open-weight and closed-weight flagship models on average. Both past editions clarified that open developers consistently and significantly outperformed closed developers on the upstream domain by providing greater transparency into the data and compute involved in building their flagship models. The 2025 FMTI continues this trend: the 5 open developers (AI21 Labs, Alibaba, DeepSeek, IBM, Meta) outscore the 8 closed developers overall and on every domain when comparing means (Figure 9). Across all three editions, the top-scoring developer releases their flagship model openly whereas the bottom-scoring developers employs a closed release strategy. And, in particular, the upstream domain contributes the most to the margin.

While the FMTI results continue to show an overall correlation between transparency and openness, they also reveal that high-profile open model developers are quite opaque. Alibaba, DeepSeek, and Meta are, arguably, the three most salient open model developers in 2025 and all three companies score in the bottom half in 2025. Consequently, the 2025 FMTI demonstrates the bifurcation among open developers: some developers clearly prioritize transparency (i.e. AI21 Labs, IBM; average = 82.5) with an average score more than 3 times that of their counterparts (i.e. Alibaba, DeepSeek, Meta; average = 25.3). These results confirm that while openness and transparency may correlate, they are meaningfully distinct: some companies score highly on the 2025 FMTI without releasing model weights (e.g. Writer) while others score poorly in spite of releasing model weights.

**Non-consumer-facing companies are considerably more transparent.** Foundation model developers employ different business strategies to commercialize their models and operate in additional related markets

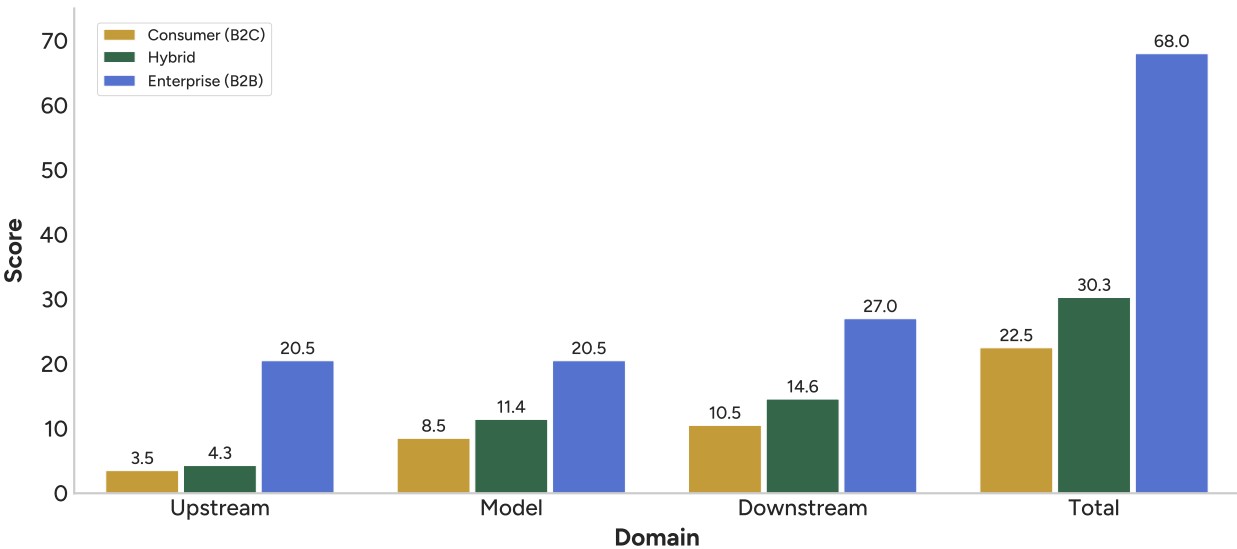

**FMTI Scores by Business Model, 2025**

Source: 2025 Foundation Model Transparency Index

Figure 10: **Scores by Business Model.** The average 2025 FMTI score for developers that employ primarily an enterprise-focused (business-to-business/B2B) vs. a consumer-focused (business-to-consumer/B2C) commercial business model vs. hybrid business model. The 4 B2B developers are AI21 Labs, Amazon, IBM, Writer. The 2 B2C developers are Midjourney, Meta. The 7 hybrid developers are Alibaba, Anthropic, DeepSeek, Google, Mistral, OpenAI, xAI.

(e.g. cloud services, consumer chat bots). We categorize companies based on the business model they employ in relation to their foundation models as either enterprise-facing (B2B), consumer-facing (B2C), or both (hybrid).[12] As shown in Figure 10, the four B2B-focused developers significantly outperform the seven hybrid developers, receiving more than double the overall score—in fact, the top-3 highest scoring companies (IBM, Writer, AI21 Labs) are all B2B. Though this trend applies across domains, this gap is especially apparent in Upstream where B2B companies on average receive around *five times* as many points are the hybrid and consumer developers. It's worth noting, however, that not all B2B companies score uniformly: IBM, Writer, and AI21 Labs receive an overall score of 95, 72, and 66 respectively whereas Amazon scores a 39—nonetheless Amazon is still far from the lowest score with the 6th highest score across all thirteen companies. The two B2C-focused companies (Midjourney and Meta), on the other hand, score the lowest out of the three categories, though only around 8 points less overall than the hybrid companies. This trend holds across domains as well.

One explanation for this is that B2B companies have a greater incentive for transparency: developers integrating models into some downstream application have different requirements and information needs compared to ordinary consumers. A company using a model for some downstream application also takes on the risks from that model, for example. As such, enterprises comparing models may look for assurances in e.g. the kinds of data used to train the model, detailed evaluations on model risks, or downstream mitigations made available to developers. IBM, for example, advertises: "Enterprise AI demands enterprise-grade trust. With some models trained on pirated data or producing biased outputs, it's easy to see why it matters. IBM® Granite® models are built with security, safety and governance at their core, giving you the confidence to build responsible AI."[13]. In comparison, companies that are primarily B2C (e.g. Midjourney) or nonetheless have a large amount of revenue coming from consumers (e.g. OpenAI) may have users who are less able to

---

[12]To provide objective and verifiable categories, we say a company is hybrid if operates both a consumer-facing mobile app on the Google Play Store and a first-party API to perform inference on their flagship foundation model. If the company only operates a mobile app or only a first-party API, we categorize it as B2C and B2B, respectively.

[13]https://www.ibm.com/granite/trust

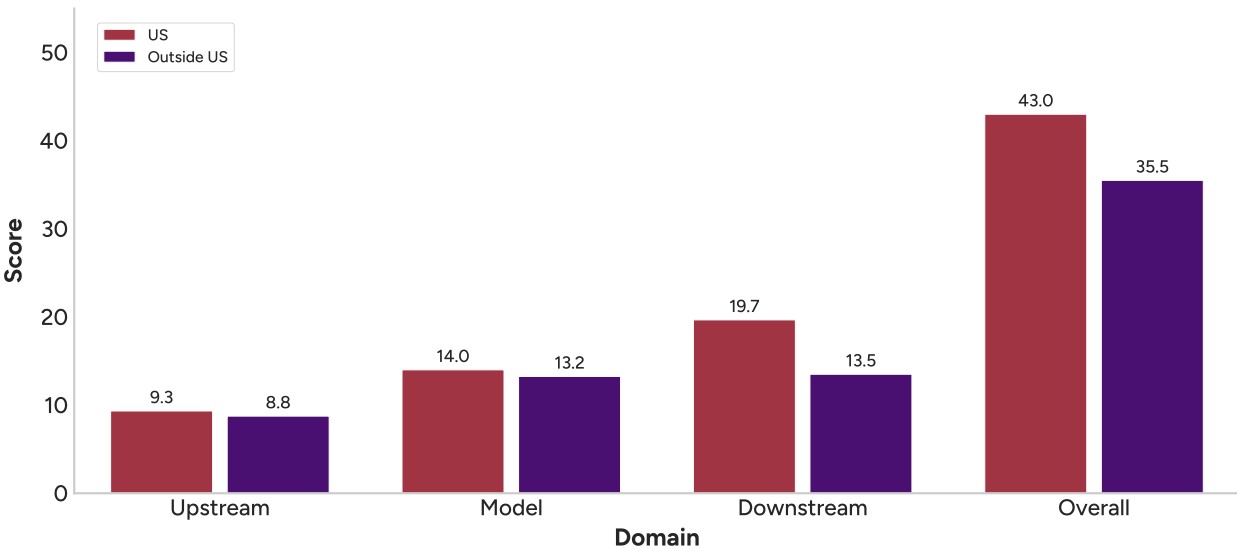

Figure 11: **Scores by Geographic Region.** The average 2025 FMTI score for developers headquartered in the United States vs. outside the United States. The 9 US developers are Amazon, Anthropic, Google, IBM, Meta, Midjourney, OpenAI, Writer and xAI. The 4 Non-US developers are AI21 Labs, Alibaba, DeepSeek, and Mistral. Several companies operate offices around the world, so we focus solely on the location of the company's headquarters. We focus on the US vs. non-US distinction rather than others (e.g. US vs. China) to ensure we have at least 4 companies contributing to each sample.

make direct use of information on e.g. Data Acquisition when deciding what model to use, giving them less of an incentive to provide disclosures on this information.

In addition to the business model, the 2025 FMTI evaluates different types of companies: we score 6 established technology firms (Alibaba, Amazon, Google, IBM, Meta, xAI) and 7 emerging AI startups (AI21 Labs, Anthropic, DeepSeek, Midjourney, Mistral, OpenAI, Writer).[14] Notably, the established firms operate in multiple markets beyond those centric on foundation models (e.g. operating cloud services or online platforms), whereas the startups' strategies are near-exclusively dependent on their ability to develop and deploy high-quality foundation models. The 6 established firms are all publicly traded, while none of the startups are at the time of the 2025 FMTI. In general, the established firms perform slightly better (mean = 46.4; median = 39) compared to the startups (mean = 35.6; median = 33.5).

All six established firms disclose that they use a self-owned cluster as their compute provider. On the other hand, only three startups disclose their compute provider: and one (DeepSeek) uses a self-owned cluster, while the other two (AI21 Labs and Writer) use AWS, Lambda Labs, and Google Cloud. However, this transparency about the compute provider (and the ownership of the compute hardware) does not translate to other properties of the compute like the usage or environmental effects. In fact, when excluding the compute provider indicator, established firms disclose on-average 18.8% of the Compute subdomain, but the startups disclose 21.4%.

**US companies score higher on-average—but also hold the two lowest scores.** Foundation model development has been historically concentrated in a few nations, most notably the United States and China (Maslej et al., 2025). However, a broader set of countries and companies therein are increasingly pursuing foundation model development, especially amidst global discourse on national sovereignty, export controls, and geopolitical tension. As the Foundation Model Transparency Index focuses on the most

---

[14]xAI's work on (frontier) AI is a relatively recent development, but the company now includes the established and previously separate social media platform X. DeepSeek is owned and funded by the established Chinese hedge fund High-Flyer.

influential corporate foundation model developers worldwide, the 2025 FMTI breakdown is 9 US companies, 2 Chinese companies, 1 French company, and 1 Israeli company (Table 1).[15]

Stratifying based on whether companies are headquartered in the United States or not (see Figure 11), we find that US companies do better on average both overall and on every underlying subdomain. US companies demonstrate significant variance with a range of 81 given that IBM scores a 95 while xAI and Midjourney score 14. However, even without the high-scoring IBM, the US average is 36.5 compared to the Chinese average of 29 across Alibaba and DeepSeek. In contrast to the large variation across US foundation model developers, the two Chinese developers have very similar practices. Of the 100 indicators, Alibaba and DeepSeek receive the same score on 88 of the indicators (see Figure 13), which is tied for the most correlated pair of companies of all 78 distinct pairs of companies.[16] For 97 of the 100 indicators, if Alibaba discloses sufficient information on an indicator, then so does DeepSeek.[17]

The discrepancies between US companies and other companies are driven by the downstream domain, specifically highlighting geographic differences in disclosures about usage data, post-deployment impact measurement, and accountability mechanisms. We acknowledge these differences may, despite our best efforts, be driven by information on these topics being less discoverable via English search queries through Google search.[18] Further, some of the constructs may be implicitly be conceptualized in a US-centric or Western model, especially given that the entire FMTI team is based in the United States: for example, Chinese companies may have alternative mechanisms for coordinating with the Chinese government to enable government oversight.

**Disclosures from Frontier Model Forum members are remarkably similarly.** The Frontier Model Forum (FMF) is an "industry-supported non-profit dedicated to advancing frontier AI safety and security".[19] We score five of its six members, namely Amazon, Anthropic, Google, Meta, and OpenAI, which are all highly influential and well-resourced AI companies. Overall, the FMF members occupy the middle of the Index, ranking between positions 4 (Anthropic; score = 43) to 8 (Meta; score = 28). On average, the 5 FMF companies do considerably worse on the upstream domain than the non-FMF companies but slightly better on the downstream domain (Figure 13). These findings align with our observations that smaller companies tend to be more willing to disclose information about how they build models that FMF companies are more guarded about, while FMF companies often expend their greater resources to develop policies (e.g. acceptable use policy, model behavior policy, privacy policy) that constitute a significant fraction of the downstream domain.

Why do the FMF companies achieve such similar scores? A straightforward hypothesis would be that the FMF coordinates the disclosures of its member companies: we are not aware of evidence to support this hypothesis, though the FMF may coordinate how its members satisfy voluntary commitments made to governments (Wang et al., 2025b). Irrespective of the direct role of the FMF, these five companies share many other commonalities and relationships beyond the FMF, which contribute to how the FMF came to be as. Within the scope of the FMTI, we study the correlation between company disclosures and how that relates to FMF membership (Figure 13).

We find that OpenAI and Anthropic have very similar practices (SMC = 0.85) even though Anthropic (score = 46) outscores OpenAI (score = 35) by 11. If OpenAI discloses sufficient information on an indicator, then so does Anthropic with two exceptions.[20] Amazon and Google also have similar indicator-level practices

---

[15]Given the significant geographic concentration of foundation model development and the limits on the number of companies we study, fine-grained geographic inferences are currently not possible with our data.

[16]The other pair with the same score on 88 of the 100 indicators is Midjourney and xAI. However, this overlap is less surprising because both companies score very low at 14 out of 100, hence they must overlap on at least 72 indicators by the pigeonhole principle.

[17]The three exceptions are the indicators on versioning protocol, external developer mitigations, and enterprise mitigations. For all three indicators, Alibaba discloses sufficient information while DeepSeek does not.

[18]Information gathering for the two Chinese companies involved a Chinese language speaker on the FMTI team performing an extra round of information gathering to offset this risk.

[19]https://www.frontiermodelforum.org/about-us/

[20]For the AI bug bounty indicator, OpenAI discloses the details of its AI bug bounty extensively whereas Anthropic does not discloses key terms of the bug bounty. For the data retention and deletion policy indicator, OpenAI sufficiently describes their policy while Anthropic leaves unclear how deletion requests propagate to changing training data for models that are

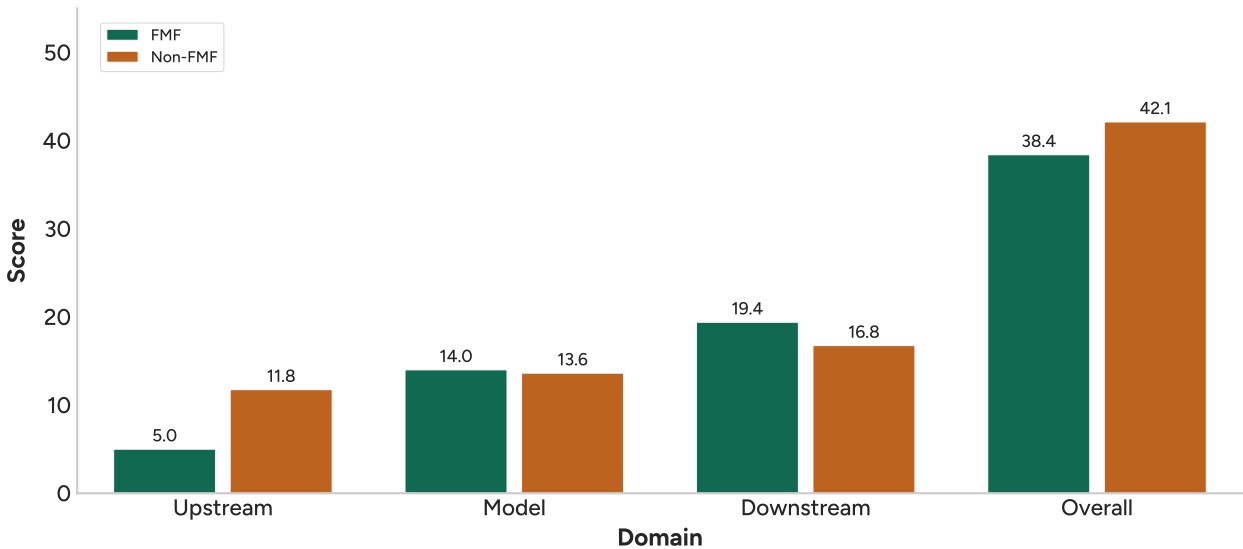

Figure 12: **Scores by Frontier Model Forum membership.** The average 2025 FMTI score for developers in the Frontier Model Forum (FMF) vs. outside the FMF. The 5 FMF developers are Amazon, Anthropic, Google, Meta, and OpenAI. The 8 non-FMF developers are AI21 Labs, Alibaba, DeepSeek, IBM, Midjourney, Mistral, Writer, and xAI. The FMF also includes Microsoft, but Microsoft did not participate in the 2025 FMTI.

(SMC = 0.82) in addition to similar overall scores of 39 and 41, respectively. However, unlike Anthropic and OpenAI, the relationship is less consistent: for 10 indicators, Google discloses sufficient information but Amazon does not, while for 8 indicators, Amazon discloses sufficient information but Google does not. In contrast, Meta patterns differently from every other scored FMF member: across all FMF pairs, every low-correlation pair involves Meta (SMC < 0.7 with all of Amazon, Anthropic, Google, and OpenAI).

**EU AI Act may increase training data transparency in future.** The EU AI Act was enacted as law in 2024: the law imposes specific obligations for foundation model developers.[21] The relevant provisions were clarified in a Code of Practice authored by 13 independent experts[22] and went into effect on August 2, 2025, with penalties for noncompliance triggering on August 2, 2026. While compliance with the EU AI Act is mandatory for any company that makes their models available on the EU market, companies can either sign onto the Code of Practice to indicate they will use it as the means for compliance or demonstrate an alternative means of compliance. At the time of writing, 7 of the 2025 FMTI companies have fully signed onto the code (Amazon, Anthropic, Google, IBM, Mistral, OpenAI, Writer), xAI has partially signed on, and 5 companies have not signed on in any form (AI21 Labs, Alibaba, DeepSeek, Midjourney, Meta). Notably, every 2025 FMTI company based outside Europe and the United States has not signed onto the Code of Practice in any form.

The EU AI Act does not require much public-facing transparency specifically from foundation model developers (Bommasani et al., 2024a). Further, given the recent release of the Code of Practice relative to the completion of the 2025 FMTI, we believe the Code has not influenced the amount of transparency measured

---

currently being trained or those that will be trained in the future. We acknowledge that there are other topics that do not correspond to specific FMTI indicators where OpenAI currently discloses significantly more than Anthropic, such as in relation to post-deployment mental health impacts: see `https://cdn.openai.com/pdf/3da476af-b937-47fb-9931-88a851620101/addendum-to-gpt-5-system-card-sensitive-conversations.pdf`.

[21]The Act uses the term "general-purpose AI model", which is defined similarly to foundation model as defined by Bommasani et al. (2021) and by the Biden White House (Executive Order 14110, 2023).

[22]Rishi Bommasani was an author of the Code of Practice and is a lead of the Foundation Model Transparency Index.

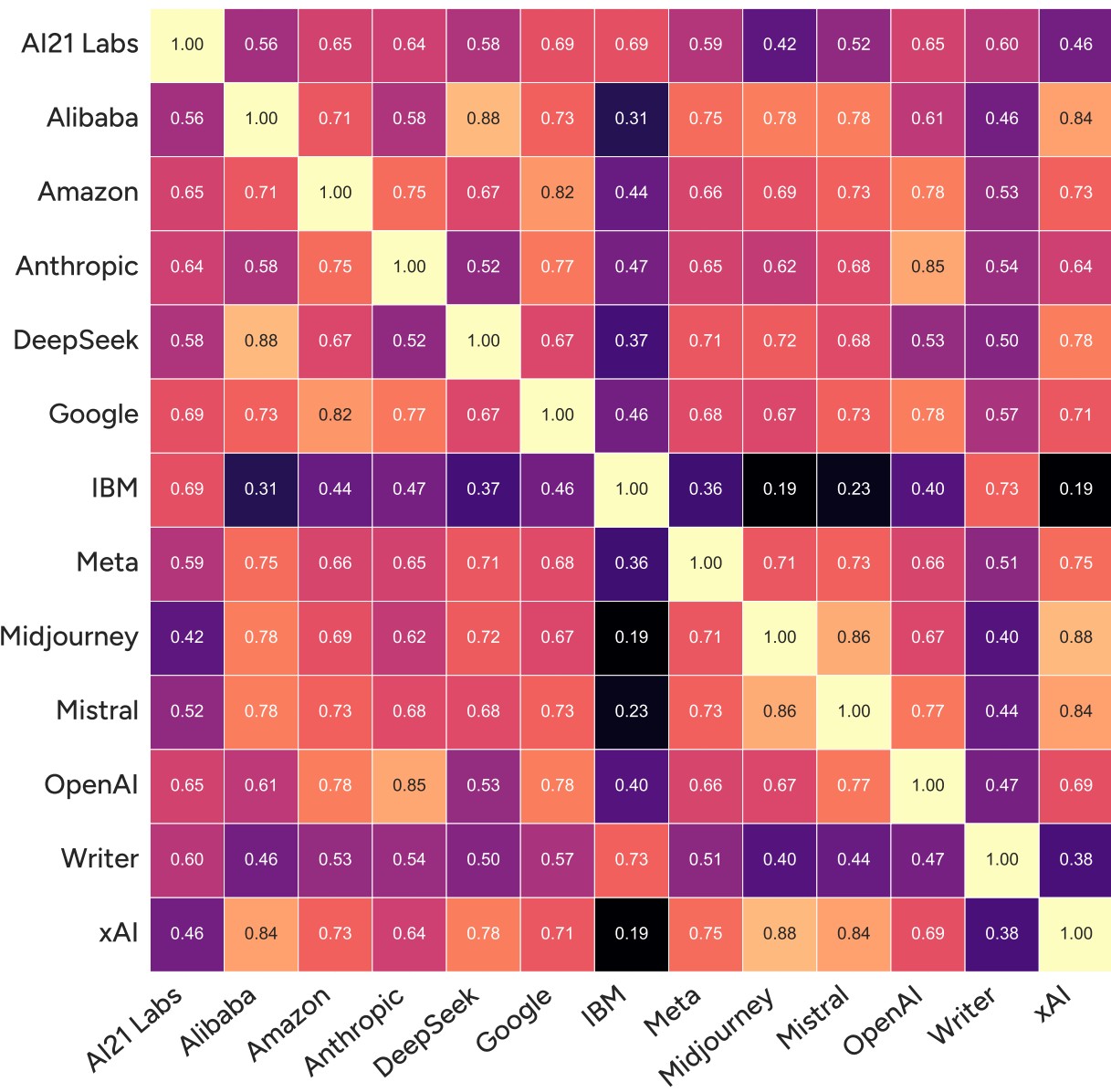

Figure 13: **Correlation in company scores.** The correlation in 2025 FMTI scores between pairs of companies, where the correlation reported is the simple matching coefficient (SMC). The SMC is the fraction of the 100 indicators that both companies receive the same score on (i.e. both companies receive a 0 or both companies receive a 1).

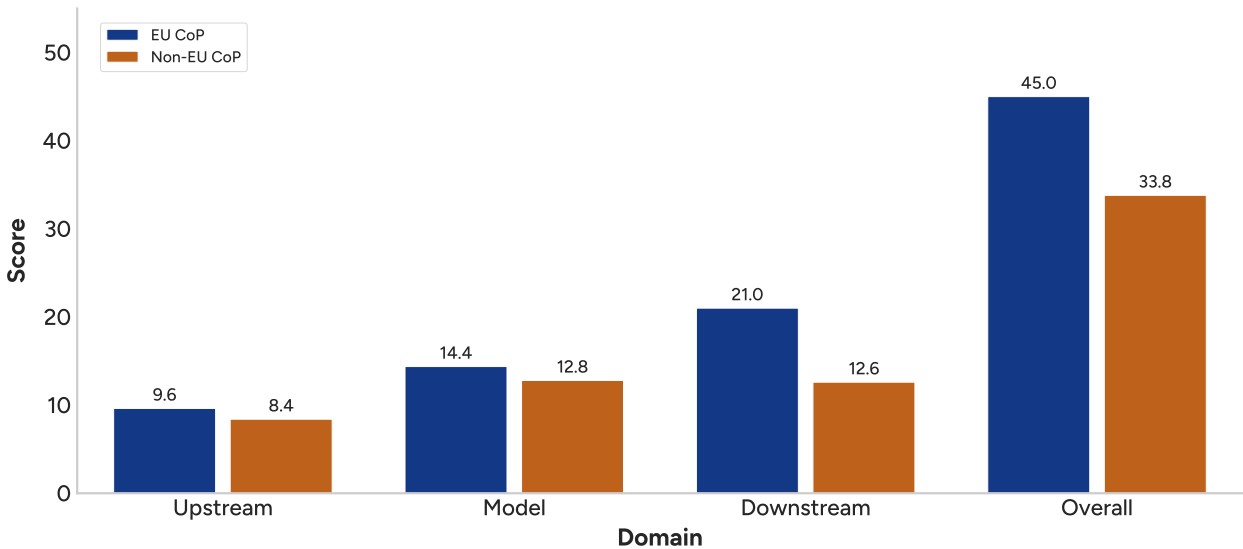

Figure 14: **Scores by EU AI Act Code of Practice signatory status.** The average 2025 FMTI score for developers that signed onto the EU AI Act General-Purpose AI Code of Practice (CoP) vs. those that have not. The 8 CoP signatory developers are Amazon, Anthropic, Google, IBM, Mistral, OpenAI, Writer, and xAI. The 5 non-signatory developers are AI21 Labs, Alibaba, DeepSeek, Meta, and Midjourney.

in the 2025 FMTI. But we do find evidence that the Code of Practice impacts the substance of company's disclosures: for example, Google mentions the risk of harmful manipulation in their most recent Frontier Safety Framework, which directly aligns with the designation of harmful manipulation as a systemic risk in the Code of Practice.[23] The average scores of Code of Practice signatories (including xAI) are 11.2 points higher than non-signatories (Figure 14) with most of the discrepancy coming from downstream disclosures. Both groups exhibit large variation: signatories include high-scoring companies like IBM and low-scoring companies like xAI,[24] while non-signatories include one high scorer in AI21 Labs and 4 low scorers.

Since the penalties under the EU AI Act are not yet enforced, and the Code of Practice was published midway through the 2025 FMTI process, we believe the policy currently has minimal impact on corporate transparency. However, we anticipate two areas where transparency may improve that would be measurable in future editions of the Foundation Model Transparency Index. First, the Code of Practice gestures towards public-facing transparency even if it is unable to mandate it given the limits of the AI Act: "Signatories are encouraged to consider whether the documented information can be disclosed, in whole or in part, to the public to promote public transparency."[25] Therefore, companies may implement this encouragement to be in the good graces of the EU AI Office as the regulator. Second, the EU AI Act mandates that foundation model developers make available to the public a summary of the training data they use.[26] This legal requirement is mandatory for developers irrespective of whether they choose onto the Code of Practice, and irrespective of whether their models are designating as posing systemic risk. The 2025 FMTI demonstrates significant and systemic opacity across almost all foundation model developers on training data transparency, which has been the case throughout the history of the FMTI. The 2025 FMTI indicators have also been deliberately

---

[23]See `https://deepmind.google/blog/strengthening-our-frontier-safety-framework/`.

[24]Of companies scored in the 2025 FMTI, Mistral is the sole company based in the European Union and the lowest-scoring company among those that fully signed onto the Code of Practice.

[25]The Code of Practice contains a more specific element in relation to copyright: "Signatories are encouraged to make publicly available and keep up-to-date a summary of their copyright policy." Akin to how the current FMTI indicators include details about other policies (e.g. model behavior policies, acceptable use policies), future FMTI indicators may deepen focus on the copyright policy in relation to data acquisition indicators given documented issues on data provenance (Longpre et al., 2023a).

[26]The specific content of this summary is defined in a template prepared by the EU AI Office: `https://digital-strategy.ec.europa.eu/en/library/explanatory-notice-and-template-public-summary-training-content-general-purpose-ai-models`.

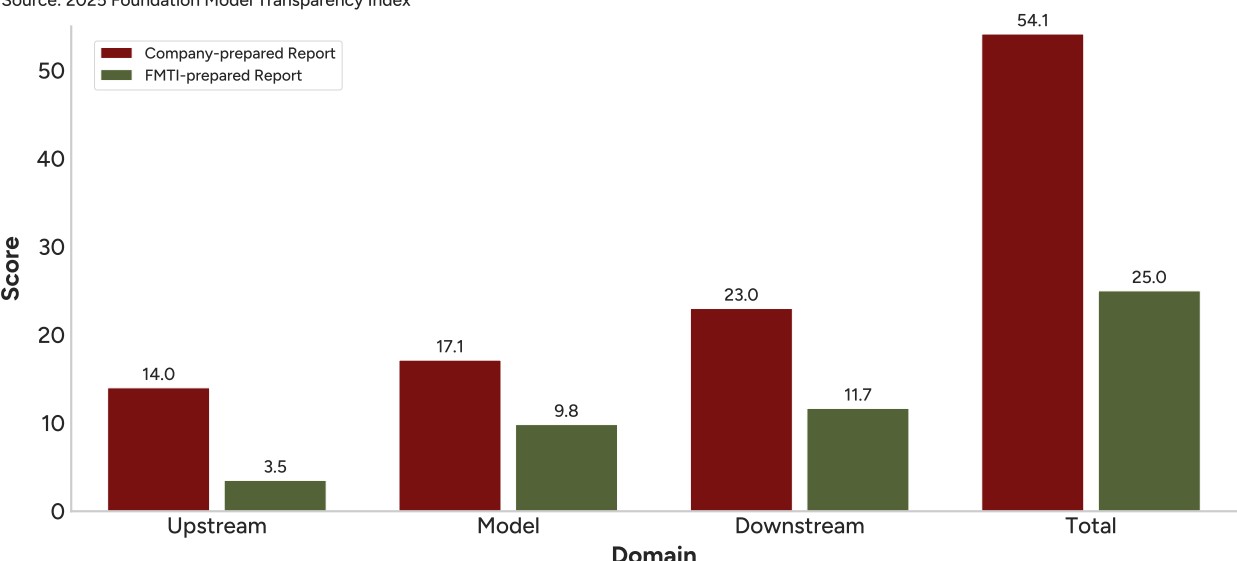

Figure 15: **Scores by FMTI 2025 reporting method.** The average 2025 FMTI score for developers that prepared reports themselves vs. those that did not and the FMTI team prepared reports for instead. The 7 company-prepared reports are for AI21 Labs, Amazon, Google, IBM, Meta, OpenAI, and Writer. The 6 FMTI-prepared reports are for Alibaba, Anthropic, DeepSeek, Midjourney, Mistral, and xAI. Some companies that did not prepare reports still engaged in the response period once they received their initial scores, namely Anthropic and DeepSeek.

designed to align with the taxonomy used in the EU AI Office training data template. Therefore, we expect this legal obligation will cause increased transparency as quantified by the 2026 FMTI and beyond.

**FMTI-prepared reports score around half that of company-prepared reports.** There are two ways in which the reports are initially prepared: by the developers and by the FMTI team. Although developer-prepared reports allow for the most comprehensive assessment of transparency on the indicators and also allow for the release of new information, this leads to a lack of coverage of important developers that are not willing to prepare reports. The preparation of reports by the FMTI team also enables a middle-ground where the developer only provides feedback on an FMTI-prepared report during the response period (this was the case for Anthropic and DeepSeek).

As shown Figure 15, the average score for FMTI-prepared reports tend to be around half that of the company-prepared reports. The principle explanation for this is that, by preparing reports, companies disclose non-public information that the FMTI team could not have found or would be difficult to find. Some indicators may assess companies on information that's unlikely to already exist publicly (e.g. Organization chart). This is especially true for the Upstream domain where 59% of the indicators are such that only the company-prepared reports score a point, compared to 17% in Model and 22% in Downstream. The gap in scores between companies who did and did not prepare reports is also much higher in Upstream versus the other domains: companies who did not prepare reports scored on on-average 25% of the indicators in Upstream versus 57% and 51% for Model and Downstream, respectively.

Another explanation for the gap is that companies who did not produce reports also tend to be less transparent in general. In other words, it could be that companies who did not prepare reports tend to also be less willing to disclose information regardless of whether it's through FMTI. For example, if we compare scores on 6 indicators that depend on an inherently public artifact or property of the model (i.e. if the company were to

get a point, they have to have also publicly released an artifact that the FMTI team would have been able to find),[27] companies who prepare reports on average score 4.6 points and companies who don't score 3.5 points.

## 5.3 Longitudinal FMTI trends

Indexes are a powerful measurement instrument because they can clarify how behavior evolves over time, which includes important structural changes that are hard to attend to in real-time. We built and maintained the Foundation Model Transparency Index over the past three years to realize this potential, especially because existing longitudinal metrics for AI are predominantly either (i) benchmark scores like performance on MMLU (Hendrycks et al., 2021), which are effective for understanding the technology but not its societal impacts or (ii) financial indicators like annual revenue, which are effective for understanding the macroscopic commercial performance of AI companies but lack specificity to the AI industry. In particular, the 2025 FMTI includes data for three years, so we can begin to see trends in the overall trajectory for transparency and underlying heterogeneity across individual companies that we have tracked for multiple years. Since the set of companies scored each year changes based on the relevance of those companies in that year to foundation model development, we perform longitudinal analyses for companies scored in both 2024 and 2025 as well as companies scored across all three years.

Overall, the average FMTI scored declined from a 58 in 2024 to a 40.69 in 2025. However, several sources may contribute to this decline (e.g. different indicators, different companies, different reporting mechanisms, different substantive disclosures about different flagship models). To control for one source of variation, we fix the companies to be those scored in both 2024 and 2025 in Figure 16. Of these 9 companies, 7 scored lower in 2025 than they did in 2024. Further, the decline in transparency is not limited to quantitative score reduction, but also procedural regression. While all 9 of these companies prepared transparency reports to engage with the FMTI in 2024, only 7 companies did so in 2024.[28] While the aggregate change suggests a systemic industry-wide decline in transparency, underlying heterogeneity across companies reveals a more complex reality. Across the 9 developers scored in both 2024 and 2025, two companies significantly increased their scores (Writer from 56 to 72, IBM from 64 to 95), four companies decreased their scores slightly (AI21 Labs, Amazon, Anthropic, Google), one company decreased by a considerable amount (OpenAI from 49 to 35), and two companies precipitously decreased their scores (Meta from 60 to 31, Mistral from 55 to 18). These changes reveal divergent evolution in practices: for example, Meta and IBM scored very similarly in 2024 (a margin of 4 points), but score very differently in 2025 (a margin of 64 brought about by Meta's score dropping by 29 points while IBM's score rose by 31). More granularly, the four companies with the largest year-over-year change brought about these changes in very different ways: Writer largely improved its downstream disclosures (+11 downstream; +16 overall), Mistral entirely curtailed its upstream disclosures along with considerable reductions in other domains (-15 upstream; -37 overall), IBM improved across the board (+12 upstream, +6 model, +13 downstream), and Meta declined across the board (-8 upstream, -13 model, -8 downstream).[29]

To accumulate more data over time, we consider the 6 companies (AI21 Labs, Amazon, Anthropic, Google, Meta, OpenAI) that have been scored in every edition of the Foundation Model Transparency Index (Figure 17). From 2023 to 2024, every company increased its score with AI21 Labs and Amazon in particular showing large improvements while the other 4 companies changed their practices more marginally. In contrast, every company decreased its score in the past year with Meta and OpenAI showing large declines. These changes are particularly striking when we consider the rankings of these companies: Meta and OpenAI were the most and second-most transparent in the inaugural 2023 FMTI but now are the least and second-least transparent of these six companies in the 2025 FMTI. Taking the change observed over the Index's entire tenure by comparing 2023 scores to 2025, we see that AI21 Labs and Amazon have significantly increased their scores (+41 and +27), Anthropic has considerably increased its score (+10), Google has stayed roughly constant (+1), OpenAI has considerably decreased its score (-13), and Meta has sharply decreased its score (-23).

---

[27]Specifically, this is "Code Access", "Open weights", "Change log", "Terms of use", "Intermediate Tokens", and "Documentation for Responsible Use"

[28]While Anthropic did not prepare its own transparency report in 2025, we acknowledge that their team extensively engaged with the FMTI team, so it is unclear whether their overall amount of effort spent engaging with the FMTI team or on transparency more generally increased or decreased across the two years.

[29]Note that the number of indicators per domain changed slightly from 2024 to 2025 as described in §3.2.

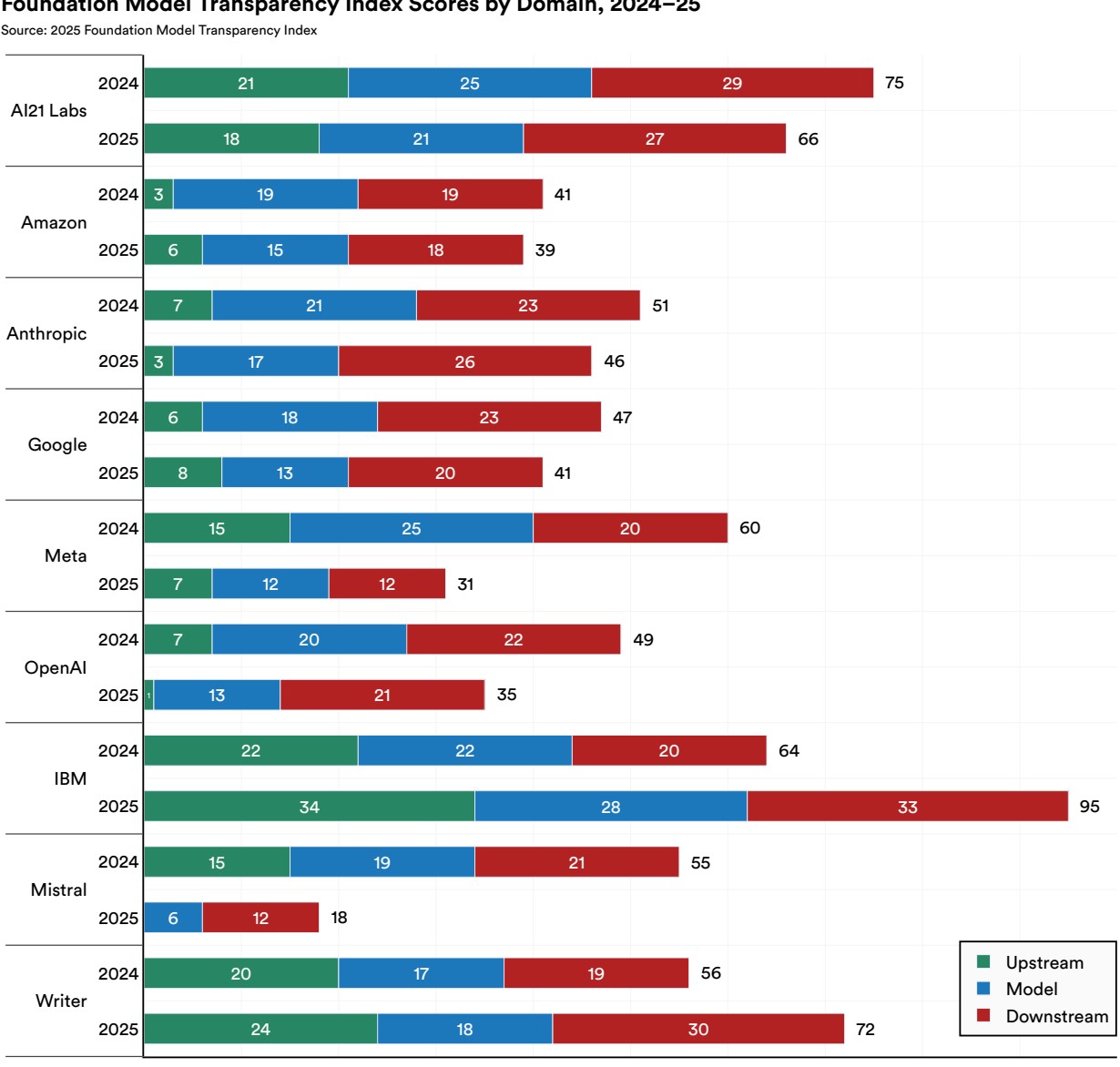

**Foundation Model Transparency Index Scores by Domain, 2024–25**

Source: 2025 Foundation Model Transparency Index

Figure 16: **Scores by Domain from 2024 to 2025.** 9 companies have been assessed in both 2024 and 2025. In 2024, all 9 companies prepared their own reports, whereas in 2025 only 7 companies did. The FMTI team prepared the transparency reports for Anthropic and Mistral for the 2025 FMTI.

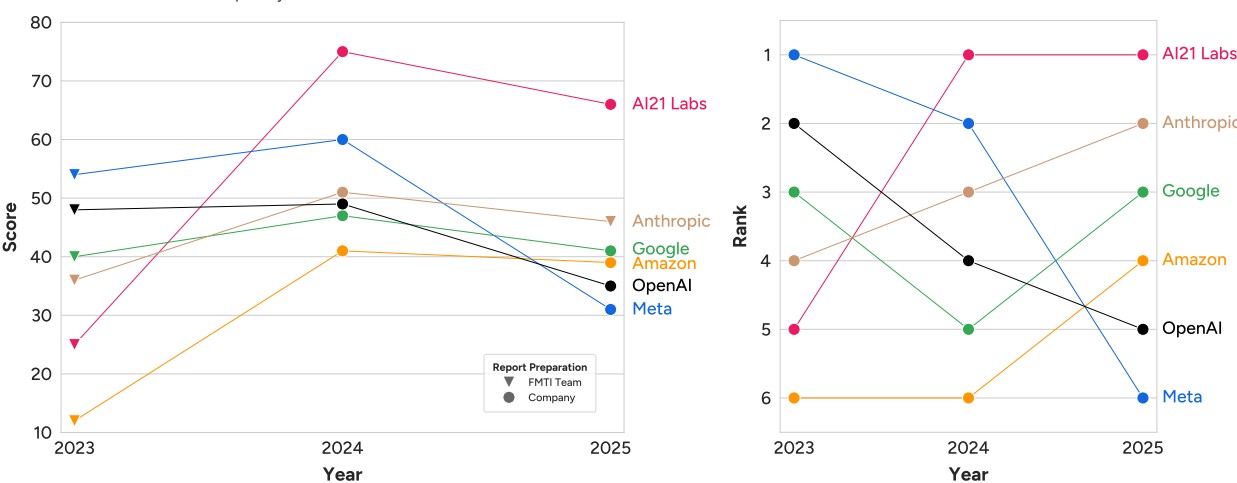

Figure 17: **Scores from 2023 to 2025.** Six companies have been assessed across all three years of the Index. In 2025, Anthropic is the only company out of these six for which the report was prepared by the FMTI team (although, the developer also provided feedback later in the process). Notably, the two companies that score the highest out of the six in 2023 end up scoring the lowest in 2025.

**FMTI disclosures that demonstrate regress and progress.** To conduct the most controlled comparison of transparency over time, we fix both the indicators and companies studied. As we describe above, 9 companies are scored in both 2024 and 2025 while 6 companies are scored in all 3 years. And as we depict in Figure 6, 9 indicators have stayed exactly the same across all 3 years. Given this set of fixed companies and indicators, we explore how each company's disclosures for these indicators have changed over time.

Overall, company disclosures have become verbose over time, in part due to changes in the reporting method from FMTI-prepared reports in 2023 to company-prepared reports in 2024 and 2025 (with the exception of Anthropic). This verbosity is largely a byproduct of the more specific instructions provided by the FMTI team to companies preparing transparency reports: we firmly encourage companies to provide self-contained disclosures for each indicator in 2025, rather than simply pointing to existing documentation like privacy policies, technical reports, and system cards. For example, for the indicator on risk evaluation, IBM directly provides the quantitative risk evaluation results in their 2025 transparency report rather than pointing to existing documents like the model's technical report:

> We evaluate the risks for each of the harms as measured by the ATTAQ framework (high score is good):
> Granite-3.0-8B-Instruct
> 1) Explicit content - 0.85
> 2) Deception - 0.87
> 3) Discrimination - 0.85
> 4) Harmful information - 0.86
> 5) Violence - 0.86
> 6) Substance abuse - 0.84
> 7) PII leakage - 0.81

We find that the increased verbosity coincides with disclosures that more directly assess the specific elements that are required to award a point, rather than just generically addressing the indicator as a topic. More substantively, the 2025 disclosures are more standardized across companies than in previous years. We attribute this change to the example disclosures we provided for the first time alongside the 2025 FMTI indicators in the reporting template we provided companies. This is most true in the case of IBM, where almost every disclosure mirrors the formatting of the examples we provide.

| Company | Indicator | 2024 | 2025 |
|---|---|---|---|
| AI21 Labs | Data Size | A corpus of 1.2 trillion tokens. | No disclosure. |
| OpenAI | Compute Provider | Microsoft Azure | No disclosure. |
| Amazon | Versioning Protocol | In Bedrock Console, each Titan model has been labeled with its model version number. When we release new versions of Titan Text LLMs, customers may experience changes in performance on their use cases. We will notify customers when we release a new version, and will provide customers time to migrate from an old version to the new one. | Amazon does not publicly disclose versioning protocols for Amazon Nova family of models, however, Amazon Bedrock assigns each model available on Amazon Bedrock a model lifecycle stage. |
| Meta | Compute Hardware (*Type and Amount*) | 16000 NVIDIA A100s | Model pre-training utilized a cumulative of 7.38M GPU hours of computation on H100-80GB (TDP of 700W) type hardware, per the table below. Training time is the total GPU time required for training each model and power consumption is the peak power capacity per GPU device used, adjusted for power usage efficiency |
| Mistral | Open Weights | Mistral 7B is an open-weights model | Medium 3 is a closed-weights model |

Table 3: **Example regressions in company disclosures from 2024 to 2025.** This table shows examples of the ways in which companies disclosed less information in 2025 than in 2024. Some regressions are straightforward: information disclosed in 2024 is no longer disclosed in 2025 (e.g. AI21 Labs & Data Size, OpenAI & Compute Provider). Some regressions accompany changes in the information disclosed in existing public artifacts like developer documentation (e.g. Amazon & Versioning Protocol). Some regressions involve only a partial reduction in the disclosed information (e.g. Meta & Compute Hardware: in 2025, the type is disclosed but not the amount). Finally, some regressions implicate larger changes in a developers' approach to model development and release (e.g. Mistral & Open Weights).

In other cases, disclosures have become more verbose because companies now acknowledge or justify their opacity in relation to some indicators whereas previously they did not write anything. For example, for indicators where Anthropic does not disclose information, they instead write:

> This information is proprietary and not disclosed publicly to protect competitive advantages and intellectual property.

—which they state for 27 of the 100 indicators. In contrast, Amazon acknowledges they do not publicly disclose information for 33 of the 100 indicators. And Google provides both acknowledgments of the lack public disclosure for 30 of 100 indicators, as well as justifications in some cases that are more specific to the particular indicator by identifying risks or challenges related to indicator-level disclosure. The justifications

they provide are as follows: data poisoning risks in relation to disclosures on training data and its acquisition, risks from revealing trade secrets and facilitating reverse engineering in relation to disclosures on compute and architecture, the absence of industry-standard methods for reporting environmental impacts, risks of compromising highly confidential business information in relation to disclosing total costs for model development, and measurement complexity for some aspects of identifying the most consequential distribution channels and downstream impacts. For example, for indicators on compute usage, Google states:

> As a Frontier Model Forum founding member, we endorse the FMF methodology, but we do not publicly disclose this specific information since specific numbers of FLOPs, along with parameters, could give competitors an idea of our proprietary approach when asked to also provide information like model architecture and a summary of training data. When combined, specific numbers could help bad actors triangulate even more specifics of our approach. The benefit of potentially exposing specific numbers is outweighed by the risk of disclosure of trade secrets, and relatedly, the risk of potential security vulnerabilities through reverse engineering.

In certain cases, we observe apples-to-apples regressions, where a company in a previous year disclosed information on a certain indicator but no longer does in 2025. These regressions align with broader structural shifts in the field of artificial intelligence as the field has transitioned from being exclusively a research discipline to a commercial market. Meta's FMTI score dropped by 23 points from 2024 to 2025 and part of this change is explained by direct reductions in information disclosure regarding new models. For example, in 2024 Meta disclosed the following permitted, restricted, and prohibited behaviors for Llama 2 in their technical report (Touvron et al., 2023):

> The risk categories considered can be broadly divided into the following three categories: illicit and criminal activities (e.g. terrorism, theft, human trafficking); hateful and harmful activities (e.g. defamation, self-harm, eating disorders, discrimination); and unqualified advice (e.g. medical advice, financial advice, legal advice)

However, Meta does not make a similar disclosure for Llama 4 via any means nor did it released a technical report for Llama 4. Other regressions in the case of Meta include the upstream indicators on instructions for creating data, model objectives, and model stages.

OpenAI is the other company with a significant score decrease from 49 in 2024 to 35 in 2025. In 2024, OpenAI disclosed information about their protocol for enforcing acceptable use policies via the GPT-4 system card:

> We use a mix of reviewers and automated systems to identify and enforce against misuse of our models. Our automated systems include a suite of machine learning and rule-based classifier detections that identify content that might violate our policies. When a user repeatedly prompts our models with policy-violating content, we take actions such as issuing a warning, temporarily suspending, or in severe cases, banning the user. Our reviewers ensure that our classifiers are correctly blocking violative content and understand how users are interacting with our systems. These systems also create signals that we use to mitigate abusive and inauthentic behavior on our platform. We investigate anomalies in API traffic to learn about new types of abuse and to improve our policies and enforcement.

Yet in 2025, the o3 system card is much more sparse on usage policy enforcement:

> [...] the model can refuse to invoke the image generation tool if it detects a prompt that may violate OpenAI's policies.

We even observe high-scoring companies like AI21 Labs exhibiting indicator-level regressions: in 2024, they disclosed training compute, energy usage, and carbon emissions ($6.00 \times 10^{23}$ FLOPs, $570,000 - 760,000$ kWh, $2 - 300$ tCO2eq) but in 2025 they do not, instead saying:

> While we are aware that there are potentially additional environmental impacts of training (e.g. water usage for cooling), each of our compute providers have active sustainability and carbon offset programs specific to their datacenter locations and operations. For details see https://blog.google/outreach-initiatives/sustainability/our-commitment-to-climate-conscious-data-center-cooling/ and https://sustainability.aboutamazon.com/natural-resources/water.

In other cases, the disclosures reveal changes that do not lead to score changes, but that are less precise than in the past. In 2024, AI21 Labs disclosed the hardware they used to train Jurassic-2 was "768 NVIDIA A100s and 2048 TPUv4s", whereas in 2025 they disclose that:

> Jamba was trained using a combination of NVIDIA A100, NVIDIA H100 and Google TPUs v4. In all, about 1,500 processors were used with roughly a 60:40 split Nvidia to Google.

## 6 Conclusion

The 2025 Foundation Model Transparency Index demonstrates the value of a sustained effort to quantify the transparency of major AI companies. Organizational approaches minimally clarify how organizational practices change over time, and potentially improve the incentives that govern company behavior to better align with the public interest. We find evidence to support the latter ambition for an index, namely in the performance of AI21 Labs, Writer, and especially IBM this year. Under this view, the Foundation Model Transparency Index belongs not only to the class of measurement instruments used to study AI companies, but also the class of mechanisms used to shape their practices. Public policy serves as another core mechanism for shaping AI companies and, in particular, advancing transparency as an instrumental good for multiple societal goals. We look forward to working with policymakers, as well as stakeholders within and external to AI companies, to build a richer information environment on leading AI companies and their societal impacts.

**Acknowledgments.** We thank the FMTI Advisory Board (Arvind Narayanan, Daniel E. Ho, Danielle Allen, Daron Acemoglu, Rumman Chowdhury) for their feedback and guidance. We thank Ben Brooks, Nathan Lambert, and Stephen Casper for review of the 2025 FMTI indicators. We thank Brian Tse, Kwan Yee Ng, Markus Anderjlung, and Yuan Cheng for assistance in engaging Chinese companies. We thank Charles Foster, Miranda Bogen, Helen Toner, Ilan Strauss, Risto Uuk, Daphne Keller, and Sarah Schwettmann for helpful discussion. We especially thank Loredana Fattorini for her work on the visuals for this project.

**Foundation Model Developers.** We thank the following individuals at their respective organizations for their engagement with our effort: We emphasize that **this acknowledgment should not be understood as an endorsement of any kind by these individuals**, but simply that they were involved in our engagement with their organizations.

- AI21 Labs — Shanen J. Boettcher, Yoav Shoham

- Amazon — Claire O'Brien Rajkumar, Sara Liu, Peter Hallinan

- Anthropic — Kamya Jagdish, Ashley Zlatinov

- Google — Reena Jana, Patrick Gage Kelley, Lauren Rock, Alex Vasiloff, Allison Woodruff, Aalok Mehta, Danielle Osler

- IBM — Derek Leist, Kate Soule, Aliza Heching, Kush Varshney

- Meta — Harrison Rudolph, Rachad Alao, Polina Zvyagina

- Mistral — Marie Pellat, Paula Kurylowicz, William El Sayed, Sophia Yang, Guillaume Lample, Arthur Mensch

- OpenAI — Cedric Whitney, David Robinson, Lama Ahmad, Sandhini Agarwal, Yo Shavit

- Writer — Rowan Reynolds, Karen Situ, Waseem AlShikh, Ellen Woodcock, May Habib

- xAI — Dan Hendrycks, Yuhuai Wu

**Conflict of Interest.** Given the nature of this work (e.g. potential to significantly impact particular companies and shape public opinion), we proactively bring attention to any potential conflicts of interest, deliberately taking a more expansive view of conflict of interest to be especially forthcoming.

- Alexander Wan is not, and has not, been affiliated with any of the companies evaluated in this effort.

- Betty Xiong is not, and has not, been affiliated with any of the companies evaluated in this effort.

- Kevin Klyman was not, and had not been affiliated with any of the companies evaluated in this effort until October 2025. In October 2025, Kevin Klyman began a role at Google. All FMTI 2025 scores were finalized before this date and he was not involved in the project after this date. Kevin's contributions were independently reviewed by Rishi Bommasani.

- Nestor Maslej is not, and has not, been affiliated with any of the companies evaluated in this effort.

- Percy Liang was a post-doc at Google (September 2011–August 2012), a consultant at Microsoft (May 2018–May 2023), and a co-founder of Together AI (July 2022–present). He is not otherwise affiliated with any of the companies evaluated in this effort.

- Rishi Bommasani is not, and has not, been affiliated with any of the companies evaluated in this effort.

- Sayash Kapoor worked at Meta until December 2020. He has not since worked for the company, and is not otherwise affiliated with any of the companies evaluated in this effort.

- Shayne Longpre interned at Google in 2022 and 2024. He has not since worked for the company, and is not otherwise affiliated with any of the companies evaluated in this effort.

In addition to the individual-level conflicts of interest, the Foundation Model Transparency Index is housed at the Stanford Center for Research on Foundation Models (CRFM) within the Stanford Institute for Human-Centered Artificial Intelligence (HAI). Beyond the scope of this project, CRFM and HAI maintain industry relationships with many of the companies assessed in the 2025 FMTI, including partnerships with Google and IBM. These industry affiliations with CRFM and HAI did not affect the research methodology, data collection, scoring process, or results.

## Broader Impact Statement

As discussed in previous editions of FMTI (Bommasani et al., 2023a; 2024b), transparency alone is insufficient. First, transparency should not be conflated with responsibility. A company that freely discloses their unethical practices can score highly on the index without meaningfully improving these practices. Companies may also use transparency to deflect scrutiny from regulators or the public while avoiding actual, substantive changes. As such, there's a risk that the index will lead to only superficial changes in behavior or even perpetuate harmful practices (Zalnieriute, 2021). Nonetheless, we consider measuring and improving the transparency of foundation model development practices to be a positive step forward, especially when developed with input from the broader community.

Additional risks may arise from changes made in the newest edition of the index. For example, the current edition incorporates a heterogeneous scoring process in which the FMTI team prepares reports for companies that did not submit them. This may disadvantage developers that have fewer resources to dedicate to responding to the index as their scores would not incorporate clarifications or new information. To mitigate this, we send all companies the initial scored reports to provide them with an opportunity to respond even if they did not prepare the disclosures.

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

# A    Automated evaluation agent architecture

AI agents are increasingly deployed for complex research tasks, from systematic literature reviews to data analysis. Given the labor-intensive nature of manually collecting and acquiring over 1000 disclosures across multiple companies, which cumulatively consumes months of expert time from the FMTI team, we developed an automated evaluation agent to test whether language model-based AI agents could reliably assess company transparency at scale. For the 2025 evaluation, we evaluated the agent on the six companies that did not prepare reports: Anthropic (Claude 4), xAI (Grok 3), Alibaba (Qwen 3), Deepseek (Deepseek R1), Midjourney (Midjourney V7), and Mistral (Medium 3). While the primary purpose of this exercise was to understand the utility of current agents for the FMTI team and similar initiatives, we also gleaned broader insights into the current capabilities and limitations of AI agents for complex evaluation tasks. All scores published in the 2025 FMTI reflect the FMTI team's judgments, with multiple steps of validation including engagement with the companies, and none of the published scores were proposed or directly determined by any AI system.

The automated evaluation system is built around Anthropic's Claude 4 API. The system loads the 100 FMTI indicators and specifies relevant web domains for each evaluated organization (e.g. company website, company Hugging Face pages, company Github pages). The agent utilizes Anthropic's native tool-calling capabilities to perform web searches constrained to company-specified domains and to extract content from PDF documents found during searches. This PDF extraction capability addresses the common practice of companies publishing detailed technical information in model cards and technical reports. The web search supports up to 10 queries per indicator, allowing for iteratively refining the search based on initial findings. The restriction on domains ensures evaluation focuses on official company communications, which is the same restriction we apply throughout the FMTI, rather than disclosures by other parties (e.g. the media) that may bias results and not be officially confirmed as accurate by the company.

The system implements concurrent processing with configurable limits and rate limiting through exponential backoff with jitter to handle API constraints. The evaluation agent enforces structured output through JSON schema validation, requiring the language model to return evaluations containing a binary score (0 or 1), evidence found, missing information, and detailed justification for the score. We implement error handling to prevent single indicator failures from terminating entire company evaluations. Results are stored with detailed metadata and incremental progress tracking, ensuring evaluation progress is preserved during system interruptions. The system generates structured reports summarizing evaluation results across companies, including pass rates and detailed indicator-by-indicator breakdowns, facilitating both automated analysis and human review of evaluation results.

## A.1    Agent prompt

```
"""Evaluate {company_name}'s {model_name} model on this FMTI transparency
    indicator:

**Indicator:** {indicator.name}
**Definition:** {indicator.definition}
**Specific Criteria:** {indicator.notes}
**Example Good Disclosure:** {indicator.example}

Search {', '.join(company_domains)} and related sources to find evidence. Look
    for:
— Technical documentation and reports
— Model cards
— Blog posts and announcements
— Policy documents
— Research papers
```

```
IMPORTANT: If you find links to PDF documents in the websites (especially
    model cards, system cards, or technical reports), use the extract_pdf tool
     to get their full content instead of just relying on search snippets.

After thorough searching, provide your evaluation. Please format your entire
    response as a single JSON object enclosed in triple backticks ('''json ...
     ''') with the following keys:
- "score": [integer: 1 if ALL criteria clearly satisfied, 0 otherwise]
- "confidence": [float: 0.0-1.0]
- "evidence_found": [array of strings: List key findings with URLs. If none,
    use an empty array.]
- "missing_information": [array of strings: What criteria were not satisfied.
    If none, use an empty array.]
- "justification": [string: Detailed explanation of your scoring decision]

Example JSON output format:
'''json
{{
  "score": 0,
  "confidence": 0.75,
  "evidence_found": [
    "Finding 1 (URL: http://example.com/doc1)",
    "Finding 2 (URL: http://example.com/blog2)"
  ],
  "missing_information": [
    "Specific detail X was not found.",
    "Criterion Y is not addressed."
  ],
  "justification": "The company provides some relevant information but does
    not fully meet all criteria for this indicator because X and Y are
    missing."
}}
'''"""
```

We use the agent's evidence found, missing information, and justification columns as the final output for the transparency report. We noticed that the agent's outputs are much more verbose than the FMTI team's transparency report, increasing the scope for false positives (irrelevant information surfaced during the search).

|  | Alibaba | Anthropic | DeepSeek | Midjourney | Mistral | xAI | Average |
|---|---|---|---|---|---|---|---|
| Overall (Agent misses info) | 16 | 9 | 9 | 4 | 6 | 4 | 8.00 |
| Overall (Agent finds additional info) | 16 | 13 | 11 | 8 | 17 | 13 | 13.00 |
| Upstream (Agent misses info) | 8 | 2 | 4 | 1 | 0 | 2 | 2.83 |
| Upstream (Agent finds additional info) | 0 | 1 | 1 | 0 | 0 | 1 | 0.50 |
| Model (Agent misses info) | 3 | 5 | 3 | 0 | 3 | 1 | 2.50 |
| Model (Agent finds additional info) | 2 | 2 | 5 | 5 | 5 | 4 | 3.83 |
| Downstream (Agent misses info) | 5 | 2 | 2 | 3 | 3 | 1 | 2.67 |
| Downstream (Agent finds additional info) | 14 | 10 | 5 | 3 | 12 | 8 | 8.67 |

Table 4: **Evaluation of information retrieval performance of 2025 FMTI AI agent.** We evaluate our AI agent for how frequently it misses information found by the FMTI team, and how many times it finds additional information.

# B    Bootstrap uncertainty of rankings

| Company | Median Rank | Lower | Upper |
|---|---|---|---|
| IBM | 1.0 | 1.0 | 1.0 |
| Writer | 2.0 | 2.0 | 3.0 |
| AI21 Labs | 3.0 | 2.0 | 3.0 |
| Anthropic | 4.0 | 4.0 | 6.0 |
| Google | 5.0 | 4.0 | 7.0 |
| Amazon | 6.0 | 4.0 | 8.0 |
| OpenAI | 7.0 | 5.0 | 10.0 |
| DeepSeek | 8.0 | 5.0 | 9.5 |
| Meta | 8.5 | 6.0 | 10.0 |
| Alibaba | 10.0 | 8.0 | 10.5 |
| Mistral | 11.0 | 10.0 | 13.0 |
| Midjourney | 12.5 | 11.0 | 13.0 |
| xAI | 12.5 | 11.0 | 13.0 |

Table 5: **Company rankings with confidence intervals.** Median rank and (lower, upper) bounds for each company, calculated through bootstrap samples.

We estimate the estimate the uncertainty of the firm-level scores through bootstrap samples. In Table 5, we include the 95% CI and median ranks of each company. Our overall rankings are stable: for example, the width of the 95% CI is three rank positions or fewer for 9 of the 13 evaluated companies.

# C    Complete indicator descriptions

Table 6: The definition and notes for all 100 indicators in the 2025 FMTI.

| Indicator | Definition | Notes |
|---|---|---|
| Data acquisition methods | What methods does the developer use to acquire data used to build the model? | Which of the following data acquisition methods does the developer use: (i) acquiring existing public datasets, (ii) crawling the web, (iii) using data acquired via its existing products and services, (iv) licensing existing data from external parties, (v) having humans create or annotate new data, (vi) using models to generate new data, or (vii) other data acquisition methods not captured by the above. For example, if the developer uses reinforcement learning from human feedback to train models using model-generated outputs with human preference annotations, this would satisfy categories (v) and (vi). Alternatively, if the developer post-trains its model using off-the-shelf preference data (for example, the Alpaca dataset), this would satisfy category (i). |
| Public datasets | What are the top-5 sources (by volume) of publicly available datasets acquired for building the model? | We define a source as the entity or means by which the developer acquires data. We define the top-5 sources as the top-5 sources by data volume. |

*Continued on next page*

| Indicator | Definition | Notes |
|---|---|---|
| Crawling | If data collection involves web-crawling, what is the crawler name and opt-out protocol? | We award this point for disclosure of the crawler name and opt-out protocols, including if/how they respect the Robots Exclusion Protocol (robots.txt). |
| Usage data used in training | What are the top-5 sources (by volume) of usage data from the developer's products and services that are used for building the model? | We define usage data as data collected from the use of a developer's products or services. |
| Notice of usage data used in training | For the top-5 sources of usage data, how are users of these products and services made aware that this data is used for building the model? | We define usage data notice as the proactive disclosure to users of how their data is used for model development. For example, via a pop-up with a description, a link to the privacy policy, or link to a description of company practices. |
| Licensed data sources | What are the top-5 sources (by volume) of licensed data acquired for building the model? | We define a source as the entity from which the developer acquires data. For example, the Associated Press is reportedly a source of licensed data for OpenAI. |
| Licensed data compensation | For each of the top-5 sources of licensed data, are details related to compensation disclosed? | We award this point if the model developer describes the compensation structure specified in the contract with the data source or indicates they are prohibited from sharing this information if contractually mandated. |
| New human-generated data sources | What are the top-5 sources (by volume) of new human-generated data for building the model? | We define a source as the entity or means by which the developer acquires data. For example, Scale AI could be a source of new human-generated data. By new, we mean the data is specifically acquired for the purposes of building the model. |
| Instructions for data generation | For each of the top-5 sources of human-generated data, what instructions does the developer provide for data generation? | The instructions should be those provided to the data source. For example, if a third-party vendor works directly with the data laborers to produce the data, the instructions from the developer to this vendor should be disclosed. |

*Continued on next page*

| Indicator | Definition | Notes |
|---|---|---|
| Data laborer practices | For the top-5 sources of human-generated data, how are laborers compensated, where are they located, and what labor protections are in place? | For each data source, we require (i) the compensation in either USD or the local currency, (ii) any countries where at least 25% of the laborers are located, and (iii) a description of any labor protections. We will award this point if the developer discloses that it is not aware of data laborer practices. |
| Synthetic data sources | What are the top-5 sources (by volume) of synthetic data acquired for building the model? | We define a source of synthetic data as a non-human mechanism (e.g. a machine learning model) used to generate the data. |
| Synthetic data purpose | For the top-5 sources of synthetically generated data, what is the primary purpose for data generation? | We define a source of synthetic data as a non-human mechanism (e.g. a machine learning model) used to generate the data. |
| Data processing methods | What are the methods the developer uses to process acquired data to determine the data directly used in building the model? | We will award this point for disclosure of all of the methods used to process acquired data. Data processing refers to any method that substantively changes the content of the data. For example, compression or changing the data file format is generally not in the scope of this indicator. |
| Data processing purpose | For each data processing method, what is its primary purpose? | Data processing refers to any method that substantively changes the content of the data. For example, compression or changing the data file format is generally not in the scope of this indicator. |
| Data processing techniques | For each data processing method, how does the developer implement the method? | Data processing refers to any method that substantively changes the content of the data. For example, compression or changing the data file format is generally not in the scope of this indicator. |
| Data size | Is the size of the data used in building the model disclosed? | To receive this point, the developer should report data size in appropriate units (e.g. bytes, words, tokens, images, frames) and broken down by modality. Data size should be reported to a precision of one significant figure (e.g. 4 trillion tokens, 200 thousand images). The size should reflect data directly used in building the model (i.e. training data) and not data that was acquired but unused, or data used to evaluate the model. |

| Indicator | Definition | Notes |
|---|---|---|
| Data language composition | For all text data used in building the model, what is the composition of languages? | To receive this point, the developer should report (i) all languages which make up at least 1% of the data and their corresponding proportions and (ii) a brief description of how languages are labeled (if a publicly available tool is used, include a link to the tool). Proportions should be reported to a precision of two significant figures and should describe proportions of documents labeled with some langauge. An "Unknown" category may be included to denote documents where the language could not be identified. |
| Data domain composition | For all the data used in building the model, what is the composition of domains covered in the data? | To receive this point, the developer should report the composition of the main domains included in the data used to train the model. This data should be at a level of granularity lower than broad claims about training on "internet data". For example, this could include the proportion of data from e-commerce, social media, news, code, etc. based on the URLs from which the data is sourced. Proportions should be reported to a precision of one significant figure. |
| External data access | Does a third-party have direct access to the data used to build the model? | By a third-party, we mean entities that are financially independent of the developer. We will award this point if at least one such entity is named as having direct access to the data. With that said, we may award this point if the developer provides justifications for prohibiting access to narrowly-scoped parts of the data. |
| Data replicability | Is the data used to build the model described in enough detail to be externally replicable? | We will award this point if the description contains (i) a list of all publicly available training data and where to obtain it and (ii) a list of all training data obtainable from third parties and where to obtain it. These conditions refer to criteria 2 and 3 under the OSI Open Source AI v1.0 definition. |
| Compute usage for final training run | Is the amount of compute used in the model's final training run disclosed? | Compute should be reported in appropriate units, which most often will be floating point operations (FLOPs), along with a description of the measurement methodology, which may involve estimation. Compute should be reported to a precision of one significant figure (e.g. $5 \times 10^{25}$ FLOPs). This number should represent the compute used to train the final model across all model stages. |
| Compute usage including R&D | Is the amount of compute used to build the model, including experiments, disclosed? | Compute should be reported in appropriate units, which most often will be floating point operations (FLOPs), along with a description of the measurement methodology, which may involve estimation. Compute should be reported to a precision of one significant figure (e.g. $7 \times 10^{26}$ FLOPs). Compared to the previous indicator, this indicator should include an estimation of the total compute used across experiments used towards the final training run for the model (such as including hyperparameter optimization or other experiments), and not just the final training run itself. |

| Indicator | Definition | Notes |
|---|---|---|
| Development duration for final training run | Is the amount of time required to build the model disclosed? | The amount of time should be specified in terms of both the continuous duration of time required and the number of hardware hours used. The continuous duration of time required to build the model should be reported in weeks, days, or hours to a precision of one significant figure (e.g. 3 weeks). The number of hardware hours should be reported to a precision of one significant figure and include the type of hardware hours. No form of decomposition into phases of building the model is required for this indicator, but it should be clear what the duration refers to (e.g. training the model, or training and subsequent evaluation and red teaming). |
| Compute hardware for final training run | For the primary hardware used to build the model, is the amount and type of hardware disclosed? | In most cases, this indicator will be satisfied by information regarding the number and type of GPUs or TPUs used to train the model. The number of hardware units should be reported to a precision of one significant figure (e.g. 800 NVIDIA H100 GPUs). We will not award this point if (i) the training hardware generally used by the developer is disclosed, but the specific hardware for the given model is not, or (ii) the training hardware is disclosed, but the amount of hardware is not. We will award this point even if information about the interconnects between hardware units is not disclosed. |
| Compute provider | Is the compute provider disclosed? | For example, the compute provider may be the model developer in the case of a self-owned cluster, a cloud provider like Microsoft Azure, Google Cloud Platform, or Amazon Web Services, or a national supercomputer. In the event that compute is provided by multiple sources or is highly decentralized, we will award this point if a developer makes a reasonable effort to describe the distribution of hardware owners. |
| Energy usage for final training run | Is the amount of energy expended in building the model disclosed? | Energy usage should be reported in appropriate units, which most often will be megawatt-hours (mWh), along with a description of the measurement methodology, which may involve estimation. Energy usage should be reported to a precision of one significant figure (e.g. 500 mWh). No form of decomposition into compute phases is required, but it should be clear whether the reported energy usage is for a single model run or includes additional runs, or hyperparameter tuning, or training other models like reward models, or other steps in the model development process that necessitate energy usage. If the developer is unable to measure or estimate this quantity due to information not being available from another party (e.g. compute provider), we will award this point if the developer explicitly discloses what information it lacks and why it lacks it. |

*Continued on next page*

| Indicator | Definition | Notes |
|---|---|---|
| Carbon emissions for final training run | Is the amount of carbon emitted in building the model disclosed? | Emissions should be reported in appropriate units, which most often will be tons of carbon dioxide emitted (tCO2), along with a description of the measurement methodology, which may involve estimation. Emissions should be reported to a precision of one significant figure (e.g. 500 tCO2). No form of decomposition into compute phases is required, but it should be clear whether the reported emissions is for a single model run or includes additional runs, or hyperparameter tuning, or training other models like reward models, or other steps in the model development process that generate emissions. If the developer is unable to measure or estimate this quantity due to information not being available from another party (e.g. compute provider), we will award this point if the developer explicitly discloses what information it lack and why it lacks it. Emissions should correspond with the energy used in the previous indicator. |
| Water usage for final training run | Is the amount of clean water used in building the model disclosed? | Clean water usage should be in appropriate units, which most often will be megaliters, along with a description of the measurement methodology, which may involve estimation. Clean water usage should be reported to a precision of one significant figure (e.g., 5000ML). No form of decomposition into compute phases is required, but it should be clear whether the reported water usage is for a single model run or includes additional runs, or hyperparameter tuning, or training other models like reward models, or other steps in the model development process that necessitates water usage. If the developer is unable to measure or estimate this quantity due to information not being available from another party (e.g. compute provider), we will award this point if the developer explicitly discloses what information it lacks and why it lacks it. |
| Internal compute allocation | How is compute allocated across the teams building and working to release the model? | To receive a point, the developer should provide the compute allocated to each team involved in training the model. We understand there might be no clear allocation of compute across different teams; in that case, report an estimate of the compute used over the last year. Compute allocation should be reported to at least one significant figure. |
| Model stages | Are all stages in the model development process disclosed? | Stages refer to each identifiable step that constitutes a substantive change to the model during the model building process. We recognize that different developers may use different terminology for these stages, or conceptualize the stages differently. We will award this point if there is a clear and complete description of these stages. |

| Indicator | Definition | Notes |
|---|---|---|
| Model objectives | For all stages that are described, is there a clear description of the associated learning objectives or a clear characterization of the nature of this update to the model? | We recognize that different developers may use different terminology for these stages, or conceptualize the stages differently. We will award this point if there is a clear description of the update to the model related to each stage, whether that is the intent of the stage (e.g. making the model less harmful), a mechanistic characterization (e.g. minimizing a specific loss function), or an empirical assessment (e.g. evaluation results conducted before and after the stage). |
| Code access | Does the developer release code that allows third-parties to train and run the model? | The released code does not need to match the code used internally. |
| Organization chart | How are employees developing and deploying the model organized internally? | To receive a point, the developer should provide both the internal organization chart for the team developing the model as well as the headcounts (or a proportion of headcounts) by the team. |
| Model cost | What is the cost of building the model? | Monetary cost should be reported in appropriate currency (e.g. USD), along with the measurement methodology, which may involve estimation. Cost should be reported to a precision of one significant figure (e.g. 200 million USD). |

| Indicator | Definition | Notes |
|---|---|---|
| Basic model properties | Are all basic model properties disclosed? | Basic model properties include: the input modality, output modality, model size, model components, and model architecture. To receive a point, all model properties should be disclosed. Modalities refer to the types or formats of information that the model can accept as input. Examples of input modalities include text, image, audio, video, tables, graphs. Model components refer to distinct and identifiable parts of the model. We recognize that different developers may use different terminology for model components, or conceptualize components differently. Examples include: (i) For a text-to-image model, components could refer to a text encoder and an image encoder, which may have been trained separately. (ii) For a retrieval-augmented model, components could refer to a separate retriever module. Model size should be reported in appropriate units, which generally is the number of model parameters, broken down by named component. Model size should be reported to a precision of one significant figure (e.g. 500 billion parameters for text encoder, 20 billion parameters for image encoder). Model architecture is the overall structure and organization of a foundation model, which includes the way in which any disclosed components are integrated and how data moves through the model during training or inference. We recognize that different developers may use different terminology for model architecture, or conceptualize the architecture differently; a sufficient disclosure includes any clear, though potentially incomplete, description of the model architecture. |
| Deeper model properties | Is a detailed description of the model architecture disclosed? | To receive a point, the model architecture should be described in enough detail to allow for an external entity to fully implement the model. Publicly available code or a configuration file for a model training library (e.g., GPT-NeoX) would be a sufficiently detailed description. |
| Model dependencies | Is the model(s) the model is derived from disclosed? | We will award this point for a comprehensive disclosure of the model or models on which the foundation model directly depends on or is derived from, as well as the method by which it was derived (e.g., through fine tuning, model merging, or distillation). Additionally, we will award a point if the developer discloses that the model is not dependent on or derived from any model. |
| Benchmarked inference | Is the compute and time required for model inference disclosed for a clearly-specified task on clearly-specified hardware? | The duration should be reported in seconds to a precision of one significant figure (e.g. 0.002 seconds). Compute usage for inference should be reported in FLOPs/second to a precision of one significant figure (e.g. $5 \times 10^{21}$ FLOPs/second). The hardware in this evaluation need not be the hardware the developer uses for inference. The developer can report this figure over some known or public dataset. |

Continued on next page

| Indicator | Definition | Notes |
|---|---|---|
| Researcher credits | Is a protocol for granting external entities API credits for the model disclosed? | A model credit access protocol refers to the steps, requirements, and considerations involved in granting credits to external entities. We will award this point if the developer discloses key details of its protocol, including (i) where external entities can request access to credits (e.g. via an access request form); (ii) explicit criteria for selecting external entities; and (iii) its policy on granting a transparent decision on whether access has been granted within a specified, reasonable period of time. Additionally, we will award a point if the developer discloses that it does not grant external entities API credits. |
| Specialized access | Does the developer disclose if it provides specialized access to the model? | Specialized access could include several categories, such as early access, subsidized access, or deeper access (e.g., to model weights or checkpoints, that are not publicly available). We will award this point if the developer discloses (i) if it provides specialized access and (ii) statistics on the number of users granted access across academia, industry, non-profits, and governments, to one significant figure. |
| Open weights | Are the model's weights openly released? | To receive this point, model weights need to be publicly available at no cost. Developers may receive this point even if there are some restrictions on the external entities that are permitted access (e.g. geographic restrictions), insofar as these restrictions are transparent (e.g. via a license or some high-level description of who has been granted access to the foundation model). |
| Agent Protocols | Are the agent protocols supported for the model disclosed? | Agent protocols are specifications that define how autonomous agents exchange messages, context, or function calls with other agents, tools, or services (e.g., Anthropic's Model Context Protocol (MCP) and Google's Agent-to-Agent (A2A) spec). To earn this point, documentation must enumerate each protocol and describe any deviations or proprietary extensions. |
| Capabilities taxonomy | Are the specific capabilities or tasks that were optimized for during post-training disclosed? | Capabilities refer to the specific and distinctive functions that the model can perform. We recognize that different developers may use different terminology for capabilities, or conceptualize capabilities differently. We will award this point for a list of capabilities specifically optimized for in the post-training phase of the model, even if some of the capabilities are not reflected in the final model. |
| Capabilities evaluation | Does the developer evaluate the model's capabilities prior to its release and disclose them concurrent with release? | The evaluations must contain precise quantifications of the model's behavior in relation to the capabilities specified in the capabilities taxonomy. We will award this point for any clear, but potentially incomplete, evaluation of multiple capabilities. |

*Continued on next page*

| Indicator | Definition | Notes |
|---|---|---|
| External reproducibility of capabilities evaluation | Are code and prompts that allow for an external reproduction of the evaluation of model capabilities disclosed? | The released code and prompts need not be the same as what is used internally, but should allow the developer's results on all capability evaluations to be reproduced. The released code must be open source, following the OSI definition of open source. |
| Train-test overlap | Does the developer measure and disclose the overlap between the training set and the dataset used to evaluate model capabilities? | We will award this point if, with every capability evaluation for which the developer reports results, the developer reports the overlap between the training set of the model and the dataset used for evaluation, as well as the general methodology for computing train-test overlap (e.g. a description of how n-gram matching was used). |
| Risks taxonomy | Are the risks considered when developing the model disclosed? | Risks refer to possible negative consequences or undesirable outcomes that can arise from the model's deployment and usage. These consequences or outcomes may arise from model limitations (functions that the model cannot perform) or issues with the model's trustworthiness (e.g., its lack of robustness, reliability, calibration). We recognize that different developers may use different terminology for risks, or conceptualize risks differently. We will award this point for a complete list of risks considered, even if some of the risks are not reflected in the final model. |
| Risks evaluation | Does the developer evaluate the model's risks prior to its release and disclose them concurrent with release? | The evaluations must contain precise quantifications of the model's behavior in relation to the risks specified in the risk taxonomy. We will award this point for clear evaluations of the majority of the states risks. |
| External reproducibility of risks evaluation | Are code and prompts to allow for an external reproduction of the evaluation of model risks disclosed? | The released code and prompts need not be the same as what is used internally, but should allow the developer's results on all risk evaluations to be reproduced. The released code must be open-source, following the OSI definition of open-source. |
| Pre-deployment risk evaluation | Are the external entities have evaluated the model pre-deployment disclosed? | By external entities, we mean entities that are significantly or fully independent of the developer. We will award this point if the developer specifies the entity that carried out the pre-deployment analysis, discloses the terms of the analysis (such as conditions for releasing the evaluation results or the developer's control over the final results), as well as any financial transaction between the parties. We will award this point if the developer discloses no external entities have evaluated the model pre-deployment, or discloses only terms of the analysis where it is not bound by NDA while still naming all external entities. |

*Continued on next page*

| Indicator | Definition | Notes |
| --- | --- | --- |
| External risk evaluation | Are the parties contracted to evaluated model risks disclosed? | We will award this point if the developer discloses statistics regarding all contracted parties that are responsible for evaluating risks (not limited to external entities or pre-deployment evaluation). This includes the number of contracted for-profit or non-profit entities, government entities, independent contractors, and researchers contracted by the developer to evaluate risks. We will award this point if the developer discloses it has no such contracts. |
| Mitigations taxonomy | Are the post-training mitigations implemented when developing the model disclosed? | By post-training mitigations, we refer to interventions implemented by the developer during the post-training phase to reduce the likelihood and/or the severity of the model's risks. We recognize that different developers may use different terminology for mitigations, or conceptualize mitigations differently. We will award this point for a complete list of mitigations considered, even if some of the mitigations are not reflected in the final model. Alternatively, we will award this point if the developer reports that it does not mitigate risk in this way. |
| Mitigations taxonomy mapped to risk taxonomy | Does the developer disclose how the post-training mitigations map onto the taxonomy of risks? | We will award this point for a complete mapping of the primary risk that each mitigation is meant to address, even if the mitigation potentially maps on to other risks in the taxonomy. Alternatively, we will award this point if the developer reports that it does not mitigate risk. |
| Mitigations efficacy | Does the developer evaluate and disclose the impact of post-training mitigations? | We will award this point if the developer discloses the results on the risk evaluations before and after the post-training mitigations are applied. Alternatively, we will award this point if the developer reports that it does not mitigate risk in this way. |
| External reproducibility of mitigations evaluation | Are code and prompts to allow for an external reproduction of the evaluation of post-training mitigations disclosed? | The released code and prompts need not be the same as what is used internally, but should allow the developer's results on all mitigations evaluations to be reproduced. The released code must be open-source, following the OSI definition of open-source. Alternatively, we will award this point if the developer reports that it does not mitigate risk. |
| Model theft prevention measures | Does the developer disclose the security measures used to prevent unauthorized copying ("theft") or unauthorized public release of the model weights? | This indicator assesses the developer's disclosures regarding how it addresses the risk that malicious actors or insiders could exfiltrate or replicate proprietary weights. Security measures could include insider threat analysis and detection, in addition to external threat management. Examples of such measures include encryption at rest, key management, remote attestation, or auditing for suspicious queries. We will award a point if the developer discloses specific steps taken to safeguard the model weights or that none are implemented. |

| Indicator | Definition | Notes |
|---|---|---|
| Release stages | Are the stages of the model's release disclosed? | Release stages include A/B testing, release on a user-facing product, GA release, open-weight release, etc. We recognize that the release of a foundation model falls along a spectrum, with many forms of partial release, and that different developers may conceptualize release differently. We will award a point if the developer provides a clear identification of the stages through which the model was released. |
| Risk thresholds | Are risk thresholds disclosed? | Risk thresholds determine when a risk level is unacceptably high to a developer (e.g. leading to the decision to not release a model), moderately high (e.g. triggering additional safety screening), or low enough to permit normal usage. We will award this point if the developer discloses explicit risk thresholds that clarify (i) which harmful outcomes are being scored, (ii) how the scores are computed (in general terms, not necessarily disclosing internal algorithms), and (iii) what triggers an action to block, delay, or otherwise modify a model's release. Alternatively, we will award a point if the developer discloses that it does not consider explicit risk thresholds during model release. |
| Versioning protocol | Is there a disclosed protocol for versioning and deprecation of the model? | We will award a point if the developer discloses how model versions are labeled, updated, deprecated, and communicated to users. |
| Change log | Is there a disclosed change log for the model? | We will award a point if the developer publishes a version-by-version record of new features, fixes, or performance improvements. |
| Foundation model roadmap | Is a forward-looking roadmap for upcoming models, features, or products disclosed? | A foundation model roadmap is a transparent statement about how the developer intends to evolve or expand its LLM offerings, including upcoming models, major feature releases, or expanded products based on the model, along with approximate timelines or version milestones. It can be high-level (e.g., "new model Q2 2025"), but must exist publicly. |

*Continued on next page*

| Indicator | Definition | Notes |
|---|---|---|
| Top distribution channels | Are the top-5 distribution channels for the model disclosed? | We define distribution channels to be either an API provider (a pathway by which users can query the model with inputs and receive outputs) or a model distributor (a pathway by which model weights are released). We recognize that distribution channels may arise without the knowledge of the model developer. For example, the weights of a model may be released through one distribution channel and then be distributed through other channels. Distribution channels can be ranked by any reasonable metric (e.g., number of queries, number of downloads, number of users, revenue). A description of the metric should be provided. API providers and model distributors may be ranked separately using different metrics as long as the total number of distribution channels equals five (if five distribution channels exist). For example, the developer may choose to disclose the top-3 API providers (ranked by the number of queries) and the top-2 model distributors (ranked by the number of downloads). |
| Quantization | Is the quantization of the model served to customers in the top-5 distribution channels disclosed? | We will award this point for a disclosure of the model precision in each of the top-5 distribution channels. |
| Terms of use | Are the terms of use of the model disclosed? | We define terms of use to include terms of service and model licenses. We will award this point for a pointer to the terms of service or model license. In the event that model's licenses are written more generally, it should be clear which assets they apply to. We recognize that different developers may adopt different business models and therefore have different types of model licenses. Examples of model licenses include responsible AI licenses, open-source licenses, and licenses that allow for commercial use. Terms of service should be disclosed for each of the top-5 distribution channels. However, we will award this point if there are terms-of-service that appear to apply to the bulk of the model's distribution channels. |

*Continued on next page*

| Indicator | Definition | Notes |
| --- | --- | --- |
| Distribution channels with usage data | What are the top-5 distribution channels for which the developer has usage data? | We define distribution channels to be either an API provider (a pathway by which users can query the model with inputs and receive outputs) or a model distributor (a pathway by which model weights are released). We recognize that distribution channels may arise without the knowledge of the model developer. For example, the weights of a model may be released through one distribution channel and then be distributed through other channels. Distribution channels can be ranked by any reasonable metric (e.g., number of queries, number of downloads, number of users, revenue). A description of the metric should be provided. We define usage data as any form of developer-exclusive data collected from any of a developer's distribution channel. A developer has access to usage data from a distribution channel if it is able to use that data for downstream purposes (e.g., analytics, training etc.). Usage data may be shared outside of the developer, but it is initially collected by the distribution channel and shared to the developer. |
| Amount of usage | For each of the top-5 distribution channels, how much usage is there? | Usage should be reported as the number of queries over the span of a month, reported to the precision of one significant figure (e.g., 50 million queries). |
| Classification of usage data | Is a representative, anonymized dataset classifying queries into usage categories disclosed? | Developers may either share a fully public dataset or a partially restricted dataset (e.g., under a research license). We will award this point if there is a clear, aggregated or sample dataset that reveals categories of tasks/queries. |
| Data retention and deletion policy | Is a policy for data retention and deletion disclosed? | A data retention and deletion policy is a policy for removing particular data from the training set and/or preventing it from being used if there is a user or external request (e.g., "right to be forgotten") that also covers internal data governance. This includes whether there is a formal process to delete or retract data from future training runs and how long raw data is retained. It also clarifies how quickly deletions propagate to the model (e.g., "only in subsequent major model releases"). |
| Geographic statistics | Across all forms of downstream use, are statistics of model usage across geographies disclosed? | We will award this point if there is a meaningful, though potentially incomplete or vague, disclosure of geographic usage statistics at the country-level. |
| Internal products and services | What are the top-5 internal products or services using the model? | An internal product or service is a product or service built by the developer. Products or services can be ranked by any reasonable metric (e.g., number of users, queries, revenue). A description of the metric should be provided. |

*Continued on next page*

| Indicator | Definition | Notes |
|---|---|---|
| External products and services | What are the top-5 external products or services using the model? | An external product or service is a product or service built by a party external to the developer. Products or services can be ranked by any reasonable metric (e.g., number of users, queries, revenue). A description of the metric should be provided. We will award a point if the developer discloses that that it does not have access to such metrics about external products or services. |
| Users of internal products and services | How many monthly active users are there for each of the top-5 internal products or services using the model? | An internal product or service is a product or service built by the developer. The number of users refers to users who engaged or interacted with the model through the internal product or service over the last month or averaged over the last X months (this should be specified). Number of users should be specified to one significant figure (e.g. 100,000). |
| Consumer / enterprise usage | Across all distribution channels for which the developer has usage data, what portion of usage is consumer versus enterprise? | Consumer usage refers to usage by individual consumers. Enterprise usage refers to usage by enterprise customers (including government use). Consumer and enterprise usage should be calculated in terms of the number of queries by or the amount of revenue from consumer or enterprise users. Percentages should be specified to two significant digits (e.g., 12% consumer, 88% enterprise). |
| Enterprise users | Across all distribution channels for which the developer has usage data, what are the top-5 enterprises that use the model? | Enterprises should be ranked by the number of queries made or the amount of revenue from usage since the model's release. We will also award this point if the developer indicates it does not have access to enterprise usage data. |
| Government use | What are the 5 largest government contracts for use of the model? | This includes known government contracts of enterprise or government-specific products and services that use the model. We will award this point if the developer discloses its top five government contracts ranked monetary value, though the developer may omit contracts where it is under NDA regarding the existence of the contract. |
| Benefits Assessment | Is an assessment of the benefits of deploying the model disclosed? | We will award this point for any quantitative assessment of the benefits or potential benefits of deploying the model. |
| AI bug bounty | Does the developer operate a public bug bounty or vulnerability reward program under which the model is in scope? | We will award this point for a publicly documented bug bounty or vulnerability reward program describing (i) in-scope vulnerabilities (e.g., prompt bypasses, data leaks), (ii) out-of-scope items, (iii) submission process, and (iv) reward tiers or recognition if applicable. We will award a point if the developer discloses it has no AI bug bounty that encourages external researchers to report security, privacy, or adversarial vulnerabilities in the model. |

*Continued on next page*

| Indicator | Definition | Notes |
|---|---|---|
| Responsible disclosure policy | Does the developer clearly define a process by which external parties can disclose model vulnerabilities or flaws? | We will award this point for a description of the process external parties can use for responsbly disclosing model vulnerabilities and flaws, which should include (i) what mechanism external parties can use to disclose vulnerabilities or flaws (e.g., a form, an email) and (ii) what process follows a disclosure (e.g., how much time must parties wait until public release). This is often included with a bug bounty, but can also be standalone. We will award a point if the developer discloses it has no responsible disclosure policy. |
| Safe harbor | Does the developer disclose its policy for legal action against external evaluators conducting good-faith research? | We will award this point if the developer discloses whether it has a policy committing it to not pursue legal action against external evaluators conducting good-faith research. This should not be only for software security vulnerabilities, but also AI flaws, and it should be based on researcher conduct standards, not at the sole discretion of the company. We will award this point if the developer provides a clear description of its policy regarding such protections for external researchers, or lack thereof. |
| Security incident reporting protocol | Are major security incidents involving the model disclosed? | A security incident reporting protocol provides post-deployment transparency about serious incidents or breaches. Security incidents refer to incidents where external security threats affect the model (e.g., data breaches or DDoS attacks on the service). We will award this point if the developer states (i) how to submit a security incident report, (ii) how quickly it will respond, and (iii) when and whether results are disclosed. Every incident need not be reported publicly, but the developer must disclose a policy determining how incidents are reported and disclosed. |
| Misuse incident reporting protocol | Are misuse incidents involving the model disclosed? | A misuse incident reporting protocol provides post-deployment transparency about incidents of misuse involving the model. As opposed to the previous indicator, this indicator is about actors misusing the model to cause real-world harm, such as misinformation operations or cybersecurity attacks. We will award this point if the developer states (i) how to submit a misuse incident report, (ii) how quickly it will respond, and (iii) when and whether results are disclosed. Every incident need not be reported publicly, but there needs to be a policy governing how incidents are reported. |
| Post-deployment coordination with government | Does the developer coordinate evaluation with government bodies? | We will award this point if the developer specifies which government bodies it is coordinating with and for what types of post-deployment evaluations. Government bodies include AI Safety Institutes, national security agencies, national labs, and international governmental enties such as UN agencies or the G7. Evaluation here may also include sharing of the developer's proprietary evaluation results for help with interpretation. |

*Continued on next page*

| Indicator | Definition | Notes |
|---|---|---|
| Feedback mechanisms | Does the developer disclose a way to submit user feedback? If so, is a summary of major categories of feedback disclosed? | We will award this point if the developer (i) discloses how users can submit feedback (e.g., via a form or a thumbs up/thumbs down for model responses) and (ii) discloses aggregated or categorized feedback data (e.g. a categorization of thumbs up and thumbs down data). |
| Permitted, restricted, and prohibited model behaviors | Are model behaviors that are permitted, restricted, and prohibited disclosed? | We refer to a policy that includes this information as a model behavior policy, or a developer's policy on what the foundation model can and cannot do (e.g. such a policy may prohibit a model from responding to NSFW content). We recognize that different developers may adopt different business models and that some business models may make enforcement of a model behavior policy more or less feasible. We will award this point if at least two of the three categories (i.e. permitted, restricted, and prohibited model behaviors) are disclosed. Alternatively, we will award this point if the developer reports that it does not impose any restrictions on its model's behavior in this way. |
| Model response characteristics | Are desired model response characteristics disclosed? | Model response characteristics include default behaviors or behaviors that the developer steers the model to take. These may include being helpful, taking an objective point of view, or using tools only when necessary. We will award points for a clear description of desired model response characteristics or a statement that there are no such characteristics. |
| System prompt | Is the default system prompt for at least one distribution channel disclosed? | A system prompt is defined as the prompt provided to the system by default that guides the system's behavior. We will award this point for the disclosure of the verbatim text of the full system prompt as well as an explanation for the context in which the system prompt is used. |
| Intermediate tokens | Are intermediate tokens used to generate model outputs available to end users? | Intermediate tokens are defined as any tokens generated by the model before the final output is shown to the user, such as model chains of thought. We will also award this point if a summary of intermediate tokens is made available to end users. If intermediate tokens or summaries are not made available, the developer should provide a justification. |
| Internal product and service mitigations | For internal products or services using the model, are downstream mitigations against adversarial attacks disclosed? | An internal product or service is a product or service built by the developer. Adversarial attacks include prompt injection, jailbreaking, or malicious queries. Mitigations against adversarial attacks might include specialized prompt filtering, content scanning, or real-time monitoring of queries or accounts. We will award this point if the developer discloses a clear statement of methods used (e.g., a specialized prompt sanitizer or adversarial pattern detector), or if the developer states it does not implement such product-level mitigations against adversarial attacks. |

*Continued on next page*

| Indicator | Definition | Notes |
|---|---|---|
| External developer mitigations | Does the developer provide built-in or recommended mitigations against adversarial attacks for downstream developers? | Downstream developers are developers who access the model through a distribution channel. Adversarial attacks include prompt injection, jailbreaking, or malicious queries. Mitigations against adversarial attacks that developers might build in or recommend include content filtering endpoints and recommended prompt templates. We will award this point if the developer discloses (i) technical mitigations (e.g., a developer provided moderation API or classifier) it offers or implements, (ii) recommended best practices or libraries for downstream developers, or (iii) an explicit statement that it does not build or recommend any particular downstream mitigations in this way.. |
| Enterprise mitigations | Does the developer disclose additional or specialized mitigations for enterprise users? | Enterprise users are, for example, large organizations with dedicated service agreements or users of enterprise-specific API deployments or products and services. Additional or specialized mitigations may address enterprise needs such as data privacy controls, advanced prompt/response monitoring, or compliance checks with regulations such as GDPR or HIPAA. Additional or specialized mitigations may include single-tenant deployments, custom filters for specific regulated industries, or advanced logging for compliance. We will award a point if the developer at least describes these mitigations or states that it does not provide such additional or specialized enterprise mitigations. |
| Detection of machine-generated content | Are mechanisms that are used for detecting content generated by this model disclosed? | A mechanism for detecting machine-generated content might include storing a copy of all outputs generated by the model to compare against, implementing a watermark on model outputs, adding cryptographic metadata (such as C2PA), or training a detector post-hoc to identify such content. We will award this point if any such mechanism is disclosed or if the developer reports that it does not have or use any such mechanism. |
| Documentation for responsible use | Does the developer provide documentation for responsible use by downstream developers? | To receive a point, the developer should provide documentation for responsible use. This might include details on how to adjust API settings to promote responsible use, descriptions of how to implement mitigations, or guidelines for responsible use. We will also award this point if the developer states that it does not provide any such documentation. For example, the developer might state that the model is offered as is and downstream developers are accountable for using the model responsibly. |
| Permitted and prohibited users | Is a description of who can and cannot use the model on the top-5 distribution channels disclosed? | We will award this point for a description of the company's policies for permitted and prohibitted users on its top-5 distribution channels. We will award this point if the developer has a more general acceptable use policy that it confirms applies across these distribution channels. We will award this point if there are no restrictions on users. |

*Continued on next page*

| Indicator | Definition | Notes |
|---|---|---|
| Permitted, restricted, and prohibited uses | Which uses are explicitly allowed, conditionally permitted, or strictly disallowed under the acceptable use policy for the top-5 distribution channels? | We will award this point for a rough characterization of two or more of permitted, restricted, and prohibited uses across the top-5 distribution channels. We will award this point if the developer has a more general acceptable use policy that it confirms applies across these distribution channels. We will award this point if there are no restrictions on users. |
| AUP enforcement process | What are the methods used by the developer to enforce the acceptable policy? | We will award this point if the developer discloses the processes (automated or manual) it uses to detect, review, and respond to potential acceptable use policy violations. We will award this point for a reasonable best-effort attempt to provide the bulk of this information, though one line indicating the developer reserves the right to terminate accounts is insufficient. Alternatively, we will award this point if the developer reports that it does not use such methods to enforce its acceptable use policy. |
| AUP enforcement frequency | Are statistics on the developer's AUP enforcement disclosed? | We will award this point if the developer discloses enforcement statistics (e.g., violation counts or actions taken) from its enforcement of its acceptable use policy. Alternatively, we will award this point if the developer reports that it does not enforce its acceptable use policy. |
| Regional policy variations | Are differences in the developer's acceptable use or model behavior policy across geographic regions disclosed? | We will award this point if the developer discloses distinctions in its AUP or MBP and provides examples of differences in multiple specific regions, or states that no differences exist. For example, some jurisdictions impose content restrictions beyond those in the developer's global policy that may necessesitate local deviations. |
| Oversight mechanism | Does the developer have an internal or external body that reviews core issues regarding the model prior to deployment? | We will award this point if the developer discloses that is has such an internal or external body and provides some description of its scope, or alternatively if the developer discloses that it has no such body. An oversight mechanism covers governance structure beyond mere external risk evaluation, asking whether a formal body regularly reviews design and deployment decisions. Core issues may include model objectives, data usage, or risk mitigation. |
| Whistleblower protection | Does the developer disclose a whistleblower protection policy? | We will award this point if the developer discloses (i) the existence of a whistleblower protection policy, (ii) what protections are afforded to whistleblowers, (iii) how reports are handled and investigated, and (iv) any external oversight of the whistleblower protection process. This might include protections for whistleblowers who report safety, ethical, or legal concerns related to the model. We will also award this point if the developer discloses that it has no such policy. |

*Continued on next page*

| Indicator | Definition | Notes |
|---|---|---|
| Government commitments | What commitments has the developer made to government bodies? | We will award this point if the company provides an exhaustive list of commitments it has made to government bodies in the jurisdictions where it offers its model. |

