# OpenReview forum: "The 2025 Foundation Model Transparency Index"
_TMLR — Decision pending for TMLR_

### Review · Reviewer_6RpH · 2026-01-29

**Summary Of Contributions:**

The paper presents the 2025 edition of the Foundation Model Transparency Index. This is the third edition of the proposed index and the paper presents the different metrics used to calculate the index, the data collection and scoring process and the obtained results. Overall, this is a very interesting contribution that addresses the increasing need for transparency by AI companies. The presentation of the index, the methodological details and the derived conclusions are excellent and provide a very useful snapshot of transparency practices in a big part of the AI industry.

At the same time, I am also concerned by a number of methodological issues associated with the data collection and scoring of the index, as well as with the fit of this type of work to TMLR. In summary, my concerns include the following:

- The initial selection of companies to test/score may miss some important players.
- The list of metrics is likely too extensive, and while this is positive in terms of completeness, it may lead to an increasing amount of effort needed in order to meet the transparency requirements, which may favour companies that have resources to dedicate to this activity vs others, a kind of measurement bias that could be avoided by simplifying the reporting requirements, focusing on the most important ones.
- It appears that all metrics are considered equally important and are scored as either 1 or 0. Both of these decisions may actually lead to a very unrepresentative scoring of transparency. Adopting a weighting the reflects the importance of different factors, and also scoring the transparency in a more granular way would likely lead to a more meaningful index.
- While the authoring team spent a lot of effort in data collection and analysis, it is not clear whether the collected transparency information is valid and meaningful, i.e. companies may have provided responses, but it is not clear whether these were verified as accurate and meaningful.
- While I appreciate the engagement of companies as a way to build trust, I believe that this could potentially lead to very distorted results. Normally, transparency information should be publicly accessible to everyone without the need to interact with the companies. The described process appears to be a kind of "negotiated" or "priviledged" transparency, favouring companies that engaged with the authorship team, while penalizing the ones that didn't.
- While the authors present some longitudinal analysis results, their validity is highly questionable given the changes in the index metrics and the underlying practices of data collection. I feel that the comparison across the three years should be presented without making any claims about transparency trends.
- I also have doubts with respect to the fit of this type of work to TMLR. While there is clearly a potential contribution to transparency practices of AI companies, this seems much more appropriate for a policy or multidisciplinary journal, rather than a journal with a mostly technical focus.

**Additional Comments:**

Beyond my concerns on matters related to the methodology of the index, I would like to express my concerns regarding the anonymity of the submission. FMTI, being in its third year, is already very well known across the AI community and even to the general tech-savvy public. That means that even without searching on the web, I already knew the authorship team behind the submission before reviewing the paper. In my opinion, this breaks the double blind review requirement, but I understand that this is not the fault of the authors and that is why I proceeded with my review as if the submission was anonymous.

**Audience:**

Yes

**Audience Explanation:**

I feel that the general public and especially people of diverse backgrounds interested in technological developments as well as policy stakeholders would be much more interested in the contents of this paper compared to the TMLR audience. The computation of the index is very hard to reproduce, primarily due to the fact that it involves a strong interaction step with the AI companies, which is both very effort-intensive and could lead to very different results depending on who attempts to engage with the companies and via what channels.

**Broader Impact Concerns:**

The authors do not sufficiently address concerns in relation to the following aspects of the FMTI:
- how the direct engagement of the measured companies in the scoring process could unfairly affect the results of the index;
- how the existence of the index could create incentives for companies to disclose "just enough" information to hit a good score without actually achieving meaningful transparency
- how the initial company selection process might lead to selection biases and an incomplete picture of the AI transparency landscape.

**Claims And Evidence:**

No

**Claims Explanation:**

Although there are no explicit claims made by the submission, it appears that the two main contributions are the establishment of a standardized way for measuring the transparency of AI companies and the potential of using this index to track the performance (in terms of transparency) of AI companies over time. While these two objectives are very appealing and very much needed, I am afraid that they are not well supported due to the design choices made by the FMTI team, namely:
- Deciding to engage companies in the FMTI scoring process is in my view a very big methodological concern that creates a host of issues: a) affects the final scoring depending on the will or availability or ability of companies to engage with the authorship team; b) introduces a subjective element in the scoring, where the relation built between the company and the authors could affect the scoring; c) prevents the reproduction of the index by third parties; d) in the long run, and assuming the FMTI becomes very popular, creates incentives for companies to provide "just enough" information to score high without necessarily providing meaningful transparency.
- While the proposed index is very comprehensive, this adds a lot of complexity and manual effort in the data collection and scoring process. Furthermore, the index requires extensive human interpretation and despite the high inter-rater agreement (a bit below than 90%), this is still a factor that compromises its objectivity and accuracy.
- The fact the index metrics and scoring processes changed over time means that there are very limited insights and conclusions that can be drawn with respect to temporal AI transparency trends. Given also the above concerns, I am afraid that the consistent measurement of the index over time is unlikely to be feasible.
- The selection of companies to be included in the Index is not sufficiently justified and very important players such as NVIDIA, MoonshotAI, Baidu, Sakana, etc. are missing. Declining to participate in the index as a reason for this means that the index is already suffering from a very important selection bias and that this is an inherent weakness.

**Requested Changes:**

I would consider the following changes as critical for the validity of the submission:
- Offering a more lightweight version of the index that is as automated as possible to compute and relies only on publicly available information and not on direct interaction with companies. Furthermore, this should include all major AI companies and not only those that agree to participate.
- Experimenting with a more granular scoring that captures the extent of transparency per metric instead of having a binary score. Experimenting also with different weighting schemes based on perceived importance of scores in order to show how the results and conclusions change based on such choices.

---

> ### Author Response · Authors · 2026-03-06
> **Review Response**
>
> We appreciate that the reviewer finds the presentation, methodological details and derived conclusions of our work to be excellent.
>
> > The list of metrics is likely too extensive, and while this is positive in terms of completeness, it may lead to an increasing amount of effort needed in order to meet the transparency requirements, which may favour companies that have resources to dedicate to this activity vs others, a kind of measurement bias that could be avoided by simplifying the reporting requirements, focusing on the most important ones.
>
> Although simplifying the reporting requirements would reduce the cost to produce reports, it would not be possible to produce a single set of “most important” indicators. Rather, different dimensions of transparency are important to different audiences. For example, information on capabilities and quantization may be important to downstream developers but less important to policymakers (who may care more about e.g., oversight/whistleblower mechanisms).
>
> We also note that the second and third highest scoring companies are both start-ups rather than large tech companies.
>
> > It appears that all metrics are considered equally important and are scored as either 1 or 0. Both of these decisions may actually lead to a very unrepresentative scoring of transparency. Adopting a weighting the reflects the importance of different factors, and also scoring the transparency in a more granular way would likely lead to a more meaningful index.
>
> As discussed previously, different audiences may find different sets of indicators important: so, although adopting a weighting may result in more representative scores for some audiences, it would lead to a less representative score for others.
>
> Adopting more granular scoring was considered, but it adds a significant amount of complexity (both in scoring and in the indicator definitions themselves). As discussed in the paper, a key purpose of FMTI is as a way to clarify the currently imprecise and nebulous concept of transparency in relation to foundation model developers. Binary indicators define a clearer threshold for what should and should not be disclosed.
>
> > While the authoring team spent a lot of effort in data collection and analysis, it is not clear whether the collected transparency information is valid and meaningful, i.e. companies may have provided responses, but it is not clear whether these were verified as accurate and meaningful.
>
> In the Appendix, we add the complete criteria for each of the indicators given to the model developers and used for scoring. A key change from the previous year’s index was to improve the precision of the requirements such that any disclosed information would also be meaningful (e.g., the external reproducibility requirement requires the disclosure of the code & prompts used).
>
> We recognize that we can not guarantee the accuracy of the responses (just like we can not guarantee the accuracy of any public disclosure made by a company), but limiting the disclosures to only information that’s verifiable would drastically limit the scope of what can be included. For example, it would be infeasible to score companies on third-party auditing.
>
> > While the authors present some longitudinal analysis results, their validity is highly questionable given the changes in the index metrics and the underlying practices of data collection. I feel that the comparison across the three years should be presented without making any claims about transparency trends.
>
> We agree that changes in scoring methodology requires a more careful consideration of longitudinal analyses. In our analysis, we anchor our comparisons against aspects of our index that stays consistent: in Section 5.2, we perform our comparisons across a fixed set of 9 developers; in Table 3 we consider regressions across indicators that are consistent across time; and, finally, from Page 35-38, we conduct a qualitative analysis of changes where we can similarly control for changes in indicators when analyzing how disclosures change.
>
> > I also have doubts with respect to the fit of this type of work to TMLR. While there is clearly a potential contribution to transparency practices of AI companies, this seems much more appropriate for a policy or multidisciplinary journal, rather than a journal with a mostly technical focus.
>
> We would like to mention that the previous two editions of FMTI have also been accepted into TMLR.

---

> ### Author Response · Authors · 2026-03-06
> **Review Response (pt. 2)**
>
> > Deciding to engage companies in the FMTI scoring process is in my view a very big methodological concern that creates a host of issues: a) affects the final scoring depending on the will or availability or ability of companies to engage with the authorship team; b) introduces a subjective element in the scoring, where the relation built between the company and the authors could affect the scoring; c) prevents the reproduction of the index by third parties; d) in the long run, and assuming the FMTI becomes very popular, creates incentives for companies to provide "just enough" information to score high without necessarily providing meaningful transparency.
>
> > While I appreciate the engagement of companies as a way to build trust, I believe that this could potentially lead to very distorted results. Normally, transparency information should be publicly accessible to everyone without the need to interact with the companies. The described process appears to be a kind of "negotiated" or "privileged" transparency, favouring companies that engaged with the authorship team, while penalizing the ones that didn't.
>
> Although using only public information would allow for more consistent scoring, engaging directly with companies allows us to improve transparency through the scoring process. This has been shown in the previous edition of FMTI, and we also show this to be the case in this year’s edition of FMTI (with scores increasing, on average, by 9.71 points).
> We would also like to emphasize that although the final index was produced through negotiations with companies, the disclosures themselves are fully publicly accessible.
>
> > The fact the index metrics and scoring processes changed over time means that there are very limited insights and conclusions that can be drawn with respect to temporal AI transparency trends. Given also the above concerns, I am afraid that the consistent measurement of the index over time is unlikely to be feasible.
>
> First, we disagree that keeping the index static necessarily allows for a clean temporal analysis. Over time, different aspects of model development and deployment become more and less important; the indicator regarding whether “intermediate tokens” are disclosed, for example, would not make sense in 2023. Retaining the same index over three years would simply mean that we would be analyzing how the 2023-notion of transparency changes.
>
> Keeping the index and scoring process static would also make it impossible to improve how we score transparency. Previous editions inform ways to ensure that the disclosed information is meaningful (e.g., by raising the bar on indicators) or ways to ensure that more information is disclosed (e.g., by engaging companies directly). We choose to prioritize the quality of the disclosed information over the marginal benefit of consistent scoring.
>
> Finally, as discussed above, we account for inconsistencies in methodology across years by anchoring our comparisons against aspects of our index that stays consistent (e.g., a fixed set of developers, a fixed set of indicators, qualitative changes beyond scores).

---

> ### Author Response · Authors · 2026-03-06
> **Review Response (pt. 3)**
>
> > The initial selection of companies to test/score may miss some important players.
>
> > The selection of companies to be included in the Index is not sufficiently justified and very important players such as NVIDIA, MoonshotAI, Baidu, Sakana, etc. are missing. Declining to participate in the index as a reason for this means that the index is already suffering from a very important selection bias and that this is an inherent weakness.
>
> > While the proposed index is very comprehensive, this adds a lot of complexity and manual effort in the data collection and scoring process. Furthermore, the index requires extensive human interpretation and despite the high inter-rater agreement (a bit below than 90%), this is still a factor that compromises its objectivity and accuracy.
>
> We discuss our developer selection process in Section 4.1. We initially contacted NVIDIA and Baidu (along with Zhipu AI, Stability AI, 01.AI, Adobe, Apple, Cohere, and Microsoft), but they ultimately declined to participate. Although we agree that including these companies would strengthen our results, the number of companies we can include is limited by the capacity of our team to produce reports.
>
> We agree that the index does require substantial manual effort. However, we argue that this manual effort is worth it to ensure the quality of the information produced. We prioritize precise scoring criteria (see the added Appendix) to ensure that meaningful information is disclosed over criteria that’s straightforward to produce. The process of engaging with companies also leads to more information disclosed as demonstrated both this year and in the previous edition of FMTI. Also, as discussed previously, transparency is too broad to be able to simplify it down to a single small set of “important indicators”. Additionally, we release the all disclosures, justifications, and scores publicly: companies are free to contest the scores to correct any inaccuracies (or they have already been contested & adjudicated through the engagement process).
>
> It is also not unambiguous what identifies the “most important” developers. We instead opt to consider a diverse set of flagship foundation models, spanning modalities, geographies, and company types.
>
> Finally, we do explore automating parts of the scoring process through our agents architecture (Section 4.2 and Appendix A). Although we find that this does uncover new information, expert effort is still needed to evaluate which of the outputs are correct.

---

> > ### Comment · Reviewer_6RpH · 2026-03-11
> > **Still having some major concerns**
> >
> > I appreciate the thoughtful responses provided by the authors. While I understand the rationale and motivation of the responses, I am afraid that some of my main concerns persist:
> > - Yes, engaging with companies has arguably led to improved transparency practices. However, these happened only for the companies that agreed to participate. This is of course commendable; on the other hand, this has not happened for those companies that did not engage with the FMTI team, so it is not a sufficient reason for adopting the engagement-based approach to creating the FMTI.
> > - The authors recognize that it is not feasible to generate FMTI reports for all notable AI model creators due to the required effort (which is understandable). This means that even if all AI model creators would agree to participate, maybe it would be infeasible to generate the FMTI. This limits the applicability of FMTI. For the same reason, FMTI cannot be made more granular than binary.
> > - Since the authors have already automated a large part of their FMTI extraction process, I still think it would be extremely important to create a "low-effort" reproducible version of FMTI that could be implemented by any researcher solely based on publicly available information.

---

### Review · Reviewer_UBav · 2026-02-13

**Summary Of Contributions:**

This paper introduces the 2025 Foundation Model Transparency Index. Through intensive work including manual labor, the paper comes up with a method that generates a single transparency number for any foundation model. It further reports that FMTI has decreased since 2024. The method uses various indicators about training sets, etc and combines them into a single number using a weighting scheme.

**Audience:**

Yes

**Audience Explanation:**

A transparency index about foundation models can be helpful in monitoring these models and creating incentives for the companies to be more transparent. It may also help the public to choose between the models. Given the widespread use of foundation models in the society, in my view, such transparency index may be helpful in both research and practice.

**Broader Impact Concerns:**

I did not see/couldn’t find a broader statement in the paper.

In my view, there are issues that may arise from the publication and adoption of the proposed AI transparency index. One issue is possible over-interpretation of the index, especially in fields and communities that are less technical. Once the index is defined and each model is reduced to a number, some people may infer understanding of the models while the number only measures disclosure coverage. Have authors thought of ways to reduce such misunderstandings from their proposed index? Do they have any concerns about this that they would like to include in a Broader Impacts statement.

**Claims And Evidence:**

Yes

**Claims Explanation:**

Paper is clear about its method and builds upon the FMTI for 2024 which appears to be an improvement in terms of methods and procedures.

**Requested Changes:**

Here I provide a critique. I hope author will find it helpful to improve their paper.



- One of the shortcomings, in my view, is the use of binary indicators. In the Appendix A, paper talks about a “binary score” automatically measured about whether evidence is found or not. Binary indicators have the potential to throw away information.A one-paragraph vague statement and detailed statement can both receive a binary 1 score. It is not clear to me where exactly the line is drawn between 0 and 1 for such an indicator.

- Such ambiguity also opens the door for gaming the transparency index by companies providing vague and unhelpful disclosures while boosting the transparency index of their models.

- The weighting used in the transparency index is subjective and normative. It mixes various scores and it is not clear to me whether choosing other weights could have led to a more informative index. Two people may agree that a transparency index is useful and necessary, yet disagree on whether training-data provenance should outweigh policy documents. I did not see any sensitivity analysis regarding the weights.

- Paper does not make an effort to distinguish between technical transparency and corporate transparency.

- Thinking from the perspective of the companies, being transparent can also hurt the companies in some instances, especially when thinking at the global stage and rivalry between the countries. For example, releasing training sets by a company can enable other companies to easily take away profits from that company. Similarly, if a company has a method for generating training data, being transparent about that method may also be disadvantage. Paper does not make it clear whether

- How this paper’s view of transparency translate to other technical fields? For example, if Boeing finds a material that is advantageous for building airplanes, should it be transparent about the material or should it keep this information to itself and use that to gain a larger share of the market?

- The burden of generating documents to be considered transparent can be significant for small companies. Many small companies might not be able to shoulder such burden, yet they might be able to provide useful information about their models. It is not clear to me whether this index can make small companies hopeless about gaining a good transparency score and completely give up about providing any kind of information about their foundation models.

---

> ### Author Response · Authors · 2026-03-06
> **Review Response**
>
> We appreciate that the reviewer finds that we improve upon the methods for FMTI 2024 and that our work is helpful for both the research and use of foundation models.
>
> > One of the shortcomings, in my view, is the use of binary indicators. In the Appendix A, paper talks about a “binary score” automatically measured about whether evidence is found or not. Binary indicators have the potential to throw away information.A one-paragraph vague statement and detailed statement can both receive a binary 1 score. It is not clear to me where exactly the line is drawn between 0 and 1 for such an indicator.
>
> > Such ambiguity also opens the door for gaming the transparency index by companies providing vague and unhelpful disclosures while boosting the transparency index of their models.
>
> We add an Appendix with the complete formal scoring specifications. As discussed in Section 3.3, a major set of changes we make are to tune the scoring criteria (using feedback from previous editions) to ensure that the disclosures are meaningful. For example, in previous editions we find that disclosures for the “model capabilities” indicator often were not meaningful (e.g., “text generation”), so in the newest edition we adjust the criteria to require developers to disclose capabilities specifically optimized during post-training.
>
> > The weighting used in the transparency index is subjective and normative. It mixes various scores and it is not clear to me whether choosing other weights could have led to a more informative index. Two people may agree that a transparency index is useful and necessary, yet disagree on whether training-data provenance should outweigh policy documents. I did not see any sensitivity analysis regarding the weights.
>
> We agree that there can be substantial disagreement regarding which subsets of the index are most important. Because what one considers the most informative depends on one one’s specific set of priorities, we opt to instead weigh each domain equally and release the underlying data publicly, allowing others to adjust the weighting more granularly as they see fit.
>
> > Thinking from the perspective of the companies, being transparent can also hurt the companies in some instances, especially when thinking at the global stage and rivalry between the countries. For example, releasing training sets by a company can enable other companies to easily take away profits from that company. Similarly, if a company has a method for generating training data, being transparent about that method may also be disadvantage.
>
> We agree that there are reasons why a company may not want to disclose information. Our claim is not that it would always be in the “company’s competitive interests” to disclose information but rather that this information is important for the public good.
>
> We do note, however, that it’s far from infeasible for a for-profit company to score highly, with IBM scoring a point on 95 out of 100 indicators.
>
> > The burden of generating documents to be considered transparent can be significant for small companies. Many small companies might not be able to shoulder such burden, yet they might be able to provide useful information about their models. It is not clear to me whether this index can make small companies hopeless about gaining a good transparency score and completely give up about providing any kind of information about their foundation models.
>
> Empirically, we find that being a smaller company is not a barrier to scoring well on the index. In fact, the second and third highest scoring companies are both startups.
>
> > I did not see/couldn’t find a broader statement in the paper.
>
> > In my view, there are issues that may arise from the publication and adoption of the proposed AI transparency index. One issue is possible over-interpretation of the index, especially in fields and communities that are less technical. Once the index is defined and each model is reduced to a number, some people may infer understanding of the models while the number only measures disclosure coverage. Have authors thought of ways to reduce such misunderstandings from their proposed index? Do they have any concerns about this that they would like to include in a Broader Impacts statement.
>
> Thank you for pointing this out - we add a Broader Impact Statement which includes a discussion of risks that may result from an overly broad interpretation of transparency scores.

---

### Review · Reviewer_9HTx · 2026-02-23

**Summary Of Contributions:**

This paper presents the 2025 edition of the Foundation Model Transparency Index (FMTI), a large-scale measurement framework for evaluating transparency practices of major foundation model developers across 100 indicators spanning upstream, model, and downstream stages. A key contribution is the substantial redesign of the index, including many new indicators and more operationalized criteria for reproducibility and disclosure, alongside expanded coverage that includes Chinese developers and a broader set of organizational and governance practices.

**Audience:**

Yes

**Audience Explanation:**

This paper addresses an issue of broad and growing importance to the TMLR audience, namely how to measure and compare transparency practices in foundation model development in a systematic and reproducible way. Even though the contribution is more measurement and governance oriented than algorithmic, it is highly relevant to researchers working on AI evaluation, responsible AI, AI policy, model documentation, auditing, and deployment standards. The index and its methodology can also serve as a useful empirical resource for future research on transparency, accountability, and governance interventions. The paper is therefore likely to be of interest to a meaningful subset of TMLR readers, especially those working at the interface of ML systems and societal impact.

**Broader Impact Concerns:**

The paper is socially important and generally beneficial, but a few broader-impact issues deserve stronger treatment.
1. the scoring design may inadvertently incentivize performative or surface-level disclosure if firms can gain points without meaningfully improving transparency quality, especially in cases where disclosure of “lack of information” is rewarded; the paper should discuss this risk and consider separating dependency-tracking from fulfilled transparency.

2. Transparency requirements create real tradeoffs involving intellectual property, security, and misuse risk, and the paper should better articulate how it balances societal transparency benefits against legitimate constraints (for example, tiered disclosure, delayed disclosure, or third-party audit models).

3. Cross-jurisdiction and cross-language differences in disclosure norms, legal regimes, and evidence availability could bias comparisons, so the paper should discuss parity of evidence collection and verification across regions, especially for non-English sources.

**Claims And Evidence:**

Yes

**Claims Explanation:**

The submission appears to provide substantial evidence for its descriptive findings through a structured indicator framework, documented scoring process, and broad evidence collection across firms, and many of its central claims about topic-level transparency gaps are plausible and well aligned with the reported measurements.

However, several stronger claims would benefit from additional methodological support, especially claims involving year-to-year comparisons under a substantially revised indicator set and narrative explanations of observed patterns across firms. In particular, the paper should more clearly establish longitudinal comparability, report reliability statistics and adjudication procedures, and provide robustness analyses for scoring and aggregation choices. These issues do not negate the value of the work, but they do limit how confidently some conclusions can be interpreted in the current version.

**Requested Changes:**

**Critical for acceptance**
- Complete and formal scoring specification (including binary/graded scoring rules, partial credit criteria, N/A handling, and aggregation weights), report inter-rater reliability metrics and the adjudication protocol, and add robustness analyses such as sensitivity to weighting/threshold choices and uncertainty estimates (for example, bootstrap confidence intervals for firm scores).
- Address longitudinal comparability more explicitly by either (a) reframing 2025 as a new baseline or (b) providing anchored year-over-year analyses on a stable subset of indicators, plus stronger discussion of construct consistency across editions.
- analyze possible bias from heterogeneous evidence-gathering modes (firm-submitted vs manually/agent-compiled evidence).

**Good to have**
- add quantitative support for narrative claims (similarity/clustering/regression analyses)
- expand discussion of measurement validity with references to psychometrics/measurement theory
- release the evidence corpus, scoring sheets, and code/prompts to support external auditability and reuse.

---

> ### Author Response · Authors · 2026-03-06
> **Review Response**
>
> We appreciate that the reviewer finds that we provide substantial evidence for our descriptive findings and that we address issues of broad importance to the TMLR audience.
>
> > In particular, the paper should more clearly establish longitudinal comparability, report reliability statistics and adjudication procedures, and provide robustness analyses for scoring and aggregation choices. These issues do not negate the value of the work, but they do limit how confidently some conclusions can be interpreted in the current version.
>
> > Complete and formal scoring specification (including binary/graded scoring rules, partial credit criteria, N/A handling, and aggregation weights), report inter-rater reliability metrics and the adjudication protocol, and add robustness analyses such as sensitivity to weighting/threshold choices and uncertainty estimates (for example, bootstrap confidence intervals for firm scores).
>
> We report reliability statistics (agreement rate) along with adjudication procedures (both between FMTI team members & between the FMTI team and companies) in Section 4.4.
>
> To clarify the scoring, we add an Appendix with the complete formal scoring specifications for each of the indicators. Additionally, all scores are binary, there are no N/A samples (indicators that are not applicable to a company are handled according to the specification given in the scoring specifications), and all indicators are given equal weight.
>
> We appreciate the suggestion to estimate uncertainty via bootstrapping: below is the 95% CI and median ranks, calculated through bootstrap samples of the indicators. Our overall rankings are stable: for example, the width of the 95% CI is three rank positions or fewer for 9 of the 13 evaluated companies.
>
> | company | median_rank | (rank_lower, rank_upper) |
> | --- | --- | --- |
> | IBM | 1.0 | (1.0, 1.0) |
> | Writer | 2.0 | (2.0, 3.0) |
> | AI21 Labs | 3.0 | (2.0, 3.0) |
> | Anthropic | 4.0 | (4.0, 6.0) |
> | Google | 5.0 | (4.0, 7.0) |
> | Amazon | 6.0 | (4.0, 8.0) |
> | OpenAI | 7.0 | (5.0, 10.0) |
> | DeepSeek | 8.0 | (5.0, 9.5) |
> | Meta | 8.5 | (6.0, 10.0) |
> | Alibaba | 10.0 | (8.0, 10.5) |
> | Mistral | 11.0 | (10.0, 13.0) |
> | Midjourney | 12.5 | (11.0, 13.0) |
> | xAI | 12.5 | (11.0, 13.0) |
>
> > Address longitudinal comparability more explicitly by either (a) reframing 2025 as a new baseline or (b) providing anchored year-over-year analyses on a stable subset of indicators, plus stronger discussion of construct consistency across editions.
>
> We agree that changes in scoring methodology requires a more careful consideration of longitudinal analyses. In our analysis, we anchor our comparisons against aspects of our index that stays consistent: in Section 5.2, we perform our comparisons across a fixed set of 9 developers; in Table 3 we consider regressions across indicators that are consistent across time; and, finally, from Page 35-38, we conduct a qualitative analysis of changes where we can similarly control for changes in indicators when analyzing how disclosures change.
>
> > analyze possible bias from heterogeneous evidence-gathering modes (firm-submitted vs manually/agent-compiled evidence).
>
> We analyze the difference between scores from reports prepared by the FMTI team versus by the developer in Section 5.2: reports prepared by the FMTI team score half that of the company prepared reports. This does create a bias towards reports prepared by developers, but this bias is intentional: from the 2024 FMTI, we know that engaging with companies directly allows for the public disclosure of more information (i.e., increased transparency). We choose to prioritize this over consistent scoring across companies.

---

> ### Author Response · Authors · 2026-03-06
> **Review Response (pt. 2)**
>
> > the scoring design may inadvertently incentivize performative or surface-level disclosure if firms can gain points without meaningfully improving transparency quality, especially in cases where disclosure of “lack of information” is rewarded; the paper should discuss this risk and consider separating dependency-tracking from fulfilled transparency.
>
> We agree that surface-level disclosures are an issue. The primary defense is to design and define indicators to guarantee substantive transparency (e.g., disclosing the all-in cost to train a model is necessarily meaningful). The 2025 indicators were updated from the original FMTI indicators to ensure this. For example, past editions awarded points for non-meaningful disclosures for the “model capabilities” indicator (e.g., “text generation”), so the 2025 edition changes the criteria to require disclosure of capabilities specifically optimized during post-training. See Section 3.3 for further discussion of this.
>
> Finally, in Section 3.3, we also discuss how we handle disclosures when the developers lack information as this was also an issue observed in previous editions. For example, the indicator for “Energy usage for the final training run” includes as a part of its scoring criteria: “If the developer is unable to measure or estimate this quantity due to information not being available from another party (e.g. compute provider), we will award this point if the developer explicitly discloses what information it lacks and why it lacks it.”
>
> > Transparency requirements create real tradeoffs involving intellectual property, security, and misuse risk, and the paper should better articulate how it balances societal transparency benefits against legitimate constraints (for example, tiered disclosure, delayed disclosure, or third-party audit models).
>
> Our work does this by clarifying where trade-offs exist rather than assuming they must exist for all indicators, which is not true. We believe there are opportunities for strict Pareto improvement: some indicators can be disclosed relative to the status quo with no cost to any of these other interests. This should be expected in a nascent industry: foundation model developers are not a mature/long-standing industry and their information disclosure practices have not necessarily been individually or collectively well-considered. In some cases, the 2025 Index makes this explicit: companies like Google and Anthropic provide indicator-level justifications for their non-disclosures in some cases.
>
> We design our indicators sensitive to competing factors. For example, for several indicators, we only require that the developer disclose the head of the distribution. Additionally, for the indicator on model-theft prevention, we find that even companies who justify opacity on other indicators on the grounds of security still receive a point for this indicator (see Page 23-24).
>
> Finally, we recognize that other scoring schemes may address some of these issues, but they also create tradeoffs that we find to be undesirable. For example, a key feature of FMTI is the contents of the disclosures themselves: delayed disclosures and third-party audit models prevent the timely release of this information. Also, although tiered disclosures allows for more nuanced scoring it adds a significant amount of complexity (both in scoring and in the indicator definitions themselves). As discussed in the paper, a key purpose of FMTI is as a way to clarify the currently imprecise and nebulous concept of transparency in relation to foundation model developers. Binary indicators define a clearer threshold for what should and should not be disclosed.
>
> > Cross-jurisdiction and cross-language differences in disclosure norms, legal regimes, and evidence availability could bias comparisons, so the paper should discuss parity of evidence collection and verification across regions, especially for non-English sources.
>
> We’re aware that cross-regional differences may bias comparisons, especially for non-English sources. As such, a Chinese-speaking author of this paper reviewed these sources for the relevant indicators: we discuss this in Section 4.2 on Information Gathering.

---

### Decision · Action_Editor_Wuou · 2026-04-22

**Recommendation:** Accept with minor revision

**Additional Comments:**

After carefully reading the reviews and the author's answers, I believe the manuscript should include the following two minor revisions:

- The reliability statistics (agreement rate) along with adjudication procedures.
- The following rationale (the authors discussed in the answers), in the main text: “We analyze the difference between scores from reports prepared by the FMTI team versus by the developer in Section 5.2: reports prepared by the FMTI team score half that of the company prepared reports. This does create a bias towards reports prepared by developers, but this bias is intentional: from the 2024 FMTI, we know that engaging with companies directly allows for the public disclosure of more information (i.e., increased transparency). We choose to prioritize this over consistent scoring across companies.”

**Audience:**

Yes

**Audience Explanation:**

The audience of TMLR would globally be interested in an independent evaluation of the transparency of the foundation models, of course.

**Claims And Evidence:**

Yes

**Claims Explanation:**

This paper presents the 2025 FM Transparency Index, and represents a huge effort on the side of the authors. There are not particular claims made, rather than proposing an updated methodology for computing the FMTI, and enough evidence is reported.

The only important limitation (that could be in part related to the claims/evidence) is the fact that only companies engaging with the authoring team are represented in the FMTI. While this is needed to provide an extensive evaluation, it also limits the number of companies represented in the paper. One reviewer proposes to use public information that is available for many more companies. I would urge the authors to implement this for the next edition of the FMTI (meaning the 2026 one).